# ADDAX: UTILIZING ZEROTH-ORDER GRADIENTS TO IMPROVE MEMORY EFFICIENCY AND PERFORMANCE OF SGD FOR FINE-TUNING LANGUAGE MODELS

**Zeman Li**
USC
zemanli@usc.edu

**Xinwei Zhang**
USC
xinweiz@usc.edu

**Peilin Zhong**
Google Research
peilinz@google.com

**Yuan Deng**
Google Research
dengyuan@google.com

**Meisam Razaviyayn**
USC
razaviya@usc.edu

**Vahab Mirrokni**
Google Research
mirrokni@google.com

## ABSTRACT

Fine-tuning language models (LMs) with the standard Adam optimizer often demands excessive memory, limiting accessibility. The "in-place" version of Stochastic Gradient Descent (IP-SGD) and Memory-Efficient Zeroth-order Optimizer (MeZO) have been proposed as solutions to improve memory efficiency. However, IP-SGD still requires decent amount of memory, and MeZO suffers from slow convergence and degraded final performance due to its zeroth-order nature. This paper introduces *Addax*, a novel method that improves both memory efficiency and algorithm performance of IP-SGD by integrating it with MeZO. Specifically, Addax computes the zeroth-order or first-order gradient of the data points in the minibatch based on their memory consumption and combines zeroth- and first-order gradient estimates to obtain the updated direction in each step. By computing the zeroth-order order gradient of data points that require more memory and the first-order gradient of the ones that require less memory, Addax overcomes the slow convergence of MeZO and excessive memory requirement of IP-SGD. Additionally, the zeroth-order gradient acts as a regularizer for the first-order gradient, further enhancing the model's final performance. Theoretically, we establish the convergence of Addax under mild assumptions, demonstrating faster convergence and less restrictive hyper-parameter choices than MeZO. Our extensive experiments with diverse LMs and tasks show that Addax consistently outperforms MeZO in terms of accuracy and convergence speed, while having a comparable memory footprint. In particular, our experiments using one A100 GPU on OPT-13B model reveal that, on average, Addax outperforms MeZO in terms of accuracy/F1 score by 14%, and runs $15\times$ faster, while having a comparable memory footprint to MeZO. In our experiments on the larger OPT-30B model, on average, Addax outperforms MeZO in terms of accuracy/F1 score by $> 16\%$ and runs $30\times$ faster on a single H100 GPU. Moreover, Addax surpasses the performance of standard fine-tuning approaches, such as IP-SGD and Adam, in most tasks in terms of Accuracy/F1 score with significantly less memory requirement.

## 1 INTRODUCTION

Fine-tuning pre-trained language models (LMs) is a crucial step in a wide range of natural language processing tasks, including text classification and sentiment analysis (Devlin et al., 2019), as well as their use in different domains (Gururangan et al., 2020; Schwartz et al., 2019; Lu et al., 2019). However, standard fine-tuning approaches (with Adam optimizer) demand excessive memory requirement due to gradient and/or the optimizer state storage, presenting a challenge as LMs grow in scale (Brown et al., 2020; OpenAI, 2023). For instance, fine-tuning a 13-billion-parameter model like OPT (Zhang et al., 2022) in mixed precision requires over 316 GB of memory, hindering accessibility for researchers and practitioners with limited resources and specialized hardware.

Recently, various memory-efficient methods for fine-tuning LMs have been proposed. In-context learning (ICL) utilizes a single *inference pass*, incorporating label examples in its context for prediction (Brown et al., 2020). Despite its limited success, ICL's performance is shown to be less effective than (Adam) fine-tuning for medium-sized LMs (Brown et al., 2020). As an alternative approach, Parameter-Efficient Fine-Tuning (PEFT) selectively tunes a fraction of the network while freezing the rest of the parameters, and significantly reduces the *parameters* needed for fine-tuning (Hu et al., 2022; Li and Liang, 2021; Lester et al., 2021). Despite its efficiency, fine-tuning LMs with PEFT may still require more memory than model inference. For example, fine-tuning OPT-13B with Adam with a batch size of 8 requires $4\times$H100 GPUs (316GB total), whereas utilizing PEFT decreases this to $2\times$H100 GPUs (158GB total) with a batch size of 16 (Brown et al., 2020). Nonetheless, this requirement is still $6\times$ greater than 25GB needed for model inference.

Another approach for memory-efficient fine-tuning is to lessen the memory footprint of the optimizer. Recently, the Memory-Efficient Zeroth-order Optimizer (MeZO) is proposed by Malladi et al. (2023). MeZO generates gradient estimators solely through forward passes with minimal memory overhead. Unlike classical zeroth-order optimization method ZO-SGD (Spall, 1992), MeZO allows in-place perturbation of model parameters to avoid storing the perturbation vector. One desirable property of MeZO (and in general approaches that reduce the memory overhead of the optimizers) is that it can be combined with other methods, such as PEFT. Moreover, Malladi et al. (2023) showed that the memory footprint of MeZO can be $12\times$ lower than Adam. While being memory efficient, *MeZO suffers from 1) slow convergence rate compared to standard fine-tuning methods such as Adam; and 2) possible degradation of the performance (e.g. accuracy) of the fine-tuned model compared to Adam* (see the experiments in Malladi et al. (2023) and Table 12). Observing these drawbacks, it is natural to ask:

> **Question:** Can we develop an optimizer for fine-tuning language models (LMs) that requires significantly less memory than standard Adam but still enjoys fast convergence and produces high-quality fine-tuned models?

In an effort to answer this question, this work proposes Addax (***ADD**ition of gr**A**dient estimates through memory-efficient e**X**ecution*), a method that: **i)** is memory efficient, **ii)** has fast convergence speed and **iii)** achieves the best performance across a wide range of fine-tuning methods and tasks. Our specific contributions are as follows:

1. **Algorithm design.** We develop Addax, a novel approach that cleverly assigns different batches of data to MeZO or in-place SGD (IP-SGD), and combines the computed gradients. This strategic assignment of data points, based on input length, allows Addax to maintain a memory footprint comparable to MeZO while achieving significant improvements. Specifically, Addax accelerates MeZO's convergence rate and boosts the final model's performance, effectively overcoming the limitations of zeroth-order optimization. Our design is driven by two novel observations: 1) computing gradients for different data points requires varying memory, and 2) integrating zeroth-order updates with first-order methods enhances the quality of the fine-tuned model.

2. **Theoretical analysis.** Theoretically, we analyze Addax's convergence under mild assumptions in two scenarios: with and without data assignment procedure. Unlike MeZO, the convergence rate of Addax is independent of model size without requiring the assumption of low-rankness of the Hessian. Additionally, we show that the hyperparameters for Addax are less restrictive compared to those of MeZO.

3. **Numerical comparisons.** We perform comprehensive experiments over a broad range of model architectures (e.g., masked LM and autoregressive LM), model scales ranging from 350M to 70B parameters, and tasks including classification, multiple-choice questions, and content generation. Compared to SGD and IP-SGD, Addax has a lower memory footprint and is on par with MeZO. Using a single A100 (40GB) GPU, Addax successfully fine-tunes the OPT-13B model on all nine tasks, while SGD fails on all and IP-SGD fails on three due to memory limitations (see Figure 1). Addax surpasses MeZO by 14% in accuracy/F1 and converges $15\times$ faster on average, with a similar memory footprint. Furthermore, Addax outperforms Adam in seven out of nine tasks while reduces up to 89% the memory. When fine-tuning larger model like OPT-30B with a single H100 (80GB) GPU, Addax achieves superior performance over IP-SGD and MeZO on various tasks, e.g. test accuracy 16% higher than MeZO, 4.8% higher than IP-SGD when IP-SGD applicable

(see Figure 2 and Table 1). Similar results are observed when fine-tuning Addax on OPT-66B and Llama-2-70B using three H100 (240GB total) GPUs (See Table 2 and 3).

Next, we will discuss preliminaries, and will leave further discussions on prior work to Appendix C.

## 2 NOTATIONS AND PRELIMINARIES

### 2.1 NOTATIONS

We are interested in optimizing the (smooth and possibly non-convex) loss function

$$\min_{\boldsymbol{\theta} \in \mathbb{R}^d} \left( \mathcal{L}(\boldsymbol{\theta}) := \mathbb{E}_{x \sim \mathcal{D}} \left[ \ell(\boldsymbol{\theta}; x) \right] \right), \tag{1}$$

parameterized by $\boldsymbol{\theta} \in \mathbb{R}^d$, where $\mathcal{D}$ denotes the (fine-tuning) data distribution and $x$ denotes the (fine-tuning) data point. Throughout, we mark the values related to zeroth- and first-order gradient with $(\cdot)^0, (\cdot)^1$, respectively, and denote the iteration and coordinate indices as $(\cdot)_t, (\cdot)_i$, where $t \in \{0, \ldots, T\}, i \in \{1, \ldots, d\}$. In addition, we assume $\boldsymbol{\theta}$ parameterized an $M$-layer network, i.e., $\boldsymbol{\theta} = (\boldsymbol{\theta}_1, \ldots, \boldsymbol{\theta}_M)$, where $\boldsymbol{\theta}_m \in \mathbb{R}^{d_m}$ and $\sum_{m=1}^M d_m = d$. The maximum sequence length in the dataset $\mathcal{D}$ is denoted by $L_{max}$. We slightly abuse the notation and denote the loss evaluated on a minibatch $\mathcal{B}$ as $\mathcal{L}(\boldsymbol{\theta}; \mathcal{B}) \triangleq \frac{1}{B} \sum_{x \in \mathcal{B}} \ell(\boldsymbol{\theta}; x)$.

### 2.2 MEMORY-EFFICIENT ZEROTH-ORDER OPTIMIZER

MeZO (Malladi et al., 2023) is a memory-efficient fine-tuning approach based on the zeroth-order stochastic gradient descent method (ZO-SGD). The update step of MeZO follows SGD, i.e., $\boldsymbol{\theta}_{t+1} = \boldsymbol{\theta}_t - \eta \widehat{\nabla} \mathcal{L} \left( \boldsymbol{\theta}_t; \mathcal{B}_t^0 \right)$, with the key difference of using zeroth-order gradient $\widehat{\nabla} \mathcal{L} \left( \boldsymbol{\theta}_t; \mathcal{B}_t^0 \right)$ estimated on minibatch $\mathcal{B}_t^0$, instead of first-order gradient $\nabla \mathcal{L} \left( \boldsymbol{\theta}_t; \mathcal{B}_t^1 \right)$. The zeroth-order gradient is estimated using Simultaneous Perturbation Stochastic Approximation (SPSA, (Spall, 1992)):

$$\widehat{\nabla} \mathcal{L}(\boldsymbol{\theta}; \mathcal{B}) = \frac{\mathcal{L}(\boldsymbol{\theta} + \epsilon \mathbf{z}; \mathcal{B}) - \mathcal{L}(\boldsymbol{\theta} - \epsilon \mathbf{z}; \mathcal{B})}{2\epsilon} \mathbf{z}, \tag{2}$$

where $\mathbf{z}$ is a random search direction, e.g., $\mathbf{z} \sim \mathcal{N}(\mathbf{0}, \mathbf{I})$. In ZO-SGD, SPSA requires generating and storing $\mathbf{z} \in \mathbb{R}^d$ for multiple times of model perturbation, $\boldsymbol{\theta} + \epsilon \mathbf{z}, \boldsymbol{\theta} - \epsilon \mathbf{z}$, which is memory inefficient. MeZO reduces memory consumption in the implementation of SPAS by only storing the random seed that generates $\mathbf{z}$, reducing memory consumption from $\mathcal{O}(d)$ to $\mathcal{O}(1)$.

However, due to the inherent bias and higher noise level in zeroth-order gradient estimate (Nesterov and Spokoiny, 2017), MeZO suffers from slower convergence and worse final model performance compared to first-order methods, c.f. Malladi et al. (2023, Table 1), and the experiments in section 4.

### 2.3 PRELIMINARIES AND MAJOR OBSERVATIONS

In this subsection, we provide the key observations behind the development of Addax:

**SGD can match the performance of Adam in fine-tuning tasks.** Although Adam and AdamW (Kingma and Ba, 2015; Loshchilov and Hutter, 2019) have demonstrated better performance than SGD in training deep learning models from scratch, they require additional memory to store optimizer states, which is unfavorable in fine-tuning LMs. On the other hand, it has been observed that SGD achieves a comparable performance to Adam in fine-tuning tasks (see, e.g., Zhang et al. (2024a) and Lv et al. (2023)). This is observation is also re-confirmed in our experiments showing that fine-tuning LMs using (16-bit) SGD can have comparable performance to fine-tuning with (32-bit) Adam. For example, in the right panel of Figure 3, in three fine-tuning tasks of RTE, CB, and COPA (Wang et al., 2019), SGD leads to higher performance while requiring significantly less memory. The success of using SGD in fine-tuning LMs is attributed to the relative "nice" landscape of fine-tuning tasks of LMs (Hao et al., 2019; Zhang et al., 2024a).

**Memory-efficient implementation of SGD via "in-place" updates.** To reduce the memory footprint of SGD, several studies have explored the utilization of *in-place (IP) updates* during backward propagation (Zhao et al., 2024; Lv et al., 2023). Instead of separating the backward propagation and weight update steps, which require storing the gradients for all layers, in-place SGD (IP-SGD)

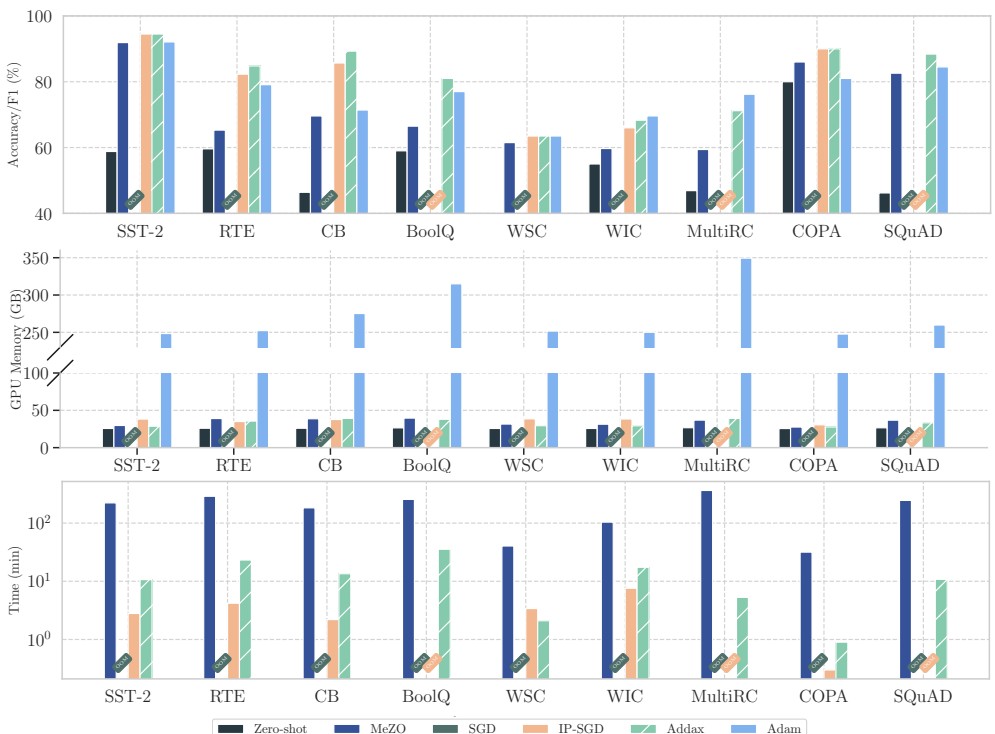

Figure 1: Accuracy/F-1 score, memory, and convergence time resulted from fine-tuning OPT-13B model with various algorithms on one A100 (40GB) GPU, except for Adam, which runs on five GPUs. The label "OOM" means the run encounters out-of-memory error during fine-tuning even with smallest batch size. Addax consistently outperforms other methods in terms of Accuracy, with GPU memory consumption comparable to MeZO. Except for Adam, all other methods are running in 16-bit mode. We do not report the time for Adam as it requires five GPUs. The exact numbers can be found in Table 12 in Appendix F.1.

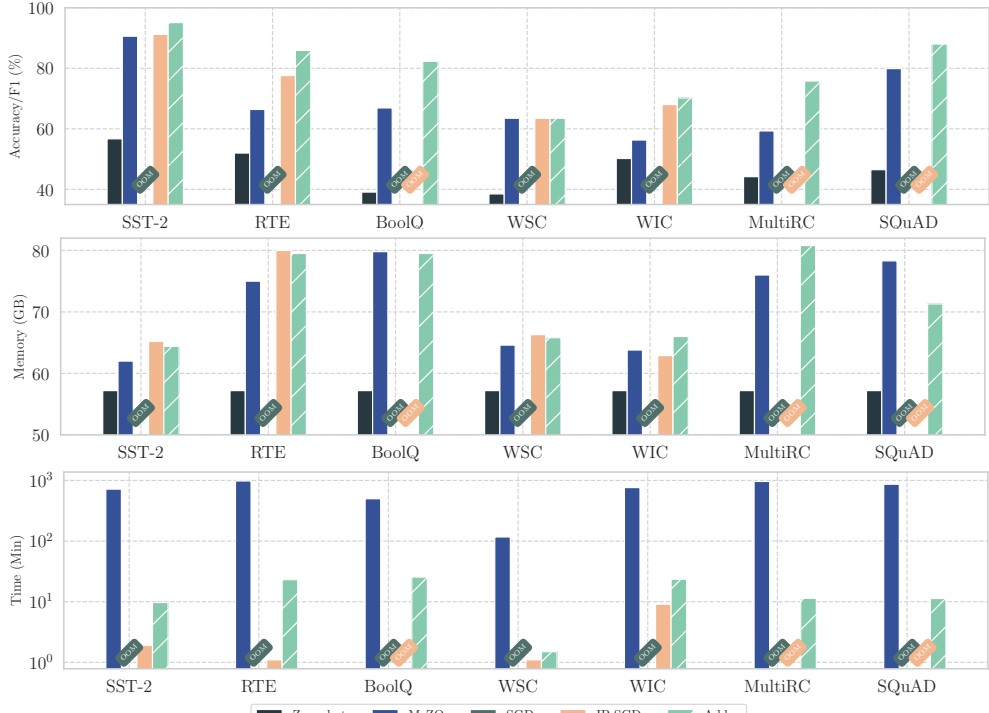

Figure 2: Accuracy/F-1 score, memory, and convergence time resulted from fine-tuning OPT-30B model with various algorithms on one H100 (80GB) GPU. The label "OOM" means the run encounters out-of-memory error during fine-tuning. Addax leads to the best final accuracy in all experiments, have comparable memory footprint to MeZO, while converges orders of magnitude faster. The exact numbers related to this figure can be found in Table 15 in Appendix F.2.

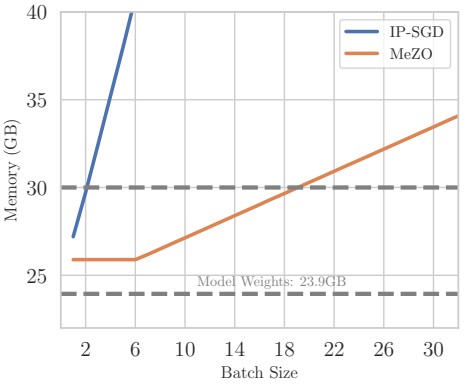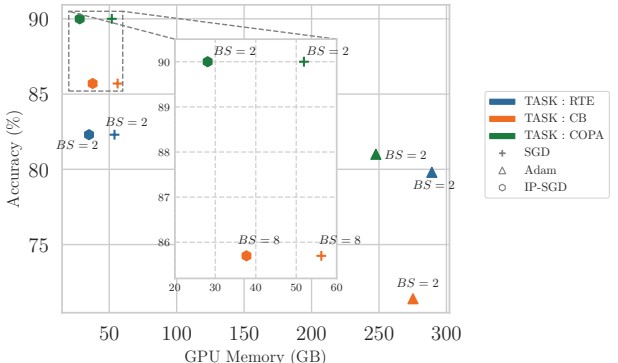

Figure 3: **Left**: Memory profile of fine-tuning OPT-13B with IP-SGD and MeZO on a synthetic dataset with a fixed sequence length of 300. **Right:** Fine-tuning OPT-13B using IP-SGD and small batch sizes (BS) can outperform Adam, while consuming significantly lower memory.

combines the two steps by updating the weights in each layer as soon as the gradients are computed, and immediately discards the gradient after it has been used. In-place update avoids storing the gradients of the full model and, thus, significantly reduces the memory footprint of SGD.

Though improving over SGD, IP-SGD may still require more memory than MeZO. To compare the memory consumption of IP-SGD and MeZO, we record the memory consumption as a function of the batch size in the left panel of Figure 3. As illustrated in the figure, with a memory constraint of 30GB, we can use a batch size of 18 for running MeZO, while we can only use the batch size of 2 for running IP-SGD.

**Memory required for computing gradient depends on the input sequence length.** We observe that *in fine-tuning LM models, the memory required for first-order gradient estimation depends on the maximum input sequence length (in tokens).* In Figure 4, we record the memory consumption of SGD, IP-SGD, and MeZO with fixed batch size and varying input sequence length. As illustrated in the figure, the memory consumption of all algorithms increases as the input sequence length increases, and the memory increase of IP-SGD is much faster than MeZO. Given the fact that the input sequence lengths vary a lot in a given fine-tuning task across different data points (see Figure 6 in Appendix D), we conclude that the memory inefficiency of IP-SGD compared with MeZO mainly attributes to the data points with longer sequence length in the dataset.

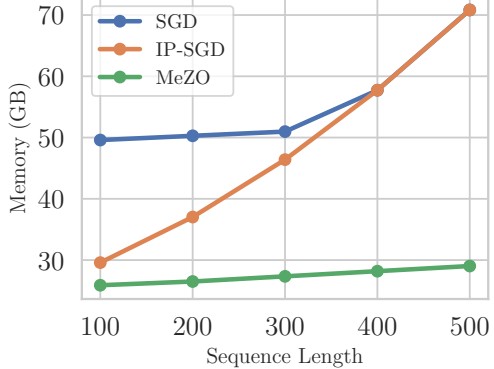

Figure 4: Memory profiling of SGD, IP-SGD, and MeZO on OPT-13B fine-tuning with synthetic datasets with varying sequence lengths (fixing batch size = 8).

With the above observations in hand, we discuss our algorithm, Addax, in the next section.

## 3 ADDAX

Building on the insights from Section 2, we propose Addax, a method that assigns data batches to either MeZO or IP-SGD based on memory needs, and then combines their gradients. In memory-limited scenarios, Addax reduces memory usage by assigning shorter sequences to IP-SGD and longer ones to MeZO, leading to a similar convergence rate as IP-SGD, much faster than MeZO, while keeping a similar memory footprint, significantly less than IP-SGD. Furthermore, incorporating zeroth-order gradients with first-order gradients can enhance the final model performance. We first propose the Addax algorithm, provide its convergence properties, and finally discuss how incorporating the zeroth-order gradient benefits SGD in the final model performance.

## 3.1 Algorithm Overview

Addax first starts by partitioning the dataset into data points with long/short sequence length, i.e., $\mathcal{D} = \mathcal{D}^0 \cup \mathcal{D}^1$ where $\mathcal{D}^0 = \{x \in \mathcal{D} \,|\, \text{length}(x) > L_T\}$ and $\mathcal{D}^1 = \{x \in \mathcal{D} \,|\, \text{length}(x) \leq L_T\}$ for some given threshold hyper-parameter $L_T$, where $\text{length}(\cdot)$ measures the sequence length of a data point. As discussed in Section 2.3, computing gradient on data points in $\mathcal{D}^1$ requires much less memory than computing the gradients in $\mathcal{D}$ because the maximum input sequence length of $\mathcal{D}^1$ is capped by $L_T$ which is smaller than $L_{max}$. With this data partition, Addax computes the zeroth-order gradient on the data points with long sequence length in $\mathcal{D}^0$ while computing the first-order gradients for the data points with short sequence length in $\mathcal{D}^1$, resulting in a considerable memory reduction compared with IP-SGD. In particular, at each iteration of the algorithm, Addax first draws a random batch $\mathcal{B}^0$ (with $|\mathcal{B}^0| = K^0$) of data points from $\mathcal{D}^0$ and a random search direction $\mathbf{z} \in \mathbb{R}^d$ with $\mathbb{E}[\mathbf{z}] = 0$ and $\mathbb{E}[\mathbf{z}\mathbf{z}^T] = \mathbf{I}$. Then, using the drawn samples, it utilizes SPSA to estimate the directional derivative of the objective function in the direction $\mathbf{z}$ at the point $\boldsymbol{\theta}$ with a small perturbation size $\epsilon$:

$$g^0 = \frac{1}{K^0} \sum_{x \in \mathcal{B}^0} \frac{\ell(\boldsymbol{\theta} + \epsilon\mathbf{z}; x) - \ell(\boldsymbol{\theta} - \epsilon\mathbf{z}; x)}{2\epsilon}.$$

Then, Addax draws a random batch $\mathcal{B}^1$ (with $|\mathcal{B}^1| = K^1$) of data from $\mathcal{D}^1$ and computes $\mathbf{g}^1 = \frac{1}{K^1} \sum_{x \in \mathcal{B}^1} \nabla\ell(\boldsymbol{\theta}; x)$. Finally, it updates the model parameters by:

$$\boldsymbol{\theta} \leftarrow \boldsymbol{\theta} - \eta\left(\alpha\mathbf{z}g^0 + (1-\alpha)\mathbf{g}^1\right), \tag{3}$$

where $\eta$ is the learning rate and $\alpha \in [0,1]$ is a mixing constant for combining the two gradient estimates. If one naïvely implements the update rule in equation 3, it needs to store the gradient $\mathbf{g}^1$ and the direction $\mathbf{z}$. However, as discussed in Section 2, one can perform the update rule in equation 3 in-place, without storing the values of $\mathbf{g}^1$ or $\mathbf{z}$. Such a memory-efficient implementation of Addax is described in algorithm 1. We leave further detailed discussions on this algorithm to Appendix A.

**Addax without data assignment (Addax-WA).** When sufficient memory is available to perform IP-SGD, one might argue that there is no need to use zeroth-order gradients. However, as demonstrated in our extensive experiments (see Figure 5 and Fine-tuning results for datasets SST-2, RTE, WSC of Addax ($L_T = 320$) in Table 15), incorporating zeroth-order gradients still results in better final accuracy for the fine-tuned model. Therefore, even when memory constraints are not a concern, we continue to utilize zeroth-order gradients by randomly selecting a data batch and combining its zeroth-order gradient with IP-SGD. This is achieved by setting $\mathcal{D}^0 \leftarrow \mathcal{D}$ and $\mathcal{D}^1 \leftarrow \mathcal{D}$ in step 3 of Algorithm 1. This version of the algorithm is referred to as *Addax-WA*.

Having introduced Addax, we would like to comment on a recent related work. Concurrently with our work, Zhang et al. (2024b) propose a "hybrid ZO-FO" fine-tuning scheme for LLMs, which also integrates zeroth- and first-order gradient estimates. However, their method differs significantly from ours. They limit backpropagation to the deeper layers and use zeroth-order optimization in the shallower layers. This approach prevents them from taking advantage of the memory savings offered by in-place update rules and does not harness the benefits of zeroth-order methods for improving the final model accuracy. In contrast, our approach utilizes in-place update rules, ensuring that memory usage does not scale significantly with model size and removing the need to limit backpropagation to specific layers. Furthermore, we allocate data to optimizers differently, leading to additional major memory savings.

## 3.2 Theoretical Analysis

Depending on whether step 3 or step 5 is executed in Algorithm 1, the algorithm follows two distinct trajectories. We analyze each case separately. We begin by discussing the convergence result of Addax when step 3 is applied, i.e., the Addax-WA algorithm. We provide only the informal version of the results, while the formal theorems can be found in Appendix G. We start by presenting the convergence of Addax in the general nonconvex setting:

**Theorem 3.1** (Informal). *Assume that the loss $\ell$ is Lipschitz smooth, and the first-order stochastic gradient is unbiased and has bounded variance. Choosing $\eta_t = \eta = \mathcal{O}(d^{-1/2}T^{-1/2})$ and $\epsilon = \mathcal{O}(d^{-3/4}T^{-1/4})$ in Addax leads to the convergence rate:*

$$\mathbb{E}\left[\|\nabla\mathcal{L}(\boldsymbol{\theta}_t)\|^2\right] = \mathcal{O}\left(\frac{1}{\sqrt{T}} \cdot \sqrt{\frac{(1-\alpha)^2}{K^1} + \frac{\alpha^2 d}{K^0}}\right).$$

---

**Algorithm 1** Addax: ADDition of grAdient estimates through memory-efficient eXecution

---
1: **Input:** $\boldsymbol{\theta}$ with $M$ Layers, $T, \mathcal{L}, L_T, L_{max}, K^0, K^1$, perturbation scale $\epsilon$, mixing parameter $\alpha \in [0, 1]$, and dataset $\mathcal{D}$.
2: **if** $L_T \geq L_{max}$ **then**
3: $\quad \mathcal{D}^0 \leftarrow \mathcal{D}, \mathcal{D}^1 \leftarrow \mathcal{D}$
4: **else**
5: $\quad \mathcal{D}^0 \leftarrow \{x \in \mathcal{D} \,|\, \text{length}(x) > L_T\}, \mathcal{D}^1 \leftarrow \{x \in \mathcal{D} \,|\, \text{length}(x) \leq L_T\}$
6: **for** $t \in \{0, 1, \cdots, T-1\}$ **do**
7: $\quad$ Randomly draw mini-batches $\mathcal{B}^0, \mathcal{B}^1$ uniformly from $\mathcal{D}^0, \mathcal{D}^1$ with $K^0, K^1$ samples.
8: $\quad (g^0, s) \leftarrow \textbf{ZerothGrad}(\boldsymbol{\theta}, \mathcal{L}, \mathcal{B}^0, \epsilon)$ (algorithm 2) $\qquad$ *# Estimate zeroth-order gradient*
9: $\quad$ **for** $m = M, \ldots, 1$ **do**
10: $\qquad \boldsymbol{g}_m^1 \leftarrow \frac{1}{K^1} \sum_{x \in \mathcal{B}^1} \nabla_{\boldsymbol{\theta}_m} \mathcal{L}(\boldsymbol{\theta}, x)$ $\qquad$ *# Estimate layer l first-order gradient*
11: $\qquad \boldsymbol{\theta}_m \leftarrow \boldsymbol{\theta}_m - \eta_t(1 - \alpha)\boldsymbol{g}_m^1$ $\qquad$ *# Update model parameters*
12: $\qquad \boldsymbol{g}_m^1 \leftarrow \text{None}$ $\qquad$ *# Clear gradients*
13: $\quad$ Reset random number generator with seed $s$
14: $\quad$ **for** $m = 1, \ldots, M$ **do**
15: $\qquad \boldsymbol{z}_m \sim \mathcal{N}(0, \mathbf{I}_{d_m})$
16: $\qquad \boldsymbol{\theta}_m \leftarrow \boldsymbol{\theta}_m - \eta_t \alpha g^0 \boldsymbol{z}_m$ $\qquad$ *# Update model parameters*
17: $\qquad \boldsymbol{z}_m \leftarrow \text{None}$
18: **Output:** $\boldsymbol{\theta}$

---

*Further, by choosing the optimal* $\alpha = \frac{K^0}{K^0 + dK^1}$, *we obtain the convergence rate* $\mathcal{O}\left(\sqrt{\frac{d}{T(K^0 + dK^1)}}\right)$.

The formal version of this Theorem and its proof can be found in Appendix G.3.

*Remark 1.* Compared with the convergence rate of zeroth-order methods Ghadimi and Lan (2013); Nesterov and Spokoiny (2017), the above convergence rate is nearly dimension-independent. In particular, the factor $\sqrt{\frac{d}{T(K^0 + dK^1)}}$ is upper bounded by $\sqrt{\frac{1}{TK^1}}$.

*Remark 2.* The restrictions on the choice of parameters for Addax are more relaxed than the ones in zeroth-order methods. In particular, ZO-SGD (or MeZO) requires choosing smaller parameters $\epsilon = \mathcal{O}(d^{-1}T^{-1/2})$ and $\eta = \mathcal{O}(1/\sqrt{dT})$ for guaranteeing convergence. These choices are clearly more restrictive than the choice of parameters in Theorem 3.1. Therefore, one can choose a larger learning rate in Addax compared to MeZO. This is also observed in our experiments (see Appendix D.5 for details).

*Remark 3.* Prior works observed that pre-trained LMs have a low effective rank Hessian (Aghajanyan et al., 2020; Li et al., 2018; Papyan, 2018; 2020). Under the assumption that the Hessian has a low effective rank, the dependency on the parameter dimension $d$ can be further improved (see Theorem G.12 in Appendix G).

Besides assuming the loss is nonconvex and smooth, we also provide the convergence of Addax for strongly convex and smooth loss functions:

**Theorem 3.2** (Informal). *Assume that* $\mathcal{L}$ *is strongly convex, per-sample loss* $\ell$ *is Lipschitz smooth, and the first-order stochastic gradients are unbiased and have bounded variance. Choosing* $\eta_t = \eta = \mathcal{O}(T^{-1} \ln(T))$ *and* $\epsilon = \mathcal{O}(\alpha^{1/2}T^{-1/2}d^{-1/2})$ *in Addax leads to the convergence rate:*

$$\mathbb{E}[\|\boldsymbol{\theta}_T - \boldsymbol{\theta}_\star\|^2] = \mathcal{O}\left(\frac{\ln(T)}{T}\left(\frac{(1-\alpha)^2}{K^1} + \frac{\alpha^2 d}{K^0}\right)\right)$$

*Further, by choosing the optimal* $\alpha = \frac{K^0}{K^0 + dK^1}$, *Addax converges with rate* $\mathcal{O}\left(\frac{\ln(T)d}{T(K^0 + dK^1)}\right)$.

The above results cover the case where step 3 is executed in Algorithm 1. When step 5 is executed, we can obtain similar results (see Theorem G.10 in Appendix G.4 for details).

## 3.3 FURTHER DISCUSSIONS ON THE BENEFITS OF UTILIZING ZEROTH-ORDER GRADIENTS

Beyond enhancing memory efficiency of IP-SGD by eliminating the need to compute gradients for input data with longer sequence lengths that are more memory-intensive, using zeroth-order

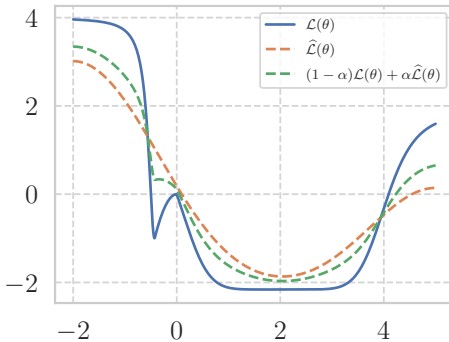 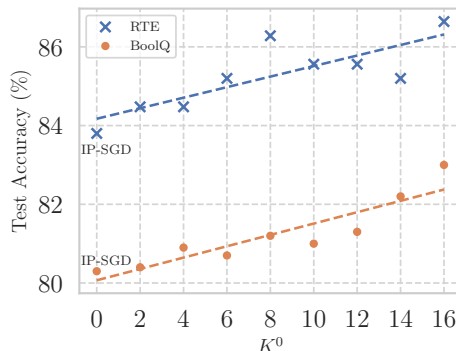

Figure 5: **Left**: An illustration of loss function $\mathcal{L}(\boldsymbol{\theta})$ alongside its Gaussian smoothed version $\widehat{\mathcal{L}}(\boldsymbol{\theta})$. Minimizing $(1-\alpha)\mathcal{L}(\boldsymbol{\theta}) + \alpha\widehat{\mathcal{L}}(\boldsymbol{\theta})$ can help escape sharp local minima and find higher quality solutions. **Right**: The regularization effect of zeroth-order gradient estimates on first-order gradient estimates. We fix $K^1 = 4$ in Addax across experiments while varying $K^0$ from 0 to 16. In the special case where $K^0 = 0$, Addax reduces to IP-SGD.

updates alongside first-order updates improve the final performance and accuracy of the fine-tuned model compared with vanilla IP-SGD. We hypothesize the following reasons for this improvement in performance:

**Zeroth-order updates may help escape spurious and sharp local minima.** It is known that zeroth-order gradient estimates are noisy estimates of the gradient (see Nesterov and Spokoiny (2017) and Lemma G.5 in Appendix G). It has also been observed that injecting noise into the gradient can be beneficial in nonconvex optimization. For example, Ge et al. (2015) and Jin et al. (2017) showed that adding noise to the gradient direction can help escape saddle points in nonconvex optimization. Moreover, Zhou et al. (2019) showed (both experimentally and theoretically) that injecting noise in the gradient direction can help the algorithm in escaping bad/spurious local minima. One can also argue that noise would help the algorithm in finding *flat minima* and avoid sharp local minima Liu et al. (2021). All these insights suggest that Addax can converge to better local minima. Our extensive experiments in Section 4 also show that Addax outperforms SGD in terms of the final performance of the fine-tuned model.

**Zeroth-order updates act as a regularizer.** To simplify the presentation, consider the case where step 3 is executed in Algorithm 1. Recall that the zeroth-order gradient is an unbiased estimator of the smoothed version of the actual loss. That is, $\mathbb{E}[g^0 \mathbf{z}] = \nabla\widehat{\mathcal{L}}(\boldsymbol{\theta})$, where $\widehat{\mathcal{L}}(\boldsymbol{\theta}) \triangleq \mathbb{E}_{\mathbf{z}}[\mathcal{L}(\boldsymbol{\theta} + \epsilon\mathbf{z})]$ is the Gaussian smoothed version of the original loss function $\mathcal{L}(\boldsymbol{\theta})$ (see Nesterov and Spokoiny (2017, Section 1) and Nemirovsky et al. (1983, Section 9.3)). Thus, the Addax update rule in equation 3 aims to solve the optimization problem

$$\min_{\boldsymbol{\theta}} \ (1-\alpha)\mathcal{L}(\boldsymbol{\theta}) + \alpha\widehat{\mathcal{L}}(\boldsymbol{\theta}).$$

Such a regularization, illustrated in Figure 5, can help escape sharp local minima and find higher-quality solutions. When step 5 is executed, then this regularization is only done for some of the data points, but still can be effective. This could be the reason that Addax finds higher-quality models in all our experiments.

## 4 EXPERIMENTS

**Experiment Settings.** We conduct fine-tuning experiments on five different models: the masked LM RoBERTa-large of (Liu et al., 2019) (350M), the OPT-13B, OPT-30B, OPT-66B (Zhang et al., 2022) and Llama-2-70B models (Touvron et al., 2023). We also explore the impact of hyper-parameters $\alpha$ and the batch size on Addax's performance, detailed in Appendix E.2. Further details on the datasets, prompts used, and implementation can be found in Appendix D. The code for our experiments is available at `https://github.com/optimization-for-data-driven-science/Addax`.

**Observations on OPT-13B experiments** Following (Malladi et al., 2023), we fine-tune the OPT-13B model using a single A100 GPU with Addax, MeZO, IP-SGD, and SGD. The results are presented

in Figure 1 and Table 12. In this configuration, we aim to select the largest possible batch sizes for MeZO, IP-SGD, and SGD and optimize $(K^0, K^1)$ and $L_T$ for Addax using one GPU. Averaging over nine tasks, Addax outperforms MeZO in this configuration by 14% in accuracy/F1 score and converges $15\times$ faster. To highlight the differences in convergence speed, we plot the convergence curves of Addax-WA and MeZO using the same batch size in Figure 11.

Remarkably, while Addax can successfully run on all datasets, SGD encounters out-of-memory errors on all nine tasks and IP-SGD fails in three out of nine tasks. For the six datasets where IP-SGD is able to fine-tune, Addax achieves an average accuracy/F1 score of 81.7 compared to 80.3 for IP-SGD. Finally, Addax even outperforms Adam in seven out of nine tasks while it reduces up to $89\%$ the memory. We also conducted additional experiments with Addax and IP-SGD, incorporating gradient checkpointing technique as described by Chen et al. (2016). Please refer to Appendix F.3 for details.

**Observations on OPT-30B, OPT-66B and Llama-2-70B experiments.** We also perform experiments on fine-tuning the larger-size models OPT-30B, OPT-66B and Llama-2-70B. In particular, we fine-tune OPT-30B model using one H100 GPU and report the results in Figure 2 and Table 15. To summarize the results, we report the averaged performance metrics (time, accuracy, and memory) in Table 1. Our experiment on OPT-30B shows that while Addax has comparable memory consumption to MeZO, on average, Addax outperforms MeZO by more than $16\%$ in terms of final accuracy and converges $30\times$ faster. Moreover, SGD and IP-SGD failed in three out of seven tasks due to out-of-memory, while Addax can run on all tasks. In terms of the final accuracy of the fine-tuned model, Addax outperforms MeZO, SGD, and IP-SGD in all experiments in OPT-30B.

Similar observations are made in fine-tuning OPT-66B and Llama-2-70B models on three H100 GPUs: SGD and IP-SGD fail in some or all tasks due to out-of-memory, while Addax can efficiently fine-tune on all tasks. Addax outperforms other methods in six of seven tasks for the OPT-66B model and all six tasks for the Llama-2-70B model. Moreover, Addax outperforms MeZO in terms of final accuracy while being orders of magnitude faster. The results are summarized in Table 2 and Table 3, repetively.

**Observations on RoBERTa-large experiments.** Beside large autoregressive language model, we also perform experiments on a smaller language model: RoBERTa-large with 350M parameters. The results can be found in Figure 7. As can be seen in this figure, 16/32-bit Addax outperforms zero-shot and MeZO across six different tasks and surpasses Adam in four out of six tasks. We also investigate the two important hyper-parameters, $\alpha$ and $\frac{K^1}{K^0+K^1}$ on Addax's performance. The results can be found in Figure 8 and Figure 9. As shown in the top row of the heatmaps of the two figures, it is observed that an increase in the ratio $\frac{K^1}{K^0+K^1}$ correlates with improved accuracy across tasks for both 16-bit and 32-bit Addax configurations. We did not identify a consistent trend for $\alpha$ across different tasks for both 16-bit and 32-bit Addax, suggesting that the optimal $\alpha$ requires tuning and could be task-specific.

**Zeroth-order gradient estimates improve model performance even when $K^1$ is small.** We report the detailed choice of the batch size for different algorithms in Table 12. Notably, fine-tuning OPT-13B using Addax with a smaller first-order batch size $K^1$ surpasses the performance of SGD with larger batch sizes. For example, Addax achieves an accuracy of 68.3 on the WIC task with $K^1 = 4$, while IP-SGD achieves 66.0 with a batch size of 12. This suggests that the zeroth-order gradient estimate in Addax provides stability (and regularization of the gradient) even when $K^1$ is small and can effectively reduce memory usage. Additional experimental results are given in Appendix E.

It is important to note that reported memory usage should be interpreted with caution, as it depends on the selected batch size. Reducing the batch size can lower memory usage, but it may come at the cost of accuracy and convergence speed. In our tables, when an entry is marked with $*$, it means that even with the smallest batch size in our grid (i.e. BS = 2), the algorithm still results in out-of-memory.

**Hyperparameter tuning.** For MeZO, IP-SGD, and SGD, we select the largest possible batch size from the hyperparameter search grid that maximizes GPU memory usage without causing out-of-memory. For Addax, we choose $K^0$, $K^1$, and $L_T$ values that optimize GPU usage during fine-tuning on the MultiRC dataset, as it is the task with the longest sequence length. Successful fine-tuning on MultiRC implies Addax can handle smaller tasks as well. For detailed procedures, please refer to Appendix D.6.

Table 1: Summary of OPT-30B fine-tuning results on one H100 GPU (80GB): The $\overline{\text{METRIC}}$, representing the average performance across short datasets (SST-2, RTE, WSC, WIC) and long datasets (BoolQ, MultiRC, SQuAD). $L_{max}$ is the maximum sequence length within a dataset. Detailed results are in Table 15.

| | Short Datasets ($L_{max} \leq 260$)) | | | Long Datasets ($L_{max} > 260$) | | |
| Method | Physical $\overline{\text{Memory}}$ | $\overline{\text{Wall-clock time to}}$ the best validation | $\overline{\text{Accuarcy/F1}}(\%)$ (Fine-Tuning) | Physical $\overline{\text{Memory}}$ | $\overline{\text{Wall-clock time to}}$ the best validation | $\overline{\text{Accuarcy/F1}}(\%)$ (Fine-Tuning) |
|---|---|---|---|---|---|---|
| MeZO | 66GB | 655.7min | 69.3 | 78GB | 776.0min | 68.7 |
| SGD | * | * | * | * | * | * |
| IP-SGD | 70GB | 3.0min | 75.1 | * | * | * |
| Addax | 68GB | 14.5min | 78.7 | 77GB | 28.5min | 82.0 |

Table 2: Summary of OPT-66B fine-tuning results on three H100 GPUs (240GB total): The $\overline{\text{METRIC}}$, representing the average performance across short datasets (SST-2, RTE, BoolQ, WSC, WIC, SQuAD) and long dataset (MultiRC). $L_{max}$ is the maximum sequence length within a dataset. Detailed results are in Table 16.

| | Short Datasets ($L_{max} \leq 420$)) | | | Long Dataset ($L_{max} > 420$) | | |
| Method | Physical $\overline{\text{Memory}}$ | $\overline{\text{Wall-clock time to}}$ the best validation | $\overline{\text{Accuarcy/F1}}(\%)$ (Fine-Tuning) | Physical $\overline{\text{Memory}}$ | $\overline{\text{Wall-clock time to}}$ the best validation | $\overline{\text{Accuarcy/F1}}(\%)$ (Fine-Tuning) |
|---|---|---|---|---|---|---|
| MeZO | 170GB | 511.5min | 72.4 | 197GB | 379.6min | 61.1 |
| SGD | * | * | * | * | * | * |
| IP-SGD | 170GB | 3.8min | 77.1 | * | * | * |
| Addax | 173GB | 21.9min | 80.6 | 215GB | 76.9min | 80.6 |

Table 3: Summary of Llama-2-70B fine-tuning results on three H100 GPUs (240GB total): The notation $\overline{\text{METRIC}}$ represents the average performance across short datasets (RTE, WSC, WIC) and long datasets (BoolQ, MultiRC, SQuAD). $L_{max}$ is the maximum sequence length within a dataset. Detailed results are in Table 17.

| | Short Datasets ($L_{max} \leq 260$)) | | | Long Dataset ($L_{max} > 260$) | | |
| Method | Physical $\overline{\text{Memory}}$ | $\overline{\text{Wall-clock time to}}$ the best validation | $\overline{\text{Accuarcy/F1}}(\%)$ (Fine-Tuning) | Physical $\overline{\text{Memory}}$ | $\overline{\text{Wall-clock time to}}$ the best validation | $\overline{\text{Accuarcy/F1}}(\%)$ (Fine-Tuning) |
|---|---|---|---|---|---|---|
| MeZO | 149GB | 4609min | 61.1 | 186GB | 792.3min | 73.3 |
| SGD | * | * | * | * | * | * |
| IP-SGD | 189.5GB | 6.2min | 78.2 | * | * | * |
| Addax | 190.1GB | 21.2min | 80.1 | 218GB | 37.2min | 88.9 |

## 5 CONCLUSION, BROADER IMPACT, AND LIMITATIONS

This paper introduces Addax, a memory-efficient fine-tuning method for Language Models (LMs). By leveraging both first- and zeroth-order stochastic gradient estimates, Addax demonstrates improved memory efficiency without sacrificing convergence speed or model performance, as validated by our extensive experiments across various models, tasks, and datasets. Addax has the potential to impact language model fine-tuning tasks for researchers and machine learning practitioners with limited resources. With a convergence time comparable to first-order methods and memory usage similar to zeroth-order methods, Addax provides a resource-efficient approach to optimizing higher-quality fine-tuned models. Furthermore, Addax has proven effective with large-scale models, and its memory-efficient nature can make large-scale fine-tuning more feasible by requiring fewer resources.

**Limitation:** Addax introduces an additional hyper-parameter, $\alpha$, which requires tuning for best performance. While the search grid for $\alpha$ is small and includes only five different values across all OPT experiments, this is still an additional burden for institutions with limited resources.

**Future works:** As future work, Addax may also have potential application in tasks other than fine-tuning. For example, Addax may be used for pre-training tasks or even combined with the Adam algorithm by passing the combined first- and zeroth-order gradients to Adam. From a theoretical perspective, conducting an in-depth theoretical analysis of how zeroth-order gradient estimates regularize the performance of first-order ones is of particular interest. We did not examine Addax combined with memory-efficient methods like PEFT (Hu et al., 2022; Li and Liang, 2021; Lester et al., 2021), quantization Dettmers et al. (2022), or memory-efficient attention Dao et al. (2022); Zandieh et al. (2023); Han et al. (2023), but hope to explore these in future work.

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

## A   More discussion on Addax

Algorithm 1 outlines the steps of Addax. The process starts by determining whether the dataset requires partitioning based on the sequence length threshold $L_T$. There are two possible scenarios: If memory constraints are not a concern, $L_T$ can exceed $L_{max}$, the maximum sequence length in the dataset $\mathcal{D}$. In this case, Addax retains the dataset $\mathcal{D}$ as a whole, with $\mathcal{D}^0$ and $\mathcal{D}^1$ both being equivalent to $\mathcal{D}$ (Step 3). However, when memory is limited and running IP-SGD on the entire dataset $\mathcal{D}$ is not feasible, Addax saves memory by partitioning the dataset according to $L_T$. It assigns samples with sequence lengths shorter than $L_T$ to $\mathcal{D}^1$, and the remaining samples to $\mathcal{D}^0$ (Step 5). This enables Addax to run in memory-constrained settings where IP-SGD would otherwise be infeasible.

In Step 8, the zeroth-order gradient estimator $g^0$ and random seed $s$ are obtained using the samples $\mathcal{B}^0$ with batch size $K_0$, which are drawn uniformly from the dataset $\mathcal{D}^0$. Similarly, Step 10 gets the layer $l$ first-order gradients $\boldsymbol{g}_l^1 \in \mathbb{R}^{d_l}$ from samples $\mathcal{B}^1$ through backward propagation.

A major step in algorithm 1 is the computation of zeroth-order directional derivative $g_0$, done in Step 8, which is the subroutine call in Algorithm 2. Algorithm 2 is also used in MeZO where the directional derivative is obtained through the classical zeroth-order gradient estimate SPSA; see equation equation 2. To get the zeroth-order gradient estimates, Algorithm 2 requires the evaluation of the loss function $\mathcal{L}$ through two forward passes at points $\boldsymbol{\theta} + \epsilon \boldsymbol{z}$ and $\boldsymbol{\theta} - \epsilon \boldsymbol{z}$. The naïve implementation of the SPSA algorithm costs twice the memory of inference because of the need to store the value of $\boldsymbol{z} \in \mathbb{R}^d$. Algorithm 2 removes this overhead by generating a random seed $s$ and resetting the random number generator each time model parameters are perturbed (see Step 3-7 in Algorithm 2). This approach guarantees that Algorithm 3 maintains a consistent direction for the random vector $\boldsymbol{z}$ across the two perturbations. Employing this in-place operation results in Algorithm 2 having memory consumption comparable to that of inference.

Steps 9 to 12 in Algorithm 1 are the main update steps based on the first-order gradient estimates for Addax. In Step 10, we back-propagate the first-order gradient estimates $\boldsymbol{g}_m^1$ for layer $m$ and update the model parameters in Step 11. In Step 12, Addax frees up the calculated first-order gradient estimates $\boldsymbol{g}_m^1$. This step is crucial because freeing up per-layer gradients ensures that memory requirements do not scale with the number of model parameters. More discussion on the related literature on this in-place update rule can be found in Appendix B.

Steps 14 to 17 in Algorithm 1 describes the updates of Addax based on zeroth-order gradient estimates. We use the same idea as in Malladi et al. (2023), where the seed $s$ is stored instead of the random vector $\boldsymbol{z}$. The random generator is reset before updating the components (see Step 13 in Algorithm 1), as described before. For each component $\boldsymbol{\theta}_l$ in $\boldsymbol{\theta}$ where $m$ ranges from 1 to $M$, the process begins by generating a random direction $\boldsymbol{z}_m \sim \mathcal{N}(0, \mathbf{I}_{d_m})$ in Step 15. Subsequently, each $\boldsymbol{\theta}_m$ is updated using zeroth-order gradient estimates. When iteration $t$ reaches $T$, Addax outputs the final model parameters $\boldsymbol{\theta}$.

In general, the in-place update operations have the same output as the update rule $\boldsymbol{\theta} \leftarrow \boldsymbol{\theta} - \eta(\alpha \mathbf{z} g^0 + (1 - \alpha)\mathbf{g}^1)$. This fine-grained control of dynamically allocated gradients ensures that Addax remains memory-efficient during fine-tuning.

## B   More discussion on the in-place updates

In this section, we provide a more detailed discussion on in-place updates. The technique of in-place gradient updates during backward propagation, as referenced in our approach, has been previously used in Zhao et al. (2024); Lv et al. (2023). In modern deep learning training frameworks, such as

---

**Algorithm 2** ZerothGrad (Malladi et al., 2023)

---

1: **Input:** parameters $\boldsymbol{\theta} \in \mathbb{R}^d$, loss $\mathcal{L} : \mathbb{R}^d \to \mathbb{R}$, samples $\mathcal{B}$, perturbation scale $\epsilon$.
2: Generate random seed $s$.
3: $\boldsymbol{\theta} \leftarrow$ **PertubParameters**$(\boldsymbol{\theta}, \epsilon, s)$
4: $\ell_+ \leftarrow \mathcal{L}(\boldsymbol{\theta}; \mathcal{B})$
5: $\boldsymbol{\theta} \leftarrow$ **PertubParameters**$(\boldsymbol{\theta}, -2\epsilon, s)$
6: $\ell_- \leftarrow \mathcal{L}(\boldsymbol{\theta}; \mathcal{B})$
7: $\boldsymbol{\theta} \leftarrow$ **PertubParameters**$(\boldsymbol{\theta}, \epsilon, s)$
8: $g \leftarrow (\ell_+ - \ell_-)/(2\epsilon)$
9: **Output:** $g, s$

---

**Algorithm 3** PertubParameters

---

1: **Input:** parameters $\boldsymbol{\theta}$ with $M$ Layers, perturbation scale $\epsilon$, random seed $s$.
2: Reset random number generator with seed $s$
3: **for** $m = 1, \ldots, M$ **do**
4:     $\boldsymbol{z}_m \sim \mathcal{N}(0, \mathbf{I}_{d_m})$
5:     $\boldsymbol{\theta}_m \leftarrow \boldsymbol{\theta}_m + \epsilon\boldsymbol{z}_m$                             *# Update model parameters*
6:     $\boldsymbol{\theta}_m \leftarrow$ None
7: **Output:** $\boldsymbol{\theta}$

---

PyTorch (Paszke et al., 2019)[1], they store the gradient tensor for computing optimizer states and update the model weights after all layers of gradients are computed. This approach is perfect for models with a small number of parameters; however, fine-tuning a large model, like OPT-13B with 13 billion parameters, requires significant memory because the gradient tensor has the same size as the number of model parameters. For example, as for OPT-13B model, each parameter needs 2 bytes or 4 bytes for gradient storage, totaling 26 GB or 52 GB of memory, respectively. Since Addax does not require any optimizer states, such memory overhead can be avoided by combining the computation of first-order gradient estimates and parameter updates into a single step. As described in algorithm 1 lines 9-12, we sequentially iterate over the $M^{\text{th}}$ layer to the $1^{\text{st}}$ layer, compute the gradient $\boldsymbol{g}_m^1$ (line 10), and perform in-place update to $\boldsymbol{\theta}_m$ (line 11). Right after that, we free the memory for gradient $\boldsymbol{g}_m$ (line 12). The loss computation and the update of zeroth-order gradient $g_0$ remain the same as algorithm 1. It is important to note that while the main update rule of mixing first-order and zeroth-order gradients remains unchanged, the implementation details significantly impact the memory consumption of fine-tuning tasks.

It is also worth noting that the in-place update rule has its own limitations. Firstly, it prevents the optimizer from using gradient accumulation, a technique that scales batch size by accumulating gradients over several batches and only updating the optimizer after a specified number of batches. Secondly, it prevents gradient normalization, as the norm of the gradient must be known for normalization. In our paper, we distinguish between SGD and IP-SGD: SGD uses gradient normalization during fine-tuning, while IP-SGD does not. This distinction leads to differences in final performance and convergence time. For all experiments, except those using Adam, we do not employ the gradient accumulation technique.

## C  DISCUSSION ON RELATED WORKS

**Stochastic First-order Optimizers in Deep Learning.** SGD (Robbins and Monro, 1951) has long been used in training deep neural networks due to its convergence rate that is independent of the number of model parameters. However, adaptive first-order optimizers have shown advantages over SGD in hyper-parameter tuning, final model performance, and convergence speed. The concept of adaptive first-order optimizers dates back to RPROP (Riedmiller and Braun, 1992). AdaGrad (Duchi et al., 2011) adjusts the learning rate based on the estimated geometry, assigning higher rates to less frequent features. RMSProp (Hinton et al., 2012) builds on RPROP, making it effective for small batch sizes. Adam (Kingma and Ba, 2015), inspired by AdaGrad and RMSProp, incorporates a running average of gradients and has become the preferred method for training neural networks due to its fast

---

[1]https://pytorch.org/

convergence and reduced need for hyper-parameter tuning. Numerous studies have demonstrated Adam's success (Gururangan et al., 2020; Liu et al., 2019; OpenAI, 2023), and researchers continue to investigate its efficacy with Transformer architectures (Vaswani et al., 2017).

**Zeroth-order optimization.** Zeroth-order optimization has been extensively studied in convex, strongly convex, and non-convex settings in the optimization literature (Nesterov and Spokoiny, 2017; Nemirovsky et al., 1983; Wang et al., 2023; Agarwal et al., 2009; Gao et al., 2018; Wang et al., 2020). It is known that the convergence rate of zeroth-order methods generally scales with the number of parameters $d$. This property makes zeroth-order methods less effective for training deep neural networks for which the number of parameters $d$ can be very large. Recently, MeZO (Malladi et al., 2023) demonstrated that in language fine-tuning tasks, ZO-SGD can perform comparably to first-order methods. By using in-place perturbation, MeZO applies ZO-SGD in a memory-efficient manner, keeping memory usage comparable to inference. The success of fine-tuning with zeroth-order methods may be due to the fact that LM fine-tuning can occur in a very low-dimensional subspace (Aghajanyan et al., 2020; Li et al., 2018). However, MeZO suffers from significantly slow convergence speed and slightly worse performance compared to first-order optimizers. Balasubramanian and Ghadimi (2022) estimate the Hessian to perform ZO optimization along important directions. Guo et al. (2024) focus on fine-tuning a minimal subset of LLM parameters using zeroth-order methods, incorporating sparsity and quantization to overcome memory limitations.

**Mixing update directions from different optimizers.** There are recent studies of mixing the update directions of different optimizers to enhance the performance of training/fine-tuning. MAS (Landro et al., 2020) integrates SGD and Adam by assigning constant weights to balance the contributions of gradient estimates from both optimizers. Concurrent to our work, Zhang et al. (2024b) explores integrating zeroth- and first-order gradient estimates through a "hybrid ZO-FO fine-tuning scheme for LLMs". However, their method is completely different than ours. In particular, they restrict backpropagation to the deeper layers of the model, using zeroth-order optimization for the shallower layers to update parameters. Although this technique enhances memory efficiency, it neglects the possibility of using in-place first-order updates, leading to significant memory usage. In contrast, our approach employs in-place first-order gradient estimates, ensuring that memory requirements do not scale with the number of model parameters, eliminating the need to restrict backpropagation to specific layers. As we discussed in the main body, the memory demands of fine-tuning LMs with small first-order batch sizes are comparable to those using zeroth-order batch sizes for many fine-tuning tasks. By utilizing both gradient estimates, our proposed method, Addax, not only proves to be more memory-efficient but also surpasses competing methods in terms of the final accuracy of the fine-tuned model.

## D  EXPERIMENT SETUP

The code is available at `https://github.com/optimization-for-data-driven-science/Addax`.

### D.1  DATASETS

Our setup mainly follows the experiments in (Malladi et al., 2023). We employ the same datasets utilized in Malladi et al. (2023). Unless otherwise noted, we apply the same data processing procedures and settings for both validation and training.

For the RoBERTa-large model, we utilize the following datasets: SST-2 (Socher et al., 2013), SST-5 (Socher et al., 2013), SNLI (Bowman et al., 2015), MNLI (Williams et al., 2018), RTE (Dagan et al., 2005; Bar-Haim et al., 2006; Giampiccolo et al., 2007; Bentivogli et al., 2009), and TREC (Voorhees and Tice, 2000). The test set is limited to 1,000 examples for both training and validation purposes. In our few-shot learning experiments, we set $k = 16$, where $k$ represents the number of examples per class for training and validation.

For the OPT experiments, we employ the SuperGLUE dataset (Wang et al., 2019), comprising BoolQ (Clark et al., 2019), CB (De Marneffe et al., 2019), COPA (Roemmele et al., 2011), MultiRC (Khashabi et al., 2018), RTE (Dagan et al., 2005; Bar-Haim et al., 2006; Giampiccolo et al., 2007; Bentivogli et al., 2009), WIC (Pilehvar and Camacho-Collados, 2018), and WSC (Levesque et al., 2012). Following the approach of Malladi et al. (2023), we also include SST-2 (Socher et al.,

2013) for development purposes, along with one question answering (QA) datasets: SQuAD (Rajpurkar et al., 2016). For each dataset, we randomly select $1,000$ examples for training, $500$ examples for validation, and $1,000$ examples for testing.

## D.2 DATASETS OVERVIEW

In this section, we provide a brief overview of the datasets used in our experiments. For both training and fine-tuning with transformers, peak memory usage is usually determined by the dataset's longest sequence length $L_{max}$. Samples shorter than $L_{max}$ are padded to this length. Figure 6 shows the histograms of sequence lengths for six different datasets (SST-2, RTE, WSC, WIC, MultiRC) tokenized by the OPT-13B tokenizer. The distributions of these data points are mostly right-skewed normal distributions, meaning that there are relatively a small number of samples with long sequence lengths in each dataset.

## D.3 PROMPTS

To ensure a fair comparison, we employ the same prompts as those used by Malladi et al. (2023), which were initially adapted from Gao et al. (2020), GPT-3 (Brown et al., 2020), and PromptSource (Bach et al., 2022). Table 4 presents the prompts employed in our RoBERTa-large experiments, while Table 5 details the prompts utilized for the OPT experiments.

Table 4: The prompts for each dataset used in our RoBERTa-large experiments. These prompts are identical to those used by Malladi et al. (2023). There are three different task types: sentiment classification (sentiment cls.), topic classification (topic cls.) and natural language inference (NLI). $C$ is the number of classes for each dataset. The label words can be filled in the `[MASK]` token of the prompt template. <S1> and <S2> are the first and second (if any) input sentences.

| Dataset | $C$ | Type | Prompt | Label words |
|---------|-----|------|--------|-------------|
| SST-2 | 2 | sentiment cls. | <S1> It was `[MASK]`. | {great, terrible} |
| SST-5 | 5 | sentiment cls. | <S1> It was `[MASK]`. | {great, good, okay, bad, terrible} |
| TREC | 6 | topic cls. | `[MASK]` : <S1> | {Description, Expression, Entity, Human, Location, Number} |
| MNLI | 3 | NLI | <S1> ? `[MASK]`, <S2> | {Yes, Maybe, No} |
| SNLI | 3 | NLI | <S1> ? `[MASK]`, <S2> | {Yes, Maybe, No} |
| RTE | 2 | NLI | <S1> ? `[MASK]`, <S2> | {Yes, No} |

## D.4 IMPLEMENTATION

For the RoBERTa-large experiments, we run Addax in two separate computational precision settings: one using 16-bit floating-point calculations (FP16), referred to as 16-bit Addax, and the other using 32-bit floating-point calculations (FP32), denoted as 32-bit Addax for clarity. For fine-tuning RoBERTa-large For all RoBERTa-large experiments, MeZO and Adam are loaded in the FP32 setting. Since fine-tuning RoBERTa-large models does not require significant memory, we set $L_T$ sufficiently large for running Addax, ensuring that $\mathcal{D}^0$ and $\mathcal{D}^1$ are equivalent to the total dataset $\mathcal{D}$. For all RoBERTa-large experiments, we use V100 GPUs for fine-tuning.

For the OPT and Llama experiments, Addax is used solely in the FP16 setting. We run SGD and IP-SGD in the FP16 setting and Adam in the FP32 setting. Unless otherwise noted, Addax, SGD, IP-SGD, and MeZO are trained in the FP16 setting, while Adam uses the FP32 setting. If the maximum sequence length within the dataset $L_{max}$ is less than $L_T$, Addax does not partition the dataset and $\mathcal{D}^0$ and $\mathcal{D}^1$ are equivalent to the total dataset $\mathcal{D}$. In the scenarios where IP-SGD fails, Addax further reduces the memory usage by assigning different batches of data to ZO-SGD and FO-SGD based on $L_T$. Specifically, Addax partitions the training data $X$ based on the length of data points, given a threshold $L_T$. The data is divided into $\mathcal{D}^0$ and $\mathcal{D}^1$, where $\mathcal{D}^0 = \{x \in \mathcal{D} \mid \text{length}(x) > L_T\}$ and $\mathcal{D}^1 = \{x \in \mathcal{D} \mid \text{length}(x) \le L_T\}$.

We do not employ advanced quantization techniques such as `LLM.int8()` (Dettmers et al., 2022) and QLoRA (Dettmers et al., 2023), nor do we integrate Addax with Parameter-Efficient Fine-Tuning methods (PEFT) (Hu et al., 2022; Li and Liang, 2021; Lester et al., 2021). For model inference, we

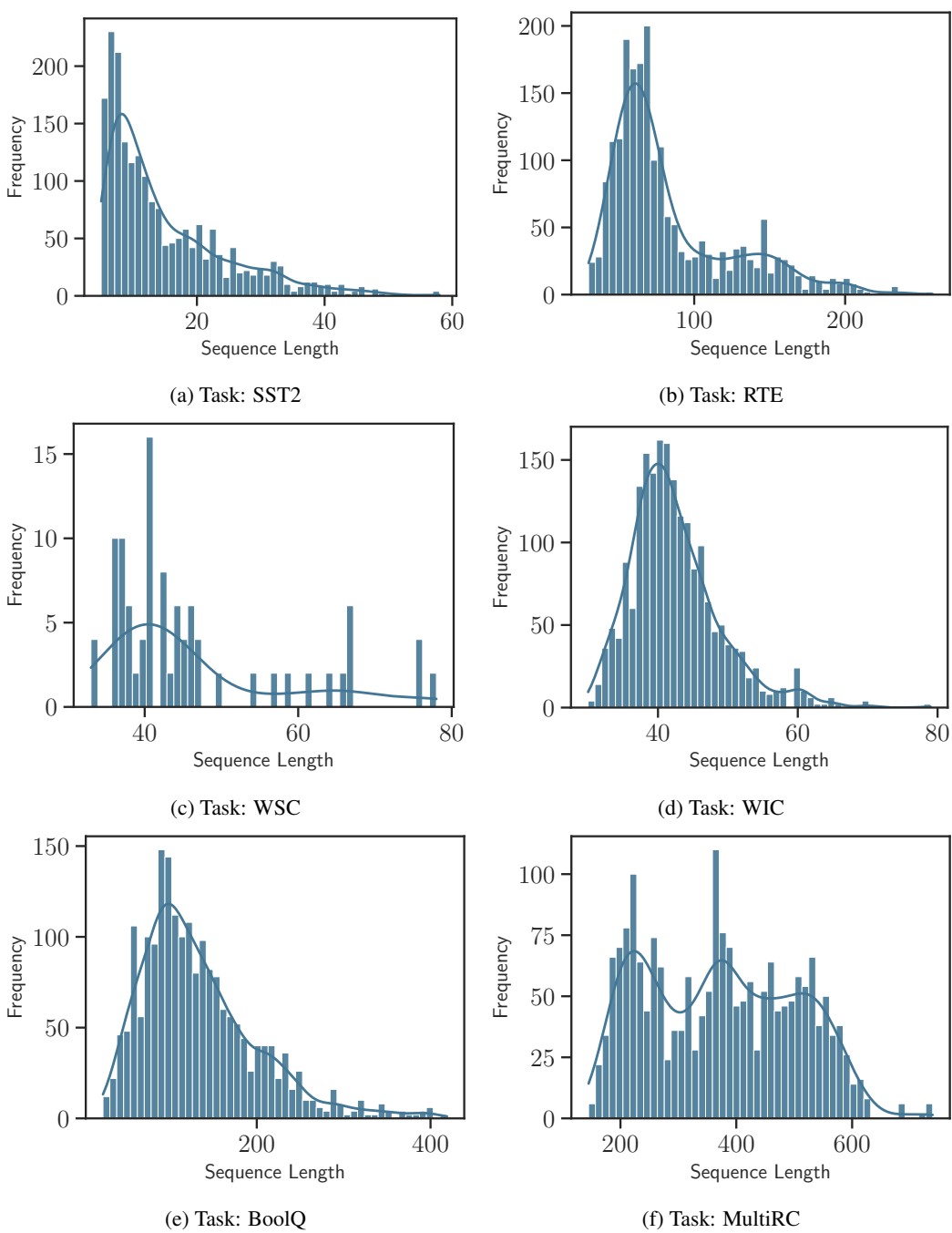

(a) Task: SST2

(b) Task: RTE

(c) Task: WSC

(d) Task: WIC

(e) Task: BoolQ

(f) Task: MultiRC

Figure 6: Histogram of sequence lengths for different datasets tokenized by the OPT-13B tokenizer. Among these datasets, MultiRC has the longest sequence length, $L_{max} = 739$.

utilize the standard PyTorch (Paszke et al., 2019) implementation of transformer. We do not use the memory-efficient approaches such as FlashAttention (Dao et al., 2022), KDEformer (Zandieh et al., 2023), and HyperAttention Han et al. (2023). Although the combination between Addax and these methods remains unexplored, we posit that their combination could significantly enhance Addax by further reducing the memory demands and augmenting performance. We leave the exploration of Addax's interplay with various memory-efficient methods to future work.

Table 5: The prompts used in the OPT experiments are identical to those used in Malladi et al. (2023). There are three types of tasks: classification (cls.), multiple-choice (mch.), and question-answering (QA). <text> is the input from the dataset and blue text are the label words. We follow the same practice as in Malladi et al. (2023): for the inference task, we incorporate different candidates into the prompt, compute the average log-likelihood for each, and select the candidate with the highest score. For question-answering (QA) tasks, answers are produced through greedy decoding.

| Dataset | Type | Prompt |
|---------|------|--------|
| SST-2 | cls. | <text> It was terrible/great |
| RTE | cls. | <premise> |
| | | Does this mean that "<hypothesis>" is true? Yes or No? |
| | | Yes/No |
| CB | cls. | Suppose <premise> Can we infer that "<hypothesis>"? Yes, No, or Maybe? |
| | | Yes/No/Maybe |
| BoolQ | cls. | <passage> <question>? |
| | | Yes/No |
| WSC | cls. | <text> |
| | | In the previous sentence, does the pronoun "<span2>" refer to <span1>? Yes or No? |
| | | Yes/No |
| WIC | cls. | Does the word "<word>" have the same meaning in these two sentences? Yes, No? |
| | | <sent1> |
| | | <sent2> |
| | | Yes/No |
| MultiRC | cls. | <paragraph> |
| | | Question: <question> |
| | | I found this answer "<answer>". Is that correct? Yes or No? |
| | | Yes/No |
| COPA | mch. | <premise> so/because <candidate> |
| SQuAD | QA | Title: <title> |
| | | Context: <context> |
| | | Question: <question> |
| | | Answer: |
| DROP | QA | Passage: <context> |
| | | Question: <question> |
| | | Answer: |

## D.5 Hyper-parameters

We present the hyper-parameters for all experiments conducted with RoBERTa-large in Table 6 and those for OPT-13B, OPT-30B, OPT-66B, Llama-2-70B in Table 7, Table 8, Table 9 and Table 10 respectively. It is important to note that for both models, the hyper-parameters grid utilized for MeZO and Adam adheres to the specifications set forth in Malladi et al. (2023).

For RoBERTa-large experiments, both Addax and MeZO employ a constant learning rate schedule, while Adam uses linear scheduling. For the training process, Addax and Adam are set for $1,000$ steps, while MeZO extends to $100,000$ steps. We check validation performance every $50$ training step and save the best validation checkpoint for testing.

For OPT experiments, Addax, SGD, IP-SGD and MeZO similarly adopt a constant learning rate schedule, with Adam maintaining its linear scheduling. Here, Adam is configured for $100$ steps, whereas Addax, SGD, IP-SGD are set for $1,000$ steps, and MeZO for $20,000$ steps. For the SGD and IP-SGD experiments in OPT-13B, we choose the batch size from our grid that maximizes the memory usage of a single A100 GPU, as larger batch sizes tend to yield better results. For OPT-30B, OPT-66B experiments and Llama-2-70B experiments, we set the batch size of SGD and IP-SGD same or less than $K^1$ of Addax. We check validation performance every $1/20$ training step and save the best validation checkpoint for testing. We leave a detailed discussion on how we selected batch size, $K^1$, $K^0$, and $L_T$ for reporting accuracy in Appendix D.6.

As explained in the main body, **Addax can use larger learning rates than MeZO,** resulting in a faster convergence. For the RoBERTa-large experiments, Addax uses the learning rate $\eta$ of $\{1e-5, 5e-5, 1e-4\}$, while MeZO uses the learning rate $\eta$ of $\{1e-7, 1e-6, 1e-5\}$. For the

OPT experiments, we fix the learning rate $\eta$ of Addax to $1e-4$, while MeZO uses a magnitude smaller learning rate $\eta$ of $\{1e-6, 1e-7\}$.

Table 6: The hyper-parameter grids used for RoBERTa-large experiments. Addax and MeZO use a constant learning rate schedule, and Adam uses linear scheduling. Addax and Adam use 1K steps and MeZO uses 100K steps. We check validation performance every 50 training steps and save the best for testing.

| Experiment | Hyper-parameters | Values |
|---|---|---|
| 16-bit/32-bit Addax | $K^0 + K^1$ | 64 |
| | $\frac{K^1}{K^0+K^1}$ | $\{0.1, 0.2, 0.3, 0.4, 0.5\}$ |
| | Learning Rate $\eta$ | $\{1e-5, 5e-5, 1e-4\}$ |
| | $\epsilon$ | $1e-3$ |
| | $\alpha$ | $\{3e-4, 1e-3, 3e-3, 4e-3,$ $5e-3, 7e-3, 1e-2, 1e-1\}$ |
| MeZO | Batch size | 64 |
| | Learning Rate $\eta$ | $\{1e-7, 1e-6, 1e-5\}$ |
| | $\epsilon$ | $1e-3$ |
| 32-bit Adam | Batch size | $\{2, 4, 8\}$ |
| | Learning Rate $\eta$ | $\{1e-5, 3e-5, 5e-5\}$ |

Table 7: The hyper-parameter grids used for OPT-13B experiments in one A100 GPU (40GB). Addax, SGD, IP-SGD and MeZO use a constant learning rate schedule, and Adam uses linear scheduling. Adam uses 200 steps. Addax, IP-SGD, SGD use 1K steps and MeZO 20K steps. We check validation performance every $1/20$ training steps and save the best for testing. Note that for IP-SGD, SGD, and MeZO, some runs may have encountered out-of-memory errors during training when fine-tuning with one A100 GPU.

| Experiment | Hyper-parameters | Values |
|---|---|---|
| Addax | $(K^1, K^0)$ | $(4, 6)$ |
| | Learning Rate $\eta$ | $1e-4$ |
| | $\epsilon$ | $1e-3$ |
| | $\alpha$ | $\{1e-4, 3e-4, 5e-4, 7e-4, 9e-4\}$ |
| | $L_T$ | $\{150, 155, 160, 165, 170\}$ |
| MeZO | Batch size | $\{2, 4, 6, 8, 10, 12, 14, 16, 20, 24, 28, 32\}$ |
| | Learning Rate $\eta$ | $\{1e-6, 1e-7\}$ |
| | $\epsilon$ | $1e-3$ |
| SGD | Batch size | $\{2, 4, 6, 8, 10, 12, 14, 16, 20, 24, 28, 32\}$ |
| | Learning Rate $\eta$ | $\{5e-3, 1e-2, 5e-2\}$ |
| IP-SGD | Batch size | $\{2, 4, 6, 8, 10, 12, 14, 16, 20, 24, 28, 32\}$ |
| | Learning Rate $\eta$ | $\{1e-4, 1.25e-4, 7.5e-4\}$ |
| Adam | Batch size | 8 |
| | Learning Rate $\eta$ | $\{1e-5, 5e-5, 8e-5\}$ |

## D.6 DETAILED METHODS IN SELECTING BS, $K^0$, $K^1$, AND $L_T$ FOR REPORTING

### D.6.1 OPT-13B EXPERIMENTS

In our OPT-13B experiments, we run Addax, MeZO, IP-SGD, and SGD using a single A100 (40GB) GPU. For MeZO, IP-SGD, and SGD, we select a batch size from the grid $\{2, 4, 6, 8, 10, 12, 14, 16, 20, 24, 28, 32\}$, choosing the largest possible batch size that maximizes GPU memory usage without encountering out-of-memory errors. A method is considered to have failed to fine-tune the dataset if it cannot run even with the smallest batch size from the grid. For example, SGD fails to fine-tune all datasets with the smallest batch size, and IP-SGD fails to fine-tune the BoolQ, MultiRC, and SQuAD datasets on a single A100 (40GB) GPU.

Table 8: The hyper-parameter grids used for OPT-30B experiments in one H100 GPU (80GB). Addax, SGD, IP-SGD and MeZO use a constant learning rate schedule Addax, IP-SGD, SGD use 1K steps and MeZO 20K steps. We check validation performance every $1/20$ training steps and save the best for testing.

| Experiment | Hyper-parameters | Values |
|---|---|---|
| Addax | $(K^1, K^0)$ | $\{(2,6), (4,6)\}$ |
| | Learning Rate $\eta$ | $1e-4$ |
| | $\epsilon$ | $1e-3$ |
| | $\alpha$ | $\{1e-4, 3e-4, 5e-4, 7e-4, 9e-4\}$ |
| | $L_T$ | $\{320, 180\}$ |
| MeZO | Batch size | $\{2, 4, 6, 8, 10, 12, 14, 16\}$ |
| | Learning Rate $\eta$ | $\{1e-6, 1e-7\}$ |
| | $\epsilon$ | $1e-3$ |
| SGD | Batch size | $\{2, 4\}$ |
| | Learning Rate $\eta$ | $\{5e-3, 1e-2, 5e-2\}$ |
| IP-SGD | Batch size | $\{2, 4\}$ |
| | Learning Rate $\eta$ | $\{1e-4, 1.25e-4, 7.5e-4\}$ |

Table 9: The hyper-parameter grids used for OPT-66B experiments in three H100 GPU (240GB total). Addax, SGD, IP-SGD and MeZO use a constant learning rate schedule Addax, IP-SGD, SGD use 1K steps and MeZO 20K steps. We check validation performance every $1/20$ training step and save the best for testing.

| Experiment | Hyper-parameters | Values |
|---|---|---|
| Addax | $(K^1, K^0)$ | $(4,6)$ |
| | Learning Rate $\eta$ | $1e-4$ |
| | $\epsilon$ | $1e-3$ |
| | $\alpha$ | $\{1e-4, 3e-4, 5e-4, 7e-4, 9e-4\}$ |
| | $L_T$ | $260$ |
| MeZO | Batch size | $\{2, 4, 6, 8, 10, 12, 14, 16\}$ |
| | Learning Rate $\eta$ | $\{1e-6, 1e-7\}$ |
| | $\epsilon$ | $1e-3$ |
| SGD | Batch size | $\{2, 4\}$ |
| | Learning Rate $\eta$ | $\{5e-3, 1e-2, 5e-2\}$ |
| IP-SGD | Batch size | $\{2, 4\}$ |
| | Learning Rate $\eta$ | $\{1e-4, 1.25e-4, 7.5e-4\}$ |

To further reduce the memory requirements of IP-SGD, we first fine-tune Addax on the longest dataset, MultiRC (Figure 6), as successfully running Addax on the longest dataset ensures it can also run on shorter ones. We found that with $(K^1, K^0) = (4,6)$ and $L_T = 170$, Addax achieves optimal performance on the MultiRC dataset while staying within the 40GB memory constraints. Thus, we use the same $(K^1, K^0) = (4,6)$ configuration for Addax when fine-tuning the remaining datasets, searching for optimal combinations of $\alpha$ and $L_T \leq 170$.

### D.6.2 OPT-30B, OPT-66B, LLAMA-2-70B EXPERIMENTS

For the OPT-30B experiments, we follow the same methodology as described in the OPT-13B experiments. For MeZO, IP-SGD, and SGD, we choose the largest possible batch size from the grid that maximizes GPU memory usage without encountering out-of-memory errors, using a single H100 GPU (80GB). For Addax in the OPT-30B experiments, we fine-tune with two different settings: $(K^1, K^0) = (4,6)$ when $L_T = 180$ and $(K^1, K^0) = (2,6)$ when $L_T = 320$. In both settings, Addax does not encounter out-of-memory issues when fine-tuning the MultiRC dataset in one GPU, indicating that it can also fine-tune the remaining datasets.

Table 10: The hyper-parameter grids used for Llama-2-70B experiments in three H100 GPU (240GB total). Addax, SGD, IP-SGD and MeZO use a constant learning rate schedule Addax, IP-SGD, SGD use 1K steps and MeZO 20K steps. We check validation performance every 1/20 training step and save the best for testing.

| Experiment | Hyper-parameters | Values |
|---|---|---|
| Addax | $(K^1, K^0)$ | $(4, 6)$ |
| | Learning Rate $\eta$ | $1e-4$ |
| | $\epsilon$ | $1e-3$ |
| | $\alpha$ | $\{1e-4, 3e-4, 5e-4, 7e-4, 9e-4\}$ |
| | $L_T$ | $240$ |
| MeZO | Batch size | $\{2, 4, 6, 8, 10, 12, 14, 16\}$ |
| | Learning Rate $\eta$ | $\{1e-6, 1e-7\}$ |
| | $\epsilon$ | $1e-3$ |
| SGD | Batch size | $\{2, 4\}$ |
| | Learning Rate $\eta$ | $\{5e-3, 1e-2, 5e-2\}$ |
| IP-SGD | Batch size | $\{2, 4\}$ |
| | Learning Rate $\eta$ | $\{1e-4, 1.25e-4, 7.5e-4\}$ |

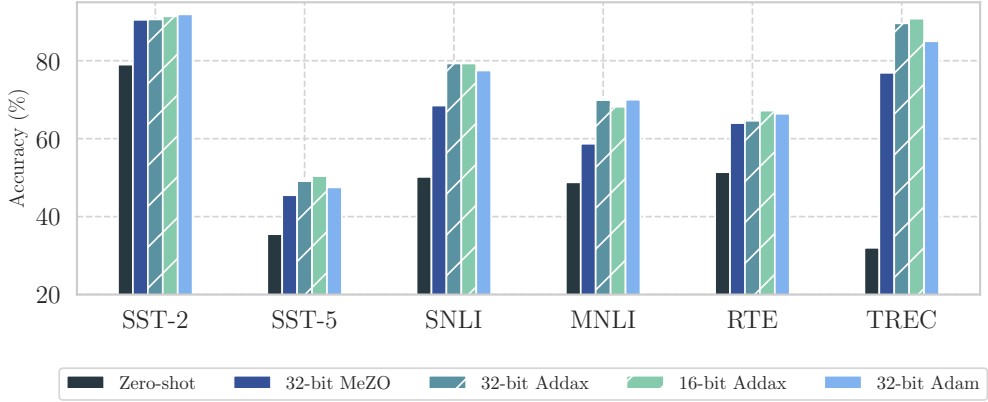

Figure 7: Experiments on RoBERTa-large: 16/32-bit Addax outperform zero-shot and MeZO across all tasks and outperform Adam in four out of six tasks. Detailed numbers can be found in Table 11.

In the OPT-66B and Llama-2-70B experiments, we fine-tune MeZO, IP-SGD, and SGD by selecting the largest possible batch size from the grid that maximizes GPU memory usage without encountering out-of-memory errors, using three H100 GPUs (a total of 240GB). For simplicity, Addax uses the same $(K^1, K^0) = (4, 6)$ configuration from the OPT-13B experiments, with $L_T = 260$ for OPT-66B and $L_T = 240$ for Llama-2-70B, allowing Addax to successfully run without out-of-memory errors.

## D.7 MEMORY PROFILING

In our memory profiling, we conform to the methodologies previously established in Malladi et al. (2023). Our implementation utilizes the default configuration provided by the `transformers` (Wolf et al., 2020) package. We do not turn on any advanced memory optimization technique such as gradient checkpointing. For multi-GPU backpropagation, we utilize the Fully Sharded Data Parallel (FSDP) (Fairscale, 2021) by PyTorch (Paszke et al., 2019). We use Nvidia's `nvidia-smi` command to monitor the GPU memory usage. We report the maximum GPU memory consumption observed throughout all experiments.

Table 11: Experiments on RoBERTa-large (350M parameters). 16-bit Addax and 32-bit Addax outperform zero-shot and MeZO across the board on 6 tasks, while surpassing Adam in four out of six tasks. All experiments use prompts (Appendix D.3). For the accuracy of 32-bit MeZO and 32-bit Adam, we report the results from Malladi et al. (2023).

| Task | SST-2 | SST-5 | SNLI | MNLI | RTE | TREC |
|---|---|---|---|---|---|---|
| Type | —sentiment— | | natural language inference | | | –topic– |
| Zero-shot | 79.0 | 35.5 | 50.2 | 48.8 | 51.4 | 32.0 |
| Samples per classes: $k\ =\ 16$ | | | | | | |
| 32-bit MeZO | 90.5 | 45.5 | 68.5 | 58.7 | 64.0 | 76.9 |
| 32-bit Addax | 90.6 | 49.1 | 79.3 | **69.9** | 64.6 | 89.6 |
| 16-bit Addax | **91.4** | **50.4** | **79.3** | 68.2 | **67.2** | **90.8** |
| 32-bit Adam | 91.9 | 47.5 | 77.5 | 70.0 | 66.4 | 85.0 |

# E   ROBERTA-LARGE EXPERIMENTS

## E.1   ROBERTA-LARGE EXPERIMENTS MAIN RESULTS

Table 11 reports the detailed numbers of the accuracy on the RoBERTa-large model across different fine-tuning methods that are shown in Figure 7. For the accuracy of MeZO and Adam, we directly report the results from Malladi et al. (2023).

## E.2   INVESTIGATION ON THE HYPER-PARAMETERS OF ADDAX

In this section, we explore the effect of different hyper-parameters, specifically reporting on the accuracy of both 32-bit and 16-bit Addax across various tasks utilizing the RoBERTa-large model using different combinations of hyper-parameters. We include the combinations of $\alpha$ and $\frac{K^1}{K^0+K^1}$ for 32-bit and 16-bit Addax in Figure 8 and Figure 9. Generally, it is observed that an increase in the ratio $\frac{K^1}{K^0+K^1}$ correlates with improved accuracy across tasks for both 16-bit and 32-bit Addax configurations, as evidenced by the top row of the heatmaps for each task in Figure 8 and Figure 9. We did not identify a consistent trend for $\alpha$ across different tasks for both 16-bit and 32-bit Addax, suggesting that the optimal $\alpha$ could be task-specific.

# F   OPT AND LLAMA EXPERIMENTS

## F.1   OPT-13B EXPERIMENTS MAIN RESULTS

Table 12 reports the detailed numbers of the accuracy on the OPT-13B model across different fine-tuning methods shown in Figure 1. Details on batch size, time to the best validation checkpoint and memory for different algorithms are also available in Table 12. We also report GPU memory consumption across tasks and different fine-tuning methods for the OPT-13B model in Figure 1, with the exact number reported in Table 12. See Appendix D.7 for memory profiling details.

## F.2   OPT-30B, OPT-66B AND LLAMA-2-70B EXPERIMENTS MAIN RESULTS

Tables 15, 16, 17 present the complete results of fine-tuning OPT-30B and OPT-66B with Addax-P, SGD, IP-SGD, and MeZO, including metrics such as accuracy, memory usage, and batch size/$(K^1, K^0)$. Figures 2 and 10 show the corresponding data.

## F.3   COMPARING IP-SGD AND ADDAX WITH GRADIENT CHECKPOINTING

In these experiments, we enabled the Gradient Checkpointing technique (Chen et al., 2016), which reduces memory requirements for storing activations during backward computation. Gradient checkpointing benefits IP-SGD and Addax, as Addax operates on shorter sequences. In particular, we observed that when we incorporate gradient checkpointing, Addax demonstrates lower memory

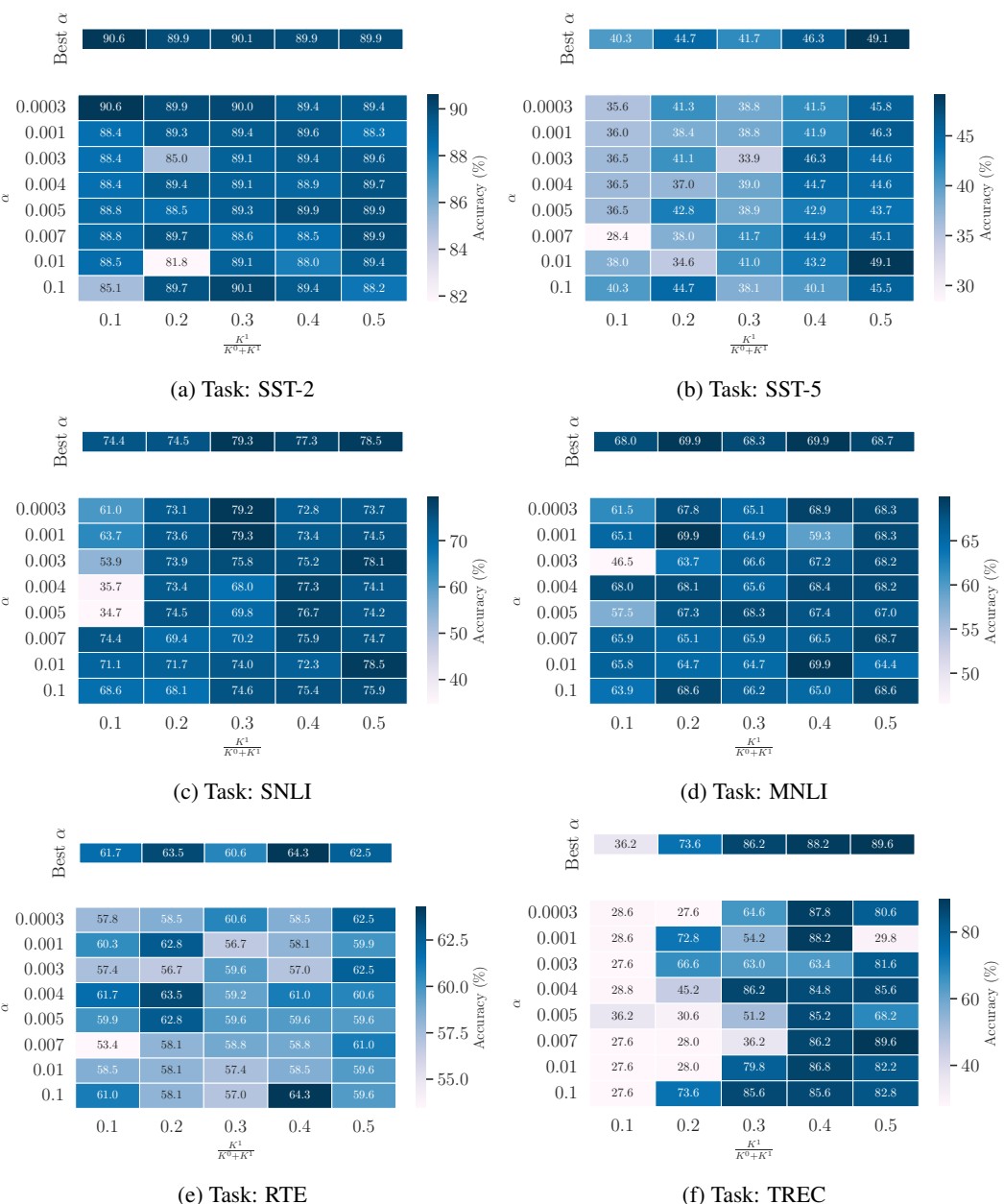

Figure 8: The accuracy (%) of the 32-bit Addax across different tasks on the RoBERTa-large model, with variable combinations of $\alpha$ and $\frac{K^1}{K^1+K^0}$.

requirements than IP-SGD. To illustrate this further, we conducted two sets of experiments. To illustrate this, we conducted two sets of experiments.

In the first experiment, we repeated the fine-tuning process with OPT-13B on the CB dataset, enabling gradient checkpointing for IP-SGD and Addax. We used the same batch size and $(K^1, K^0)$ settings as in previous experiments. The results of these experiments are presented in Table 13. Notice that enabling gradient checkpointing reduces memory usage but increases convergence time without affecting the final model quality. With gradient checkpointing enabled, the memory usage of IP-SGD decreased from 37.3 GB to 33.3 GB, while the memory usage of Addax decreased from 39.2 GB to 29.2 GB. This demonstrates that Addax indeed benefits more from gradient checkpointing regarding memory reduction. This advantage arises because Addax stores intermediate activations only for shorter sequences, which require less memory. On the other hand, IP-SGD requires storage of intermediate activations for longer sequences, which require more memory.

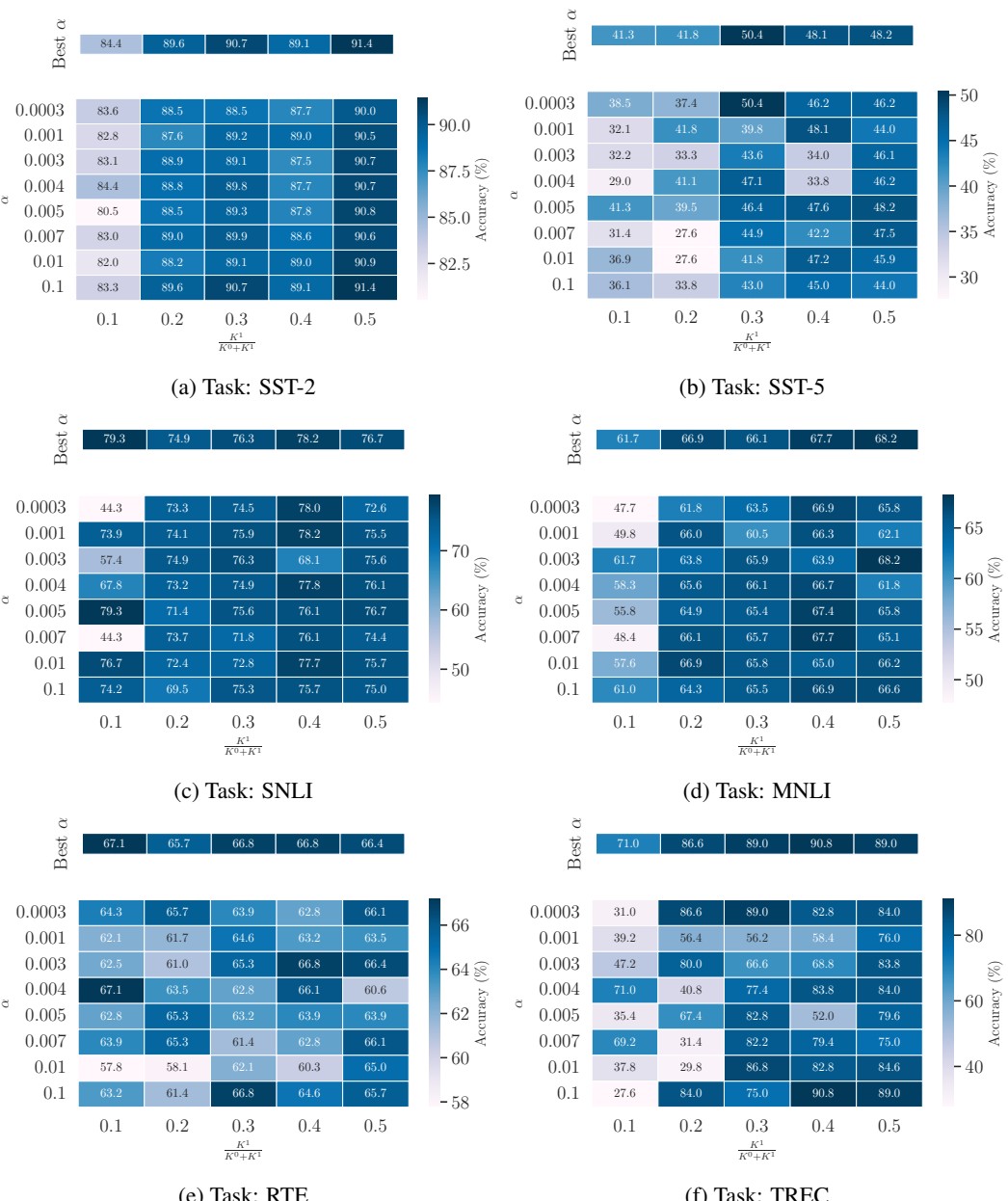

Figure 9: The accuracy (%) of the 16-bit Addax across different tasks on the RoBERTa-large model, with variable combinations of $\alpha$ and $\frac{K^1}{K^1+K^0}$.

In the second experiment, with gradient checkpointing enabled for Addax and IP-SGD, we selected the largest possible batch sizes to maximize GPU memory utilization on a single A100 (40GB) GPU. The results of these experiments are presented in Table 14. In this experiment, Addax achieves an accuracy of 91.1, while IP-SGD achieves only 87.5. Due to its data partitioning strategy based on the varying memory requirements of data points, Addax can accommodate a larger first-order batch size than IP-SGD, leading to significantly better final model quality. This result further illustrates that enabling gradient checkpointing benefits Addax as well.

In conclusion, the use of gradient checkpointing would benefit Addax in terms of memory requirements. Gradient checkpointing trades convergence time for reduced memory usage without altering the optimization process or the final model quality. This is true for both Addax and IP-SGD.

Table 12: Experiments on OPT-13B (with 1,000 examples) using a single A100 (40GB) GPU except for Adam, which is run on five H100 GPUs (350GB total). Addax consistently outperforms zero-shot, MeZO, and IP-SGD across nine tasks, and surpasses Adam in seven of the nine tasks. An asterisk (*) indicates runs that encountered out-of-memory errors during training, even with the smallest possible batch size.

| Metrics | Task Task type | SST-2 | RTE | CB | BoolQ | WSC | WIC | MultiRC | ReCoRD multiple choice | SQuAD generation |
|---|---|---|---|---|---|---|---|---|---|---|
| | | | | | classification | | | | | |
| Accuracy/F1 (%) | Zero-shot | 58.8 | 59.6 | 46.4 | 59.0 | 38.5 | 55.0 | 46.9 | 80.0 | 46.2 |
| | MeZO | 91.9 | 65.3 | 69.6 | 66.5 | 61.5 | 59.7 | 59.4 | 86.0 | 82.6 |
| | SGD | * | * | * | * | * | * | * | * | * |
| | IP-SGD | 94.5 | 82.3 | 85.7 | * | 63.5 | 66.0 | * | 90.0 | * |
| | Adam | 92.1 | 79.1 | 71.4 | 77.0 | 63.5 | **69.6** | **76.2** | 81.0 | 84.5 |
| | Addax | **94.5** | **84.8** | **89.3** | **81.0** | **63.5** | 68.3 | 71.2 | **90.0** | **88.4** |
| Memory (GB) | MeZO | 29.7 | 39.0 | 38.7 | 39.6 | 31.6 | 31.4 | 36.9 | 27.6 | 36.8 |
| | SGD | * | * | * | * | * | * | * | * | * |
| | IP-SGD | 38.3 | 35.0 | 37.7 | * | 38.6 | 38.4 | * | 30.6 | * |
| | Adam | 248.4 | 252.3 | 275.2 | 315.0 | 251.7 | 250.1 | 349.4 | 247.7 | 259.8 |
| | Addax | 28.7 | 35.6 | 39.2 | 38.0 | 29.4 | 29.3 | 39.2 | 27.7 | 33.3 |
| Batch Size | MeZO | 32 | 16 | 14 | 8 | 32 | 32 | 6 | 32 | 10 |
| | SGD | * | * | * | * | * | * | * | * | * |
| | IP-SGD | 16 | 2 | 2 | * | 12 | 12 | * | 32 | * |
| | 32-bit Adam | | | | | 8 | | | | |
| $(K^1, K^0)$ | Addax | | | | | (4, 6) | | | | |
| Time (Min) | MeZO | 222.5 | 289.2 | 182.8 | 255.4 | 40.3 | 103.9 | 363.8 | 31.7 | 245.5 |
| | SGD | * | * | * | * | * | * | * | * | * |
| | IP-SGD | 2.8 | 4.2 | 2.2 | * | 3.4 | 7.6 | * | 0.3 | * |
| | Addax | 10.2 | 23.2 | 13.5 | 35.5 | 2.1 | 17.4 | 5.3 | 0.9 | 10.8 |

Table 13: Experiments on OPT-13B fine-tuning on dataset CB with gradient checkpointing enabling or not. Using gradient checkpointing would benefit Addax more than IP-SGD in terms of memory footprint.

| Metrics | IP-SGD | IP-SGD (Gradient Checkpointing) | Addax | Addax (Gradient Checkpointing) |
|---|---|---|---|---|
| Accuracy (%) | 85.7 | 85.7 | 89.3 | 89.3 |
| Batch Size/$(K^1, K^0)$ | 2 | 2 | (4,6) | (4,6) |
| Memory (GB) | 37.7 | 33.3 | 39.2 | 29.2 |
| Time (min) | 2.2 | 3.63 | 13.5 | 15.9 |

Table 14: Experiments on OPT-13B fine-tuning on dataset CB with gradient checkpointing enabling using a single A100 (40GB) GPU. The batch size/$(K^1, K^0)$ is chosen as the largest within the 40 GB memory constraint.

| Metrics | IP-SGD (Gradient Checkpointing) | Addax (Gradient Checkpointing) |
|---|---|---|
| Accuracy (%) | 87.5 | 91.1 |
| Batch Size/$(K^1, K^0)$ | 10 | (24,12) |
| Memory (GB) | 35.5 | 38.8 |
| Time (min) | 6.1 | 24.6 |

## F.4 CONVERGENCE SPEED OF DIFFERENT TUNING METHODS ON THE OPT-13B MODEL

In this section, we demonstrate that 16-bit Addax reaches a convergence speed comparable to 16-bit SGD, despite SGD using $4\times$ more first-order samples for backward propagation. Meanwhile, Addax memory consumption is comparable to MeZO. The comparison of convergence speeds across the three methods is illustrated in Figure 11. For MeZO and SGD, the batch size is set to 16, while for Addax, we configure $(K^1, K^0)$ as $(4, 12)$. The learning rate for Addax is set at $\eta = 1e-4$. For SGD, the learning rates are $\eta = \{5e-3, 1e-2, 5e-2\}$. For MeZO, we utilize learning rates of $\eta = \{1e-6, 1e-7\}$. We select the hyper-parameters that yield the best validation accuracy across three methods. We utilize a single H100 GPU (80GB total) for running both Addax and MeZO, whereas SGD requires two H100 GPUs (160GB total). MeZO requires significantly more steps (20K steps) to converge compared to Addax and SGD (1K steps). Addax with smaller first-order batch

Table 15: Experiments on OPT-30B (with 1000 examples) on a one H100 (80GB) GPU. * means the run encounters out-of-memory errors during training.

| Metrics | Task
Task type | SST-2 | RTE | BoolQ | WSC | WIC | MultiRC | SQuAD |
|---|---|---|---|---|---|---|---|---|
| | | | | classification | | | | generation |
| Accuracy/F1 (%) | Zero-shot | 56.7 | 52.0 | 39.1 | 38.5 | 50.2 | 44.2 | 46.5 |
| | SGD | * | * | * | * | * | * | * |
| | MeZO | 90.6 | 66.4 | 66.9 | 63.5 | 56.3 | 59.3 | 79.9 |
| | IP-SGD $BS = 2$ | 89.6 | 77.6 | * | 63.5 | 68.0 | * | * |
| | IP-SGD $BS = 4$ | 91.2 | * | * | 63.5 | 66.5 | * | * |
| | Addax ($L_T = 320$) | 93.9 | 83.4 | 80.8 | 63.5 | 66.8 | **75.8** | 85.9 |
| | Addax ($L_T = 180$) | **95.1** | **85.9** | **82.3** | 63.5 | **70.2** | 67.8 | **88.0** |
| Batch Size | MeZO | ——16—— | | | 10 | 16 | 16 | 6 | 12 |
| | IP-SGD $BS = 2$ | | | | 2 | | | | |
| | IP-SGD $BS = 4$ | | | | 4 | | | | |
| $(K^1, K^0)$ | Addax ($L_T = 320$) | | | | (2, 6) | | | | |
| | Addax ($L_T = 180$) | | | | (4, 6) | | | | |
| Memory (GB) | MeZO | 62.0 | 75.0 | 79.8 | 64.6 | 63.8 | 76.0 | 78.3 |
| | IP-SGD $BS = 2$ | 62.5 | 80.0 | * | 64.4 | 62.9 | * | * |
| | IP-SGD $BS = 4$ | 65.2 | * | * | 66.3 | 66.5 | * | * |
| | Addax ($L_T = 320$) | 64.9 | 75.5 | 78.4 | 62.6 | 62.6 | 81.0 | 69.1 |
| | Addax ($L_T = 180$) | 64.4 | 79.5 | 79.5 | 65.8 | 66.0 | 80.8 | 71.3 |
| Time (min) | MeZO | 719.3 | 980.0 | 499.0 | 116.9 | 762.6 | 962.8 | 866.2 |
| | IP-SGD $BS = 2$ | 1.9 | 1.1 | * | 1.0 | 7.9 | * | * |
| | IP-SGD $BS = 4$ | 1.9 | * | * | 1.1 | 9.1 | * | * |
| | Addax ($L_T = 320$) | 4.0 | 9.8 | 32.3 | 1.4 | 19.7 | 11.4 | 3.7 |
| | Addax ($L_T = 180$) | 9.7 | 23.1 | 25.5 | 1.5 | 23.5 | 48.6 | 11.3 |

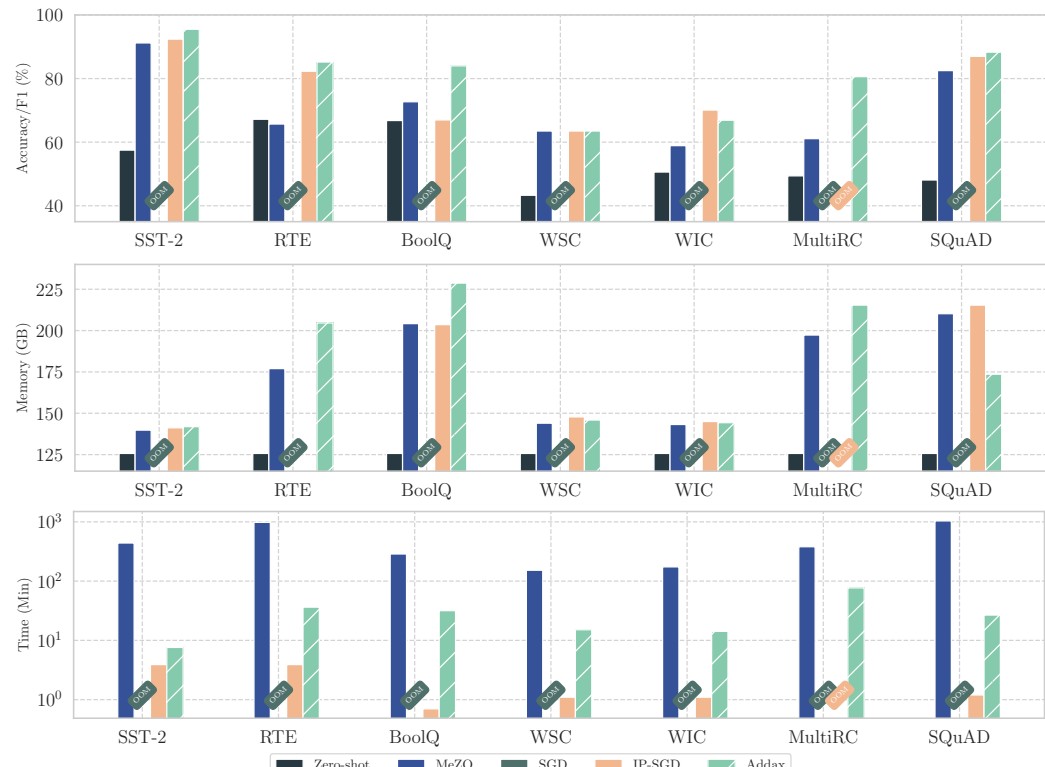

Figure 10: Accuracy/F-1 score, memory, and convergence time resulted from fine-tuning OPT-66B model with zero-shot, MeZO, SGD, IP-SGD, and Addax on 3 H100 GPU (240GB total). The label "OOM" means the run encounters out-of-memory error during fine-tuning. The exact numbers related to this figure can be found in Table 16 in Appendix F.2.

Table 16: Experiments on OPT-66B (with 1000 examples) on three GPUs (240GB total). * means the run encounters out-of-memory errors during training.

| Metrics | Task
Task type | SST-2 | RTE | BoolQ | WSC | WIC | MultiRC | SQuAD |
|---|---|---|---|---|---|---|---|---|
| | | | | classification | | | | generation |
| Accuracy/F1 (%) | Zero-shot | 57.5 | 67.2 | 66.8 | 43.3 | 50.6 | 49.4 | 48.1 |
| | SGD | * | * | * | * | * | * | * |
| | MeZO | 91.2 | 65.7 | 72.7 | 63.5 | 58.9 | 61.1 | 82.5 |
| | IP-SGD (BS=2) | 89.1 | 82.3 | 67.0 | 63.5 | 65.8 | * | 87.0 |
| | IP-SGD (BS=4) | 92.4 | 78.7 | * | 63.5 | **70.1** | * | * |
| | Addax ($L_T = 260$) | **95.5** | **85.2** | **84.0** | **63.5** | 66.9 | **80.6** | **88.3** |
| Batch Size | MeZO | | | 16 | | | 8 | 16 |
| | IP-SGD (BS=2) | | | | 2 | | | |
| | IP-SGD (BS=4) | | | | 4 | | | |
| $(K^1, K^0)$ | Addax ($L_T = 260$) | | | | $(4,6)$ | | | |
| Memory (GB) | MeZO | 139.8 | 177.0 | 204.2 | 144.0 | 143.2 | 197.3 | 210.2 |
| | IP-SGD | 136.5 | 166.2 | 203.6 | 145.4 | 139.4 | * | 215.4 |
| | IP-SGD | 141.2 | 213.3 | * | 147.8 | 145.0 | * | * |
| | Addax ($L_T = 260$) | 141.9 | 204.6 | 228.7 | 145.9 | 144.3 | 215.4 | 173.6 |
| Time (min) | MeZO | 439.1 | 980.5 | 286.6 | 152.4 | 173.7 | 379.6 | 1036.2 |
| | IP-SGD | 0.4 | 2.8 | 0.7 | 4.9 | 3.0 | * | 1.2 |
| | IP-SGD | 3.9 | 1.7 | * | 1.1 | 9.1 | * | * |
| | Addax ($L_T = 260$) | 7.6 | 36.3 | 31.7 | 15.1 | 14.2 | 76.9 | 26.7 |

Table 17: Experiments on Llama-2-70B (with 1000 examples) on three H100 (80GB) GPUs. * means the run encounters out-of-memory errors during training.

| Metrics | Task
Task type | RTE | BoolQ | WSC | WIC | MultiRC | SQuAD |
|---|---|---|---|---|---|---|---|
| | | | classification | | | | generation |
| Accuracy/F1 (%) | Zero-shot | 60.6 | 75.9 | 55.8 | 49.8 | 45.8 | 70.5 |
| | SGD | * | * | * | * | * | * |
| | MeZO | 52.7 | 63.1 | 75.0 | 55.6 | 64.4 | 92.3 |
| | IP-SGD $BS = 2$ | 85.2 | * | 75.0 | 73.4 | * | * |
| | IP-SGD $BS = 4$ | * | * | 75.0 | 74.3 | * | * |
| | Addax ($L_T = 240$) | **89.9** | **87.9** | **76.0** | **74.5** | **85.3** | **93.4** |
| Batch Size | MeZO | | 16 | | | 6 | 16 |
| | IP-SGD $BS = 2$ | | | 2 | | | |
| | IP-SGD $BS = 4$ | | | 4 | | | |
| | Addax ($L_T = 240$) | | | $(4,6)$ | | | |
| Memory (GB) | MeZO | 159.4 | 195.9 | 143.6 | 143.6 | 169.3 | 192.9 |
| | IP-SGD $BS = 2$ | 235.2 | * | 150.8 | 151.6 | * | * |
| | IP-SGD $BS = 4$ | * | * | 164.0 | 182.5 | * | * |
| | Addax ($L_T = 240$) | 239.5 | 231.7 | 162.9 | 167.9 | 236.1 | 187.3 |
| Time (min) | MeZO | 1288.7 | 565.0 | 6133.7 | 6405.5 | 879.9 | 932.0 |
| | IP-SGD $BS = 2$ | 2.6 | * | 5.0 | 9.5 | * | * |
| | IP-SGD $BS = 4$ | * | * | 1.3 | 11.0 | * | * |
| | Addax ($L_T = 240$) | 31.7 | 28.0 | 5.0 | 27.0 | 30.0 | 53.7 |

size $K^1 = 4$ achieves a convergence speed similar to SGD with a batch size of 16, despite requiring significantly less memory.

# G THEORETICAL RESULTS AND PROOFS

## G.1 LIST OF ASSUMPTIONS

**Assumption G.1.** $\ell(\boldsymbol{\theta}; x)$ is $L$-Lipschitz smooth, i.e.,

$$\|\nabla\ell(\boldsymbol{\theta}; x) - \nabla\ell(\boldsymbol{\theta}'; x)\| \leq L\|\boldsymbol{\theta} - \boldsymbol{\theta}'\|, \forall\boldsymbol{\theta}, \boldsymbol{\theta}' \in \mathbb{R}^d, x \in \mathcal{D}.$$

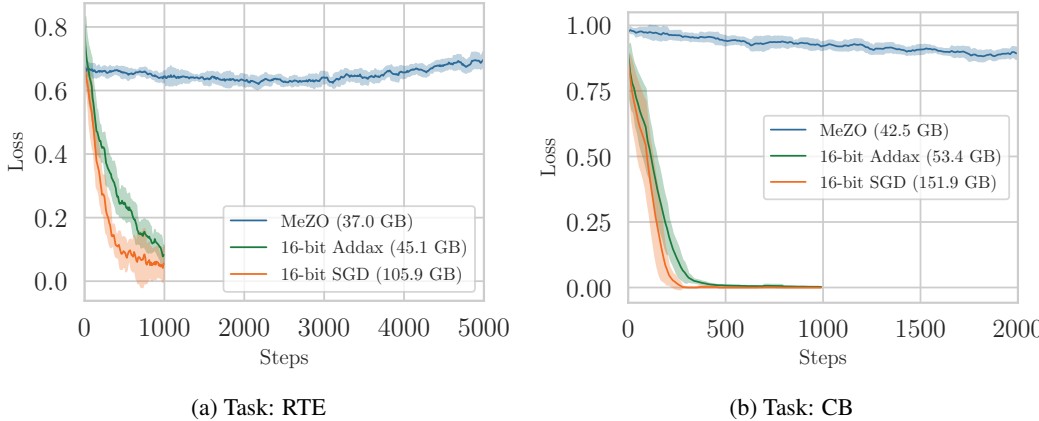

(a) Task: RTE

(b) Task: CB

Figure 11: Convergence speed of three fine-tuning methods ( Addax-WA, MeZO, and SGD) on two fine-tuning datasets with the OPT-13B model. We set the batch size to 16 for MeZO and SGD and fix $(K^1, K^0) = (4, 12)$ for Addax-WA. We utilize a single H100 GPU (80GB total) for running both Addax-WA and MeZO, whereas SGD requires two H100 GPUs (160 GB total) to run with $BS = 16$. MeZO requires significantly more steps to converge compared to Addax-WA and SGD. Addax-WA with $4\times$ less first-order samples achieves a convergence speed similar to SGD, despite requiring significantly less memory.

**Assumption G.2.** The stochastic gradient is unbiased and has bounded variance, i.e.,

$$\mathbb{E}_x[\nabla\ell(\boldsymbol{\theta};x)] = \nabla\mathcal{L}(\boldsymbol{\theta}), \ \mathbb{E}_x[\|\nabla\ell(\boldsymbol{\theta};x) - \nabla\mathcal{L}(\boldsymbol{\theta})\|^2] \leq \sigma^2, \forall\boldsymbol{\theta} \in \mathbb{R}^d.$$

**Assumption G.3** (Low effective rank Hessian)**.** There exists a matrix $0 \preceq \boldsymbol{H} \preceq L \cdot \boldsymbol{I}_d$ such that $\nabla^2\mathcal{L}(\mathbf{x}) \preceq \boldsymbol{H}$, and the effective rank of $\boldsymbol{H}$ is at most $r$, i.e.,

$$\text{tr}\,(\boldsymbol{H}) \leq Lr.$$

**Assumption G.4.** $\mathcal{L}(\boldsymbol{\theta})$ is $\mu$-convex, i.e.,

$$\mathcal{L}(\boldsymbol{\theta}) \geq \mathcal{L}(\boldsymbol{\theta}') + \langle\nabla\mathcal{L}(\boldsymbol{\theta}'), \boldsymbol{\theta} - \boldsymbol{\theta}'\rangle + \frac{\mu}{2}\|\boldsymbol{\theta} - \boldsymbol{\theta}'\|^2, \forall\boldsymbol{\theta},\boldsymbol{\theta}' \in \mathbb{R}^d.$$

## G.2 Useful Lemmas

**Lemma G.5** (Gao et al. (2018), Lemma 4.1 (b))**.** *Suppose Assumption G.1 holds, then the expected gradient estimated with SPSA is a biased estimation of $\nabla\mathcal{L}(\boldsymbol{\theta})$ and satisfies*

$$\left\|\mathbb{E}_{\mathcal{B},\mathbf{z}}[\hat{\nabla}\mathcal{L}(\boldsymbol{\theta};\mathcal{B})] - \nabla\mathcal{L}(\boldsymbol{\theta})\right\|^2 \leq \frac{\epsilon^2 L^2 d^2}{4}.$$

**Lemma G.6** (Malladi et al. (2023), Lemma 2)**.** *Suppose Assumption G.1 and Assumption G.2 holds, then the variance of the gradient estimated with SPSA satisfies*

$$\text{Var}(\hat{\nabla}\mathcal{L}(\boldsymbol{\theta};\mathcal{B})) = \mathbb{E}_{\mathcal{B}}\left[\left\|\mathbb{E}_{\mathcal{B}}[\hat{\nabla}\mathcal{L}(\boldsymbol{\theta};\mathcal{B})] - \hat{\nabla}\mathcal{L}(\boldsymbol{\theta};\mathcal{B})\right\|^2\right] \leq \frac{d}{K}\sigma^2.$$

**Lemma G.7** (Zhang et al. (2023), Lemma C.1 (iv))**.** *Let $\mathbf{z} \sim \mathcal{N}(0, \boldsymbol{I}_d)$, $\boldsymbol{v} \in \mathbb{R}^d$ be some fixed vector, $\boldsymbol{H} \in \mathbb{R}^{d\times d}$ be some fixed matrix independent of $\mathbf{z}$, we have:*

$$\mathbb{E}_{\mathbf{z}}[(\mathbf{z}^\top\boldsymbol{v})^2\mathbf{z}^\top\boldsymbol{H}\mathbf{z}] = \frac{d}{d+2}\left(2\boldsymbol{v}^\top\boldsymbol{H}\boldsymbol{v} + \|\boldsymbol{v}\|^2\,\text{tr}\,(\boldsymbol{H})\right).$$

## G.3 Convergence analysis of Addax in smooth nonconvex setting when $\mathcal{D}^0$ and $\mathcal{D}^1$ are same

The following Theorem is the precise statement of the result behind Theorem 3.1 presented in the main body.

**Theorem G.8.** *Under Assumptions G.1, G.2, by running algorithm 1 for $T$ iterations with $0 < \eta_t = \eta \leq \frac{2-\alpha}{4L}, \forall t$, the output satisfies*

$$
\mathbb{E}[\|\nabla\mathcal{L}(\boldsymbol{\theta}_t)\|^2] \leq \frac{4(\mathcal{L}(\boldsymbol{\theta}_0) - \mathcal{L}_\star)}{\eta T(2-\alpha)} \tag{4}
$$
$$
+ \frac{\alpha(1+\alpha-\alpha^2/2)\epsilon^2 L^2 d^2}{2(2-\alpha)} + \frac{4\eta L}{(2-\alpha)}\left(\frac{(1-\alpha)^2}{2K^1} + \frac{\alpha^2 d}{2K^0}\right)\sigma^2.
$$

*Proof:* By Assumption G.1:

$$
\mathbb{E}_t[\mathcal{L}(\boldsymbol{\theta}_{t+1})] \leq \mathcal{L}(\boldsymbol{\theta}_t) + \mathbb{E}_t[\langle\nabla\mathcal{L}(\boldsymbol{\theta}_t), \boldsymbol{\theta}_{t+1} - \boldsymbol{\theta}_t\rangle] + \frac{L}{2}\mathbb{E}_t[\|\boldsymbol{\theta}_{t+1} - \boldsymbol{\theta}_t\|^2]
$$
$$
\stackrel{(a)}{=} \mathcal{L}(\boldsymbol{\theta}_t) - \eta_t\left\langle\nabla\mathcal{L}(\boldsymbol{\theta}_t), (1-\alpha)\nabla\mathcal{L}(\boldsymbol{\theta}_t) + \alpha\mathbb{E}_{\mathcal{B}^0}[\hat{\nabla}\mathcal{L}(\theta_t;\mathcal{B}^0)]\right\rangle
$$
$$
+ \frac{L\eta_t^2}{2}\left\|(1-\alpha)\nabla\mathcal{L}(\boldsymbol{\theta}_t) + \alpha\mathbb{E}_{\mathcal{B}^0}[\hat{\nabla}\mathcal{L}(\theta_t;\mathcal{B}^0)]\right\|^2
$$
$$
+ \frac{L\eta_t^2(1-\alpha)^2}{2}\mathbb{E}_{\mathcal{B}^1}[\|\nabla\mathcal{L}(\boldsymbol{\theta}_t) - \nabla\mathcal{L}(\theta_t;\mathcal{B}^1)\|^2] + \frac{L\eta_t^2\alpha^2}{2}\text{Var}(\hat{\nabla}\mathcal{L}(\theta_t;\mathcal{B}^0)) \tag{5}
$$
$$
\stackrel{(b)}{\leq} \mathcal{L}(\boldsymbol{\theta}_t) - (1-\alpha)\eta_t\|\nabla\mathcal{L}(\boldsymbol{\theta}_t)\|^2 - \alpha\eta_t\left\langle\nabla\mathcal{L}(\boldsymbol{\theta}_t), \mathbb{E}_{\mathcal{B}^0}[\hat{\nabla}\mathcal{L}(\theta_t;\mathcal{B}^0)]\right\rangle
$$
$$
+ \frac{L\eta_t^2}{2}\left\|(1-\alpha)\nabla\mathcal{L}(\boldsymbol{\theta}_t) + \alpha\mathbb{E}_{\mathcal{B}^0}[\hat{\nabla}\mathcal{L}(\theta_t;\mathcal{B}^0)]\right\|^2
$$
$$
+ \frac{L\eta_t^2(1-\alpha)^2}{2K^1}\sigma^2 + \frac{L\eta_t^2\alpha^2 d}{2K^0}\sigma^2,
$$

where $(a)$ substitutes the update of $\boldsymbol{\theta}$ and takes expectations to $\boldsymbol{g}^0, \boldsymbol{g}^1$; $(b)$ follows from the Lemma G.6. The third term on the Right-Hand-Side (RHS) can be further bounded as

$$
- \alpha\eta_t\left\langle\nabla\mathcal{L}(\boldsymbol{\theta}_t), \mathbb{E}_{\mathcal{B}^0}[\hat{\nabla}\mathcal{L}(\theta_t;\mathcal{B}^0)]\right\rangle
$$
$$
\stackrel{(a)}{=} -\frac{\alpha\eta_t}{2}\|\nabla\mathcal{L}(\boldsymbol{\theta}_t)\|^2 - \frac{\alpha\eta_t}{2}\left\|\mathbb{E}_{\mathcal{B}^0}[\hat{\nabla}\mathcal{L}(\theta_t;\mathcal{B}^0)]\right\|^2 + \frac{\alpha\eta_t}{2}\left\|\nabla\mathcal{L}(\boldsymbol{\theta}_t) - \mathbb{E}_{\mathcal{B}^0}[\hat{\nabla}\mathcal{L}(\theta_t;\mathcal{B}^0)]\right\|^2 \tag{6}
$$
$$
\stackrel{(b)}{\leq} -\frac{\alpha\eta_t}{2}\|\nabla\mathcal{L}(\boldsymbol{\theta}_t)\|^2 - \frac{\alpha\eta_t}{2}\left\|\mathbb{E}_{\mathcal{B}^0}[\hat{\nabla}\mathcal{L}(\theta_t;\mathcal{B}^0)]\right\|^2 + \frac{\alpha\eta_t\epsilon^2 L^2 d^2}{8},
$$

where $(a)$ uses the fact that $\|\mathbf{u} + \mathbf{v}\|^2 = \|\mathbf{u}\|^2 + \|\mathbf{v}\|^2 + 2\langle\mathbf{u}, \mathbf{v}\rangle$; $(b)$ applies Lemma G.5 to the last term. The fourth term on the RHS of equation 5 can be bounded as

$$
\frac{L\eta_t^2}{2}\left\|(1-\alpha)\nabla\mathcal{L}(\boldsymbol{\theta}_t) + \alpha\mathbb{E}_{\mathcal{B}^0}[\hat{\nabla}\mathcal{L}(\theta_t;\mathcal{B}^0)]\right\|^2
$$
$$
= \frac{L\eta_t^2}{2}\left\|\nabla\mathcal{L}(\boldsymbol{\theta}_t) + \alpha\left(\mathbb{E}_{\mathcal{B}^0}[\hat{\nabla}\mathcal{L}(\theta_t;\mathcal{B}^0)] - \nabla\mathcal{L}(\boldsymbol{\theta}_t)\right)\right\|^2
$$
$$
\stackrel{(a)}{\leq} L\eta_t^2\|\nabla\mathcal{L}(\boldsymbol{\theta}_t)\|^2 + \alpha^2 L\eta_t^2\left\|\left(\mathbb{E}_{\mathcal{B}^0}[\hat{\nabla}\mathcal{L}(\theta_t;\mathcal{B}^0)] - \nabla\mathcal{L}(\boldsymbol{\theta}_t)\right)\right\|^2 \tag{7}
$$
$$
\stackrel{(b)}{\leq} L\eta_t^2\|\nabla\mathcal{L}(\boldsymbol{\theta}_t)\|^2 + \frac{\alpha^2\eta_t^2\epsilon^2 L^3 d^2}{4},
$$

where $(a)$ applies Cauchy-Schwarz inequality; $(b)$ applies Lemma G.5 to the last term. Substitute equation 6, equation 7 back to equation 5, we have

$$
\mathbb{E}_t[\mathcal{L}(\boldsymbol{\theta}_{t+1})] \leq \mathcal{L}(\boldsymbol{\theta}_t) - (1 - \frac{\alpha}{2} - L\eta_t)\eta_t\|\nabla\mathcal{L}(\boldsymbol{\theta}_t)\|^2 - \frac{\alpha\eta_t}{2}\left\|\mathbb{E}_{\mathcal{B}^0}[\hat{\nabla}\mathcal{L}(\theta_t;\mathcal{B}^0)]\right\|^2
$$
$$
+ \frac{\alpha\eta_t\epsilon^2 L^2 d^2(1+2\alpha\eta_t L)}{8} + \frac{L\eta_t^2(1-\alpha)^2}{2K^1}\sigma^2 + \frac{L\eta_t^2\alpha^2}{2K^0}\sigma^2. \tag{8}
$$

Choose $\eta_t \leq \frac{2-\alpha}{4L}$, we have $1 - \frac{\alpha}{2} - L\eta_t \geq \frac{2-\alpha}{4} > 0, 1 + 2\alpha\eta_t L \leq 1 + \alpha - \frac{\alpha^2}{2}$ and

$$
\frac{(2-\alpha)\eta_t}{4}\|\nabla\mathcal{L}(\boldsymbol{\theta}_t)\|^2 + \frac{\alpha\eta_t}{2}\left\|\mathbb{E}_{\mathcal{B}^0}[\hat{\nabla}\mathcal{L}(\theta_t;\mathcal{B}^0)]\right\|^2 \leq \mathcal{L}(\boldsymbol{\theta}_t) - \mathbb{E}_t[\mathcal{L}(\boldsymbol{\theta}_{t+1})]
$$
$$
+ \frac{\alpha\eta_t\epsilon^2 L^2 d^2(1+\alpha-\alpha^2/2)}{8} + \frac{L\eta_t^2(1-\alpha)^2}{2K^1}\sigma^2 + \frac{L\eta_t^2\alpha^2}{2K^0}\sigma^2. \tag{9}
$$

Sum from $t = 0$ to $T$, we have

$$\sum_{t=0}^{T} \left( \frac{(2-\alpha)\eta_t}{4} \mathbb{E}[\|\nabla\mathcal{L}(\boldsymbol{\theta}_t)\|^2] + \frac{\alpha\eta_t}{2} \mathbb{E}\left[\left\|\mathbb{E}_{\mathcal{B}^0}[\hat{\nabla}\mathcal{L}(\theta_t; \mathcal{B}^0)]\right\|^2\right] \right) \le \mathcal{L}(\boldsymbol{\theta}_0) - \mathbb{E}[\mathcal{L}(\boldsymbol{\theta}_{T+1})]$$
$$+ \sum_{t=0}^{T} \eta_t \cdot \frac{\alpha(1 + \alpha - \alpha^2/2)\epsilon^2 L^2 d^2}{8} + \sum_{t=0}^{T} \eta_t^2 \cdot \left( \frac{L(1-\alpha)^2}{2K^1}\sigma^2 + \frac{L\alpha^2 d}{2K^0}\sigma^2 \right). \tag{10}$$

Choose $\eta_t = \eta \le \frac{2-\alpha}{4L}, \forall t$, and divide both side by $\frac{(2-\alpha)\eta T}{4}$, we have

$$\mathbb{E}[\|\nabla\mathcal{L}(\boldsymbol{\theta}_t)\|^2] + \frac{2\alpha}{2-\alpha}\mathbb{E}\left[\left\|\mathbb{E}_{\mathcal{B}^0}[\hat{\nabla}\mathcal{L}(\theta_t; \mathcal{B}^0)]\right\|^2\right] \le \frac{4(\mathcal{L}(\boldsymbol{\theta}_0) - \mathcal{L}_\star)}{\eta T(2-\alpha)}$$
$$+ \frac{\alpha(1 + \alpha - \alpha^2/2)\epsilon^2 L^2 d^2}{2(2-\alpha)} + \frac{4\eta L}{(2-\alpha)} \left( \frac{(1-\alpha)^2}{2K^1} + \frac{\alpha^2 d}{2K^0} \right)\sigma^2, \tag{11}$$

which completes the proof.

**Corollary G.9.** *By choosing* $\eta = \min\left\{ \frac{2-\alpha}{4L}, \sqrt{\frac{2(\mathcal{L}(\boldsymbol{\theta}_0) - \mathcal{L}_\star)}{TL\sigma^2\left(\frac{(1-\alpha)^2}{K^1} + \frac{\alpha^2 d}{K^0}\right)}} \right\}$ *and*

$$\epsilon \le \left( \frac{2(\mathcal{L}(\boldsymbol{\theta}_0) - \mathcal{L}_\star)\sigma^2 \left((1-\alpha)^2/K^1 + \alpha^2 d/K^0\right)}{T} \right)^{1/4} \cdot \frac{1}{L^{3/4}d\sqrt{\alpha(1 + \alpha - \alpha^2/2)}},$$

*algorithm 1 converges with rate*

$$\mathbb{E}[\|\nabla\mathcal{L}(\boldsymbol{\theta}_t)\|^2] \le 5\sqrt{2L} \cdot \frac{\sqrt{\frac{(1-\alpha)^2}{K^1} + \frac{\alpha^2 d}{K^0}}}{2-\alpha} \cdot \sigma\sqrt{\frac{\mathcal{L}(\boldsymbol{\theta}_0) - \mathcal{L}_\star}{T}}$$
$$= \mathcal{O}\left( \frac{1}{\sqrt{T}} \cdot \sqrt{\frac{(1-\alpha)^2}{K^1} + \frac{\alpha^2 d}{K^0}} \right)$$

## G.4 CONVERGENCE ANALYSIS OF ADDAX IN SMOOTH NONCONVEX SETTING WHEN $\mathcal{D}^0$ AND $\mathcal{D}^1$ ARE DIFFERENT

Assume that datasets $\mathcal{D}^0$ and $\mathcal{D}^1$ are different. Consider loss functions $\mathcal{L}$, $\mathcal{L}^0$, and $\mathcal{L}^1$, where $\mathcal{L}^0$ and $\mathcal{L}^1$ are evaluated on datasets $\mathcal{D}^0$ and $\mathcal{D}^1$ of sizes $N^0$ and $N^1$, respectively. The combined loss function $\mathcal{L}$ is defined as:

$$\mathcal{L} = \frac{(N^0 \mathcal{L}^0 + N^1 \mathcal{L}^1)}{N^0 + N^1}$$

**Theorem G.10.** *Assuming that the loss functions* $\mathcal{L}$, $\mathcal{L}^0$, *and* $\mathcal{L}^1$ *satisfy Assumptions G.1 and G.2 and* $L_T \le L_{max}$, *running Algorihtm 1 for $T$ iterations with a learning rate* $0 < \eta_t = \eta \le \frac{2\alpha}{L}, \forall t$, $\epsilon^{1/2} \le \frac{1}{\|\nabla\mathcal{L}(\boldsymbol{\theta}_t)\|}$ *and choose* $\alpha = \frac{N^0}{N^0 + N^1}$, *the output satisfies*

$$\mathbb{E}[\|\nabla\mathcal{L}(\boldsymbol{\theta}_t)\|^2] \le \frac{\mathcal{L}(\boldsymbol{\theta}_0) - \mathcal{L}_\star}{\eta T(1-\alpha)}$$
$$+ \frac{\epsilon^{1/2}Ld\alpha(6 + \epsilon^{3/2}Ld\alpha^2)}{4(1-\alpha)} + \frac{\eta L}{(1-\alpha)} \left( \frac{(1-\alpha)^2}{2K^1} + \frac{\alpha^2 d}{2K^0} \right)\sigma^2 \tag{12}$$

*Proof:* By Assumption G.1:

$$
\begin{aligned}
\mathbb{E}_t[\mathcal{L}(\boldsymbol{\theta}_{t+1})] &\leq \mathcal{L}(\boldsymbol{\theta}_t) + \mathbb{E}_t[\langle \nabla\mathcal{L}(\boldsymbol{\theta}_t), \boldsymbol{\theta}_{t+1} - \boldsymbol{\theta}_t \rangle] + \frac{L}{2}\mathbb{E}_t[\|\boldsymbol{\theta}_{t+1} - \boldsymbol{\theta}_t\|^2] \\
&\stackrel{(a)}{=} \mathcal{L}(\boldsymbol{\theta}_t) - \eta_t \left\langle \nabla\mathcal{L}(\boldsymbol{\theta}_t), (1-\alpha)\nabla\mathcal{L}^1(\boldsymbol{\theta}_t) + \alpha\mathbb{E}_{\mathcal{B}^0}[\hat{\nabla}\mathcal{L}^0(\theta_t; \mathcal{B}^0)] \right\rangle \\
&\quad + \frac{L\eta_t^2}{2} \left\| (1-\alpha)\nabla\mathcal{L}^1(\boldsymbol{\theta}_t) + \alpha\mathbb{E}_{\mathcal{B}^0}[\hat{\nabla}\mathcal{L}^0(\theta_t; \mathcal{B}^0)] \right\|^2 \\
&\quad + \frac{L\eta_t^2(1-\alpha)^2}{2}\mathbb{E}_{\mathcal{B}^1}[\|\nabla\mathcal{L}^1(\boldsymbol{\theta}_t) - \nabla\mathcal{L}^1(\boldsymbol{\theta}_t; \mathcal{B}^1)\|^2] + \frac{L\eta_t^2\alpha^2}{2}\mathrm{Var}(\hat{\nabla}\mathcal{L}^0(\theta_t; \mathcal{B}^0)) \\
&\stackrel{(b)}{\leq} \mathcal{L}(\boldsymbol{\theta}_t) - \eta_t \left\langle \nabla\mathcal{L}(\boldsymbol{\theta}_t), (1-\alpha)\nabla\mathcal{L}^1(\boldsymbol{\theta}_t) + \alpha\mathbb{E}_{\mathcal{B}^0}[\hat{\nabla}\mathcal{L}^0(\theta_t; \mathcal{B}^0)] \right\rangle \\
&\quad + \frac{L\eta_t^2}{2}\left( \left\| (1-\alpha)\nabla\mathcal{L}^1(\boldsymbol{\theta}_t) + \alpha\nabla\mathcal{L}^0(\boldsymbol{\theta}_t) \right\| + \left\| \mathbb{E}_{\mathcal{B}^0}[\hat{\nabla}\mathcal{L}^0(\boldsymbol{\theta}_t; \mathcal{B}^0)] - \nabla\mathcal{L}^0(\boldsymbol{\theta}_t) \right\| \right)^2 \\
&\quad + \frac{L\eta_t^2(1-\alpha)^2}{2K^1}\sigma^2 + \frac{L\eta_t^2\alpha^2 d}{2K^0}\sigma^2 \\
&\stackrel{(c)}{\leq} \mathcal{L}(\boldsymbol{\theta}_t) - \eta_t \left\langle \nabla\mathcal{L}(\boldsymbol{\theta}_t), (1-\alpha)\nabla\mathcal{L}^1(\boldsymbol{\theta}_t) + \alpha\mathbb{E}_{\mathcal{B}^0}[\hat{\nabla}\mathcal{L}^0(\theta_t; \mathcal{B}^0)] \right\rangle \\
&\quad + \frac{L\eta_t^2}{2}\|\nabla\mathcal{L}(\boldsymbol{\theta}_t)\|^2 + \frac{L^2\eta_t^2\epsilon d}{2}\|\nabla\mathcal{L}(\boldsymbol{\theta}_t)\| + \frac{\epsilon^2 L^3\eta_t^2\alpha^2 d^2}{8} \\
&\quad + \frac{L\eta_t^2(1-\alpha)^2}{2K^1}\sigma^2 + \frac{L\eta_t^2\alpha^2 d}{2K^0}\sigma^2
\end{aligned}
$$

$$(13)$$

where $(a)$ substitutes the update of $\boldsymbol{\theta}$ and takes expectation of $\boldsymbol{g}^0$, $\boldsymbol{g}^1$; $(b)$ add and subtract $\alpha\nabla\mathcal{L}^0(\boldsymbol{\theta}_t)$ and Cauchy-Schwartz inequality to the third term and the last two terms follow from the Lemma G.6; $(c)$ follows from the Lemma G.5. The second term on the Right-Hand-Side (RHS) can be further bounded by

$$
\begin{aligned}
&- \eta_t \left\langle \nabla\mathcal{L}(\boldsymbol{\theta}_t), (1-\alpha)\nabla\mathcal{L}^1(\boldsymbol{\theta}_t) + \alpha\mathbb{E}_{\mathcal{B}^0}[\hat{\nabla}\mathcal{L}^0(\theta_t; \mathcal{B}^0)] \right\rangle \\
&= -\eta_t \left\langle \nabla\mathcal{L}(\boldsymbol{\theta}_t), (1-\alpha)\nabla\mathcal{L}^1(\boldsymbol{\theta}_t) + \alpha\nabla\mathcal{L}^0(\boldsymbol{\theta}_t) - \alpha\nabla\mathcal{L}^0(\boldsymbol{\theta}_t) + \alpha\mathbb{E}_{\mathcal{B}^0}[\hat{\nabla}\mathcal{L}^0(\theta_t; \mathcal{B}^0)] \right\rangle \\
&= -\eta_t \|\nabla\mathcal{L}(\boldsymbol{\theta}_t)\|^2 - \alpha\eta_t \left\langle \nabla\mathcal{L}(\boldsymbol{\theta}_t), \mathbb{E}_{\mathcal{B}^0}[\hat{\nabla}\mathcal{L}^0(\theta_t; \mathcal{B}^0)] - \nabla\mathcal{L}^0(\boldsymbol{\theta}_t) \right\rangle
\end{aligned}
$$

$$(14)$$

The second term in equation 14 can be further bounded by

$$
\begin{aligned}
&- \alpha\eta_t \left\langle \nabla\mathcal{L}(\boldsymbol{\theta}_t), \mathbb{E}_{\mathcal{B}^0}[\hat{\nabla}\mathcal{L}^0(\theta_t; \mathcal{B}^0)] - \nabla\mathcal{L}^0(\boldsymbol{\theta}_t) \right\rangle \\
&\stackrel{(a)}{\leq} \alpha\eta_t \|\nabla\mathcal{L}(\boldsymbol{\theta}_t)\| \left\| \mathbb{E}_{\mathcal{B}^0}[\hat{\nabla}\mathcal{L}^0(\theta_t; \mathcal{B}^0)] - \nabla\mathcal{L}^0(\boldsymbol{\theta}_t) \right\| \\
&\stackrel{(b)}{\leq} \frac{\alpha\eta_t\epsilon Ld}{2}\|\nabla\mathcal{L}(\boldsymbol{\theta}_t)\|
\end{aligned}
$$

$$(15)$$

where $(a)$ applies Cauchy-Schwartz inequality; $(b)$ applies Lemma G.5. Substitute equation 16 back to equation 14, we have

$$
\begin{aligned}
&- \eta_t \left\langle \nabla\mathcal{L}(\boldsymbol{\theta}_t), (1-\alpha)\nabla\mathcal{L}^1(\boldsymbol{\theta}_t) + \alpha\mathbb{E}_{\mathcal{B}^0}[\hat{\nabla}\mathcal{L}^0(\theta_t; \mathcal{B}^0)] \right\rangle \\
&\leq -\eta_t \|\nabla\mathcal{L}(\boldsymbol{\theta}_t)\|^2 + \frac{\alpha\eta_t\epsilon Ld}{2}\|\nabla\mathcal{L}(\boldsymbol{\theta}_t)\|
\end{aligned}
$$

$$(16)$$

Substitute equation 16 back to equation 13, we have

$$
\begin{aligned}
\mathbb{E}_t[\mathcal{L}(\boldsymbol{\theta}_{t+1})] \leq{}& \mathcal{L}(\boldsymbol{\theta}_t) - (1 - \frac{L\eta_t}{2})\eta_t \left\| \nabla\mathcal{L}(\boldsymbol{\theta}_t) \right\|^2 \\
& + \frac{\epsilon L d(\alpha + L\eta_t)}{2}\eta_t \left\| \nabla\mathcal{L}(\boldsymbol{\theta}_t) \right\| + \frac{\epsilon^2 L^3 \eta_t \alpha^2 d^2}{8}\eta_t + \frac{L\eta_t^2(1-\alpha)^2}{2K^1}\sigma^2 + \frac{L\eta_t^2 \alpha^2 d}{2K^0}\sigma^2 \\
\stackrel{(a)}{\leq}{}& \mathcal{L}(\boldsymbol{\theta}_t) - (1 - \frac{L\eta_t}{2})\eta_t \left\| \nabla\mathcal{L}(\boldsymbol{\theta}_t) \right\|^2 \\
& + \frac{\epsilon^{1/2} L d(\alpha + L\eta_t)}{2}\eta_t + \frac{\epsilon^2 L^3 \eta_t \alpha^2 d^2}{8}\eta_t + \frac{L\eta_t^2(1-\alpha)^2}{2K^1}\sigma^2 + \frac{L\eta_t^2 \alpha^2 d}{2K^0}\sigma^2
\end{aligned}
\tag{17}
$$

where $(a)$ choose that $\epsilon^{1/2} \leq \frac{1}{\|\nabla\mathcal{L}(\boldsymbol{\theta}_t)\|}$, then $\epsilon^{1/2} \left\| \nabla\mathcal{L}(\boldsymbol{\theta}_t) \right\| \leq 1$.

Choose $\eta_t \leq \frac{2\alpha}{L}$, we have $1 - \frac{L\eta_t}{2} \geq 1 - \alpha > 0$, $\eta_t L \leq 2\alpha$ and

$$
\begin{aligned}
(1-\alpha)\eta_t \left\| \nabla\mathcal{L}(\boldsymbol{\theta}_t) \right\|^2 \leq{}& \mathcal{L}(\boldsymbol{\theta}_t) - \mathbb{E}_t[\mathcal{L}(\boldsymbol{\theta}_{t+1})] \\
& + \frac{\epsilon^{1/2} L d\alpha(6 + \epsilon^{3/2} L d\alpha^2)}{4}\eta_t + \frac{L\eta_t^2(1-\alpha)^2}{2K^1}\sigma^2 + \frac{L\eta_t^2 \alpha^2 d}{2K^0}\sigma^2
\end{aligned}
\tag{18}
$$

Sum from $t = 0$ to $T$ we have

$$
\begin{aligned}
\sum_{t=0}^{T} \left( (1-\alpha)\eta_t \mathbb{E}[\|\nabla\mathcal{L}(\boldsymbol{\theta}_t)\|^2] \right) \leq{}& \mathcal{L}(\boldsymbol{\theta}_0) - \mathbb{E}[\mathcal{L}(\boldsymbol{\theta}_{T+1})] \\
& + \sum_{t=0}^{T} \frac{\epsilon^{1/2} L d\alpha(6 + \epsilon^{3/2} L d\alpha^2)}{4}\eta_t + \sum_{t=0}^{T} \eta_t^2 \cdot \left( \frac{L(1-\alpha)^2}{2K^1}\sigma^2 + \frac{L\alpha^2 d}{2K^0}\sigma^2 \right)
\end{aligned}
\tag{19}
$$

Choosing $\eta_t = \eta \leq \frac{2\alpha}{L}, \forall t$ and dividing both sides by $(1-\alpha)\eta T$, we have

$$
\begin{aligned}
\mathbb{E}[\|\nabla\mathcal{L}(\boldsymbol{\theta}_t)\|^2] \leq{}& \frac{\mathcal{L}(\boldsymbol{\theta}_0) - \mathcal{L}_\star}{\eta T(1-\alpha)} \\
& + \frac{\epsilon^{1/2} L d\alpha(6 + \epsilon^{3/2} L d\alpha^2)}{4(1-\alpha)} + \frac{\eta L}{(1-\alpha)} \left( \frac{(1-\alpha)^2}{2K^1} + \frac{\alpha^2 d}{2K^0} \right)\sigma^2
\end{aligned}
\tag{20}
$$

where the expectation is taken over $t$ (uniformly) and the randomness of the algorithm. This completes the proof.

**Corollary G.11.** *By choosing* $\eta = \min\left\{ \frac{2\alpha}{L}, \sqrt{\frac{2(\mathcal{L}(\boldsymbol{\theta}_0) - \mathcal{L}_\star)}{TL\sigma^2\left(\frac{(1-\alpha)^2}{K^1} + \frac{\alpha^2 d}{K^0}\right)}} \right\}$ *and*

$$
\epsilon \leq \min \left\{
\begin{array}{c}
\frac{1}{\overline{\|\nabla\mathcal{L}(\boldsymbol{\theta}_t)\|^2}} \\[4pt]
\left( \frac{(\mathcal{L}(\boldsymbol{\theta}_0) - \mathcal{L}_\star)\sigma^2\left((1-\alpha)^2/K^1 + \alpha^2 d/K^0\right)}{2T} \right)^{1/4} \cdot \frac{2}{L^{3/4} d\alpha^{3/2}} \\[4pt]
\frac{2(\mathcal{L}(\boldsymbol{\theta}_0) - \mathcal{L}_\star)\sigma^2\left((1-\alpha)^2/K^1 + \alpha^2 d/K^0\right)}{9TLd^2\alpha^2}
\end{array}
\right\}
$$

*Addax converges with rate*

$$
\begin{aligned}
\mathbb{E}[\|\nabla\mathcal{L}(\boldsymbol{\theta}_t)\|^2] &\leq \frac{\sqrt{2L}}{2} \cdot \frac{\sqrt{\frac{(1-\alpha)^2}{K^1} + \frac{\alpha^2 d}{K^0}}}{1-\alpha} \cdot \sigma \sqrt{\frac{\mathcal{L}(\boldsymbol{\theta}_0) - \mathcal{L}_\star}{T}} \\
&= \mathcal{O}\left( \frac{1}{\sqrt{T}} \cdot \sqrt{\frac{(1-\alpha)^2}{K^1} + \frac{\alpha^2 d}{K^0}} \right)
\end{aligned}
$$

### G.5 CONVERGENCE OF ADDAX WITH LOW EFFICIENT RANK HESSIAN

**Theorem G.12.** *Under Assumption G.1-Assumption G.3, by running algorithm 1 for T iterations with $\eta_t = \eta \le \min\{\frac{1}{(1-\alpha)L}, \frac{2-\alpha}{1-\alpha+2\alpha^2 L(2+r)}\}, \forall t$, the output satisfies*

$$\mathbb{E}[\|\nabla\mathcal{L}(\boldsymbol{\theta}_t)\|^2] \le \frac{\mathcal{L}(\boldsymbol{\theta}_0) - \mathcal{L}_\star}{\eta C_1 T} + \frac{\alpha\epsilon^2 L^2 d^2 (1 + 2\eta\alpha L r)}{8C_1}$$
$$+ \frac{\eta L\sigma^2}{2C_1}\left(\frac{(1-\alpha)^2}{K^1} + \frac{2(2+r)\alpha^2}{K^0}\right), \tag{21}$$

*where $C_1 = 1 - \frac{\alpha}{2} - \frac{\eta L}{2}\left(1 - \alpha + 2\alpha^2 L(2+r)\right)$.*

*Proof:* Using the Taylor expansion with Lagrange remainder, we have:

$$\mathcal{L}(\boldsymbol{\theta}_{t+1}) = \mathcal{L}(\boldsymbol{\theta}_t) + \langle\nabla\mathcal{L}(\boldsymbol{\theta}_t), \boldsymbol{\theta}_{t+1} - \boldsymbol{\theta}_t\rangle + \frac{1}{2}(\boldsymbol{\theta}_{t+1} - \boldsymbol{\theta}_t)^\top\nabla\mathcal{L}(\boldsymbol{\theta}')(\boldsymbol{\theta}_{t+1} - \boldsymbol{\theta}_t), \tag{22}$$

where $\boldsymbol{\theta}' = \lambda\boldsymbol{\theta}_t + (1-\lambda)\boldsymbol{\theta}_{t+1}$, for some $\lambda \in [0, 1]$. Taking expectation conditioned on everything until $t$, we have:

$$\mathbb{E}_t[\mathcal{L}(\boldsymbol{\theta}_{t+1})] = \mathcal{L}(\boldsymbol{\theta}_t) + \langle\nabla\mathcal{L}(\boldsymbol{\theta}_t), \mathbb{E}_t[\boldsymbol{\theta}_{t+1} - \boldsymbol{\theta}_t]\rangle + \frac{1}{2}\mathbb{E}_t\left[(\boldsymbol{\theta}_{t+1} - \boldsymbol{\theta}_t)^\top\nabla\mathcal{L}(\boldsymbol{\theta}')(\boldsymbol{\theta}_{t+1} - \boldsymbol{\theta}_t)\right]$$

$$\overset{(a)}{\le} \mathcal{L}(\boldsymbol{\theta}_t) - \eta_t\left\langle\nabla\mathcal{L}(\boldsymbol{\theta}_t), (1-\alpha)\nabla\mathcal{L}(\boldsymbol{\theta}_t) + \alpha\mathbb{E}_{\mathcal{B}^0}[\hat{\nabla}\mathcal{L}(\boldsymbol{\theta}_t; \mathcal{B}^0)]\right\rangle$$

$$+ \frac{\eta_t^2}{2}\mathbb{E}\left[\left\langle(1-\alpha)\nabla\mathcal{L}(\boldsymbol{\theta}_t; \mathcal{B}^1) + \alpha\hat{\nabla}\mathcal{L}(\boldsymbol{\theta}_t; \mathcal{B}^0),\right.\right.$$

$$\left.\left. \boldsymbol{H}\left((1-\alpha)\nabla\mathcal{L}(\boldsymbol{\theta}_t; \mathcal{B}^1) + \alpha\hat{\nabla}\mathcal{L}(\boldsymbol{\theta}_t; \mathcal{B}^0)\right)\right\rangle\right]$$

$$= \mathcal{L}(\boldsymbol{\theta}_t) - \eta_t(1-\alpha)\|\nabla\mathcal{L}(\boldsymbol{\theta}_t)\|^2 - \eta_t\alpha\left\langle\nabla\mathcal{L}(\boldsymbol{\theta}_t), \mathbb{E}_{\mathcal{B}^0}[\hat{\nabla}\mathcal{L}(\boldsymbol{\theta}_t; \mathcal{B}^0)]\right\rangle$$

$$+ \frac{\eta_t^2(1-\alpha)^2}{2}\mathbb{E}_{\mathcal{B}_1}\left[\langle\nabla\mathcal{L}(\boldsymbol{\theta}_t; \mathcal{B}^1), \boldsymbol{H}\nabla\mathcal{L}(\boldsymbol{\theta}_t; \mathcal{B}^1)\rangle\right]$$

$$+ \frac{\eta_t^2\alpha^2}{2}\mathbb{E}_{\mathcal{B}^0}\left[\left\langle\hat{\nabla}\mathcal{L}(\boldsymbol{\theta}_t; \mathcal{B}^0), \boldsymbol{H}\hat{\nabla}\mathcal{L}(\boldsymbol{\theta}_t; \mathcal{B}^0)\right\rangle\right]$$

$$+ \eta_t^2\alpha(1-\alpha)\left\langle\mathbb{E}_{\mathcal{B}^0}[\hat{\nabla}\mathcal{L}(\boldsymbol{\theta}_t; \mathcal{B}^0)], \boldsymbol{H}\mathbb{E}_{\mathcal{B}^1}[\nabla\mathcal{L}(\boldsymbol{\theta}_t; \mathcal{B}^1)]\right\rangle$$

$$\overset{(b)}{\le} \mathcal{L}(\boldsymbol{\theta}_t) - \eta_t(1-\alpha)\|\nabla\mathcal{L}(\boldsymbol{\theta}_t)\|^2 - \eta_t\alpha\left\langle\nabla\mathcal{L}(\boldsymbol{\theta}_t), \mathbb{E}_{\mathcal{B}^0}[\hat{\nabla}\mathcal{L}(\boldsymbol{\theta}_t; \mathcal{B}^0)]\right\rangle$$

$$+ \frac{\eta_t^2(1-\alpha)^2 L}{2}\left(\|\nabla\mathcal{L}(\boldsymbol{\theta}_t)\|^2 + \frac{\sigma^2}{K^1}\right)$$

$$+ \frac{\eta_t^2\alpha^2}{2}\mathbb{E}_{\mathcal{B}^0}\left[\left\langle\hat{\nabla}\mathcal{L}(\boldsymbol{\theta}_t; \mathcal{B}^0), \boldsymbol{H}\hat{\nabla}\mathcal{L}(\boldsymbol{\theta}_t; \mathcal{B}^0)\right\rangle\right]$$

$$+ \eta_t^2\alpha(1-\alpha)\left\langle\nabla\mathcal{L}(\boldsymbol{\theta}_t), \boldsymbol{H}\mathbb{E}_{\mathcal{B}^0}[\hat{\nabla}\mathcal{L}(\boldsymbol{\theta}_t; \mathcal{B}^0)]\right\rangle, \tag{23}$$

where $(a)$ substitute the update rule of algorithm 1; $(b)$ uses the fact that $\mathcal{B}^0$ and $\mathcal{B}^1$ are independent to the last term, and applies Assumption G.2 to the fourth term. Next, we bound the last two terms separately. For the fifth term, we have:

$$\mathbb{E}_{\mathcal{B}^0}\left[\left\langle\hat{\nabla}\mathcal{L}(\boldsymbol{\theta}_t; \mathcal{B}^0), \boldsymbol{H}\hat{\nabla}\mathcal{L}(\boldsymbol{\theta}_t; \mathcal{B}^0)\right\rangle\right] = \mathbb{E}_{\mathcal{B}^0}\left[\text{tr}\left(\left\langle\hat{\nabla}\mathcal{L}(\boldsymbol{\theta}_t; \mathcal{B}^0), \boldsymbol{H}\hat{\nabla}\mathcal{L}(\boldsymbol{\theta}_t; \mathcal{B}^0)\right\rangle\right)\right]$$

$$\overset{(a)}{=} \mathbb{E}_{\mathcal{B}^0}\left[\text{tr}\left(\boldsymbol{H}\hat{\nabla}\mathcal{L}(\boldsymbol{\theta}_t; \mathcal{B}^0)^\top\mathcal{L}(\boldsymbol{\theta}_t; \mathcal{B}^0)\right)\right]$$

$$\overset{(b)}{=} \text{tr}\left(\boldsymbol{H}\mathbb{E}_{\mathcal{B}^0}\left[\hat{\nabla}\mathcal{L}(\boldsymbol{\theta}_t; \mathcal{B}^0)^\top\mathcal{L}(\boldsymbol{\theta}_t; \mathcal{B}^0)\right]\right) \tag{24}$$

where the $(a)$ uses the property to trace that $\text{tr}(ABC) = \text{tr}(BCA)$, and $(b)$ uses the fact that trace is a linear operator so $\mathbb{E}[\text{tr}(\cdot)] = \text{tr}(\mathbb{E}[\cdot])$. We have:

$$\mathbb{E}_{\mathcal{B}^0, \mathbf{z}}\left[\hat{\nabla}\mathcal{L}(\boldsymbol{\theta}_t; \mathcal{B}^0)^\top\hat{\nabla}\mathcal{L}(\boldsymbol{\theta}_t; \mathcal{B}^0)\right] = \mathbb{E}_{\mathcal{B}^0, \mathbf{z}}\left[\left(\frac{\mathcal{L}(\boldsymbol{\theta}_t + \epsilon\mathbf{z}; \mathcal{B}^0) - \mathcal{L}(\boldsymbol{\theta}_t - \epsilon\mathbf{z}; \mathcal{B}^0)}{2\epsilon}\right)^2\mathbf{z}\mathbf{z}^\top\right]$$

$$\overset{(a)}{\leq} \frac{1}{2\epsilon^2}\mathbb{E}_{\mathcal{B}^0,\mathbf{z}}\left[(2\epsilon\mathbf{z}^\top\nabla\mathcal{L}(\boldsymbol{\theta}_t;\mathcal{B}^0))^2\mathbf{z}\mathbf{z}^\top\right]$$

$$+ \frac{1}{2\epsilon^2}\mathbb{E}_{\mathcal{B}^0,\mathbf{z}}\left[\left(\mathcal{L}(\boldsymbol{\theta}_t+\epsilon\mathbf{z};\mathcal{B}^0)-\mathcal{L}(\boldsymbol{\theta}_t-\epsilon\mathbf{z};\mathcal{B}^0)-2\epsilon\mathbf{z}^\top\nabla\mathcal{L}(\boldsymbol{\theta}_t;\mathcal{B}^0)\right)^2\mathbf{z}\mathbf{z}^\top\right]$$

$$\overset{(b)}{\leq} 2\mathbb{E}_{\mathcal{B}^0,\mathbf{z}}\left[(\mathbf{z}^\top\nabla\mathcal{L}(\boldsymbol{\theta}_t;\mathcal{B}^0))^2\mathbf{z}\mathbf{z}^\top\right]$$

$$+ \frac{1}{\epsilon^2}\mathbb{E}_{\mathcal{B}^0,\mathbf{z}}\left[\left(\mathcal{L}(\boldsymbol{\theta}_t+\epsilon\mathbf{z};\mathcal{B}^0)-\mathcal{L}(\boldsymbol{\theta}_t;\mathcal{B}^0)-\epsilon\mathbf{z}^\top\nabla\mathcal{L}(\boldsymbol{\theta}_t;\mathcal{B}^0)\right)^2\mathbf{z}\mathbf{z}^\top\right]$$

$$+ \frac{1}{\epsilon^2}\mathbb{E}_{\mathcal{B}^0,\mathbf{z}}\left[\left(\mathcal{L}(\boldsymbol{\theta}_t;\mathcal{B}^0)-\mathcal{L}(\boldsymbol{\theta}_t-\epsilon\mathbf{z};\mathcal{B}^0)-\epsilon\mathbf{z}^\top\nabla\mathcal{L}(\boldsymbol{\theta}_t;\mathcal{B}^0)\right)^2\mathbf{z}\mathbf{z}^\top\right]$$

$$\overset{Assumption\ G.1}{\leq} 2\mathbb{E}_{\mathcal{B}^0,\mathbf{z}}\left[(\mathbf{z}^\top\nabla\mathcal{L}(\boldsymbol{\theta}_t;\mathcal{B}^0))^2\mathbf{z}\mathbf{z}^\top\right] + \frac{2}{\epsilon^2}\mathbb{E}_{\mathcal{B}^0,\mathbf{z}}\left[\left(\frac{dL\epsilon^2}{2}\right)^2\mathbf{z}\mathbf{z}^\top\right]$$

$$\overset{(c)}{=} 2\mathbb{E}_{\mathcal{B}^0,\mathbf{z}}\left[(\mathbf{z}^\top\nabla\mathcal{L}(\boldsymbol{\theta}_t;\mathcal{B}^0))^2\mathbf{z}\mathbf{z}^\top\right] + \frac{d^2L^2\epsilon^2}{2}\boldsymbol{I}_d, \tag{25}$$

where $(a)$ extracts the constant, add and subtract $2\epsilon\nabla\mathcal{L}(\boldsymbol{\theta}_t;\mathcal{B}^0)$ and uses the Cauchy–Schwarz inequality; $(b)$ add and subtract $\mathcal{L}(\boldsymbol{\theta}_t;\mathcal{B}_0)$ to the second term, then applies the Cauchy–Schwarz inequality; by Assumption G.1, we have $\left|\mathcal{L}(\boldsymbol{\theta}+\epsilon\mathbf{z};\mathcal{B})-\mathcal{L}(\boldsymbol{\theta};\mathcal{B})-\epsilon\mathbf{z}^\top\nabla\mathcal{L}(\boldsymbol{\theta};\mathcal{B})\right|\leq\frac{L\epsilon^2d}{2}$; and $(c)$ uses the fact that $\mathbb{E}[\mathbf{z}\mathbf{z}^\top]=\boldsymbol{I}_d$ as $\mathbf{z}\sim\mathcal{N}(0,\boldsymbol{I}_d)$. Substitute equation 25 to equation 24, we have:

$$\mathbb{E}_{\mathcal{B}^0}\left[\left\langle\hat{\nabla}\mathcal{L}(\boldsymbol{\theta}_t;\mathcal{B}^0),\boldsymbol{H}\hat{\nabla}\mathcal{L}(\boldsymbol{\theta}_t;\mathcal{B}^0)\right\rangle\right]$$

$$\leq 2\mathrm{tr}\left(\boldsymbol{H}\mathbb{E}_{\mathcal{B}^0,\mathbf{z}}\left[(\mathbf{z}^\top\nabla\mathcal{L}(\boldsymbol{\theta}_t;\mathcal{B}^0))^2\mathbf{z}\mathbf{z}^\top\right]\right) + \mathrm{tr}\left(\frac{d^2L^2\epsilon^2}{2}\boldsymbol{H}\right)$$

$$= 2\mathbb{E}_{\mathcal{B}^0,\mathbf{z}}\left[(\mathbf{z}^\top\nabla\mathcal{L}(\boldsymbol{\theta}_t;\mathcal{B}^0))^2\mathbf{z}^\top\boldsymbol{H}\mathbf{z}\right] + \frac{rd^2L^3\epsilon^2}{2}$$

$$\overset{Lemma\ G.7}{=} \frac{2d}{d+2}\mathbb{E}_{\mathcal{B}^0}\left[2\nabla\mathcal{L}(\boldsymbol{\theta}_t;\mathcal{B}^0)^\top\boldsymbol{H}\nabla\mathcal{L}(\boldsymbol{\theta}_t;\mathcal{B}^0)+\left\|\nabla\mathcal{L}(\boldsymbol{\theta}_t;\mathcal{B}^0)\right\|^2\mathrm{tr}(\boldsymbol{H})\right] + \frac{rd^2L^3\epsilon^2}{2}$$

$$\overset{Assumption\ G.3}{\leq} \frac{2dL(2+r)}{d+2}\mathbb{E}_{\mathcal{B}_0}\left[\left\|\nabla\mathcal{L}(\boldsymbol{\theta}_t;\mathcal{B}^0)\right\|^2\right] + \frac{rd^2L^3\epsilon^2}{2}$$

$$\overset{Assumption\ G.2}{\leq} \frac{2dL(2+r)}{d+2}\left(\left\|\nabla\mathcal{L}(\boldsymbol{\theta}_t)\right\|^2+\frac{\sigma^2}{K^0}\right) + \frac{rd^2L^3\epsilon^2}{2}. \tag{26}$$

For the last term in equation 23, we applies the Cauchy–Schwarz inequality:

$$\left\langle\nabla\mathcal{L}(\boldsymbol{\theta}_t),\boldsymbol{H}\mathbb{E}_{\mathcal{B}^0}[\hat{\nabla}\mathcal{L}(\boldsymbol{\theta}_t;\mathcal{B}^0)]\right\rangle \overset{(a)}{\leq} \frac{L}{2}\left(\left\|\nabla\mathcal{L}(\boldsymbol{\theta}_t)\right\|^2+\left\|\mathbb{E}_{\mathcal{B}^0}[\hat{\nabla}\mathcal{L}(\boldsymbol{\theta}_t;\mathcal{B}^0)]\right\|^2\right), \tag{27}$$

where $(a)$ applies the fact that $\langle a,b\rangle\leq\frac{1}{2}(\|a\|^2+\|b\|^2)$. Substitute equation 26, equation 27, and equation 6 back to equation 23, we have:

$$\mathbb{E}_t[\mathcal{L}(\boldsymbol{\theta}_{t+1})] \leq \mathcal{L}(\boldsymbol{\theta}_t) - \eta_t\left(1-\frac{\alpha}{2}-\frac{\eta_tL}{2}\left(1-\alpha+2\alpha^2L(2+r)\right)\right)\left\|\nabla\mathcal{L}(\boldsymbol{\theta}_t)\right\|^2$$

$$- \frac{\eta_t\alpha(1-\eta_t(1-\alpha)L)}{2}\left\|\mathbb{E}_{\mathcal{B}^0}[\hat{\nabla}\mathcal{L}(\boldsymbol{\theta}_t;\mathcal{B}^0)]\right\|^2$$

$$+ \frac{\eta_t\alpha\epsilon^2L^2d^2\left(1+2\eta_t\alpha Lr\right)}{8} + \frac{\eta_t^2L\sigma^2}{2}\left(\frac{(1-\alpha)^2}{K^1}+\frac{2(2+r)\alpha^2}{K^0}\right).$$

By setting $\eta_t=\eta\leq\min\{\frac{1}{(1-\alpha)L},\frac{2-\alpha}{1-\alpha+2\alpha^2L(2+r)}\}$, summing from $t=0,\dots,T-1$, and divide both side by $\eta C_1 T$, with $C_1=1-\frac{\alpha}{2}-\frac{\eta L}{2}\left(1-\alpha+2\alpha^2L(2+r)\right)$, we have:

$$\mathbb{E}[\left\|\nabla\mathcal{L}(\boldsymbol{\theta}_t)\right\|^2] \leq \frac{\mathcal{L}(\boldsymbol{\theta}_0)-\mathcal{L}_\star}{\eta C_1 T} + \frac{\alpha\epsilon^2L^2d^2\left(1+2\eta\alpha Lr\right)}{8C_1}$$

$$+ \frac{\eta L\sigma^2}{2C_1}\left(\frac{(1-\alpha)^2}{K^1}+\frac{2(2+r)\alpha^2}{K^0}\right). \tag{28}$$

The theorem is proved.

**Corollary G.13.** *By choosing*

$$\eta = \min\left\{\frac{1}{(1-\alpha)L}, \frac{2-\alpha}{1-\alpha+2\alpha^2 L(2+r)}, \sqrt{\frac{2(\mathcal{L}(\boldsymbol{\theta}_0)-\mathcal{L}_\star)}{TL\sigma^2\left(\frac{(1-\alpha)^2}{K^1}+\frac{2(2+r)\alpha^2}{K^0}\right)}}\right\}$$

*and*

$$\epsilon \le \left(\frac{32(\mathcal{L}(\boldsymbol{\theta}_0)-\mathcal{L}_\star)\sigma^2\left((1-\alpha)^2/K^1 + 2(2+r)\alpha^2/K^0\right)}{T}\right)^{1/4} \cdot \frac{1}{L^{3/4}d\sqrt{\alpha}},$$

*algorithm 1 converges with rate*

$$\mathbb{E}[\|\nabla\mathcal{L}(\boldsymbol{\theta}_t)\|^2] \le \sqrt{2L} \cdot \frac{\sqrt{\frac{(1-\alpha)^2}{K^1}+\frac{2(2+r)\alpha^2}{K^0}}}{2-\alpha-\eta L\left(1-\alpha+2\alpha^2 L(2+r)\right)} \cdot \sigma\sqrt{\frac{\mathcal{L}(\boldsymbol{\theta}_0)-\mathcal{L}_\star}{T}}$$

$$= \mathcal{O}\left(\frac{1}{\sqrt{T}}\sqrt{\frac{(1-\alpha)^2}{K^1}+\frac{2(2+r)\alpha^2}{K^0}}\right)$$

## G.6 Convergence analysis of Addax in smooth strongly convex setting

**Theorem G.14.** *Under Assumptions G.1, G.2, and G.4, by running algorithm 1 for $T$ iterations with $0 < \eta_t = \eta \le \frac{1}{2L}, \forall t$, the output satisfies*

$$\mathbb{E}_t[\|\boldsymbol{\theta}_T - \boldsymbol{\theta}_\star\|^2] \le \left(1-\frac{\eta_t\mu}{2}\right)^T \|\boldsymbol{\theta}_0 - \boldsymbol{\theta}_\star\|^2$$
$$+ \frac{\alpha^2(1/\mu + \eta_t)\epsilon^2 L^2 d^2}{\mu} + \frac{2\eta_t(1-\alpha)^2}{K^1\mu}\sigma^2 + \frac{2\eta_t\alpha^2 d}{K^0\mu}\sigma^2. \tag{29}$$

By Assumption G.4, with $\mu > 0$, we have:

$$\mathbb{E}_t[\|\boldsymbol{\theta}_{t+1} - \boldsymbol{\theta}_\star\|^2] = \mathbb{E}_t[\|\boldsymbol{\theta}_{t+1} - \boldsymbol{\theta}_t + \boldsymbol{\theta}_t - \boldsymbol{\theta}_\star\|^2]$$

$$= \|\boldsymbol{\theta}_t - \boldsymbol{\theta}_\star\|^2 + \mathbb{E}_t\left[\|\boldsymbol{\theta}_{t+1} - \boldsymbol{\theta}_t\|^2 + 2\langle\boldsymbol{\theta}_{t+1} - \boldsymbol{\theta}_t, \boldsymbol{\theta}_t - \boldsymbol{\theta}_\star\rangle\right]$$

$$\overset{(a)}{=} \|\boldsymbol{\theta}_t - \boldsymbol{\theta}_\star\|^2 + \eta_t^2\left\|(1-\alpha)\nabla\mathcal{L}(\boldsymbol{\theta}_t) + \alpha\mathbb{E}_{\mathcal{B}^0}[\hat{\nabla}\mathcal{L}(\theta_t;\mathcal{B}^0)]\right\|^2$$
$$+ \eta_t^2(1-\alpha)^2\mathbb{E}_{\mathcal{B}^1}[\|\nabla\mathcal{L}(\boldsymbol{\theta}_t) - \nabla\mathcal{L}(\theta_t;\mathcal{B}^1)\|^2] + \eta_t^2\alpha^2\text{Var}(\hat{\nabla}\mathcal{L}(\theta_t;\mathcal{B}^0))$$
$$- 2\eta_t\left\langle\boldsymbol{\theta}_t - \boldsymbol{\theta}_\star, (1-\alpha)\nabla\mathcal{L}(\boldsymbol{\theta}_t) + \alpha\mathbb{E}_{\mathcal{B}^0}[\hat{\nabla}\mathcal{L}(\theta_t;\mathcal{B}^0)]\right\rangle$$

$$\overset{(b)}{\le} \|\boldsymbol{\theta}_t - \boldsymbol{\theta}_\star\|^2 + 2\eta_t^2\|\nabla\mathcal{L}(\boldsymbol{\theta}_t)\|^2 + \frac{\eta_t^2\alpha^2\epsilon^2 L^2 d^2}{2} + \frac{\eta_t^2(1-\alpha)^2}{K^1}\sigma^2 + \frac{\eta_t^2\alpha^2 d}{K^0}\sigma^2$$
$$- 2\eta_t\langle\boldsymbol{\theta}_t - \boldsymbol{\theta}_\star, \nabla\mathcal{L}(\boldsymbol{\theta}_t)\rangle - 2\alpha\eta^t\left\langle\boldsymbol{\theta}_t - \boldsymbol{\theta}_\star, \mathbb{E}_{\mathcal{B}^0}[\hat{\nabla}\mathcal{L}(\theta_t;\mathcal{B}^0)] - \nabla\mathcal{L}(\boldsymbol{\theta}_t)\right\rangle$$

$$\overset{(c)}{\le} \|\boldsymbol{\theta}_t - \boldsymbol{\theta}_\star\|^2 + 4\eta_t^2 L\left(\mathcal{L}(\boldsymbol{\theta}_t) - \mathcal{L}_\star\right) - \eta_t\left(2(\mathcal{L}(\boldsymbol{\theta}_t) - \mathcal{L}_\star) + \mu\|\boldsymbol{\theta}_t - \boldsymbol{\theta}_\star\|^2\right)$$
$$- 2\alpha\eta^t\left\langle\boldsymbol{\theta}_t - \boldsymbol{\theta}_\star, \mathbb{E}_{\mathcal{B}^0}[\hat{\nabla}\mathcal{L}(\theta_t;\mathcal{B}^0)] - \nabla\mathcal{L}(\boldsymbol{\theta}_t)\right\rangle$$
$$+ \frac{\eta_t^2\alpha^2\epsilon^2 L^2 d^2}{2} + \frac{\eta_t^2(1-\alpha)^2}{K^1}\sigma^2 + \frac{\eta_t^2\alpha^2 d}{K^0}\sigma^2, \tag{30}$$

where $(a)$ substitutes the update of $\boldsymbol{\theta}$ and takes expectations to $\boldsymbol{g}^0, \boldsymbol{g}^1$ and conditions on $\boldsymbol{\theta}_t$;$(b)$ plugs in equation 7 to the second term, and the third term follows from Lemma G.6; $(c)$ uses the fact that $\mathcal{L}(\boldsymbol{\theta}) \le \mathcal{L}(\boldsymbol{\theta}_\star) + \frac{1}{2L}\|\nabla\mathcal{L}(\boldsymbol{\theta})\|^2$ for convex and smooth $\mathcal{L}(\cdot)$. This implies that $\|\nabla\mathcal{L}(\boldsymbol{\theta})\|^2 \le 2L(\mathcal{L}(\boldsymbol{\theta}) - \mathcal{L}(\boldsymbol{\theta}_\star))$ and applies to the second term. We apply Assumption G.4 to the sixth term in $(c)$ by setting $\boldsymbol{\theta}, \boldsymbol{\theta}'$ to $\boldsymbol{\theta}_\star$ and $\boldsymbol{\theta}_t$, respectively. The middle term can be further bounded as:

$$-2\left\langle\boldsymbol{\theta}_t - \boldsymbol{\theta}_\star, \mathbb{E}_{\mathcal{B}^0}[\hat{\nabla}\mathcal{L}(\theta_t;\mathcal{B}^0)] - \nabla\mathcal{L}(\boldsymbol{\theta}_t)\right\rangle$$

$$\overset{(a)}{\leq} \frac{\mu}{2\alpha} \|\boldsymbol{\theta}_t - \boldsymbol{\theta}_\star\|^2 + \frac{2\alpha}{\mu} \left\| \mathbb{E}_{\mathcal{B}^0}[\hat{\nabla}\mathcal{L}(\theta_t; \mathcal{B}^0)] - \nabla\mathcal{L}(\theta_t) \right\|^2$$

$$\leq \frac{\mu}{2\alpha} \|\boldsymbol{\theta}_t - \boldsymbol{\theta}_\star\|^2 + \frac{\alpha\epsilon^2 L^2 d^2}{2\mu}, \tag{31}$$

where $(a)$ applies Young's inequality for inner products, i.e., $-2\langle \mathbf{a}, \mathbf{b} \rangle \leq \frac{1}{\tau}\|\mathbf{a}\|^2 + \tau\|\mathbf{b}\|^2$ with $\tau = \frac{2\alpha}{\mu}$ in our case. Then we have:

$$\mathbb{E}_t[\|\boldsymbol{\theta}_{t+1} - \boldsymbol{\theta}_\star\|^2] \leq \left(1 - \frac{\eta_t\mu}{2}\right) \|\boldsymbol{\theta}_t - \boldsymbol{\theta}_\star\|^2 - 2\eta_t(1 - 2\eta_t L)\left(\mathcal{L}(\boldsymbol{\theta}_t) - \mathcal{L}_\star\right)$$

$$+ \frac{\eta_t\alpha^2(1/\mu + \eta_t)\epsilon^2 L^2 d^2}{2} + \frac{\eta_t^2(1-\alpha)^2}{K^1}\sigma^2 + \frac{\eta_t^2\alpha^2 d}{K^0}\sigma^2, \tag{32}$$

By setting $\eta_t \leq \frac{1}{2L}$, we have $1 - 2\eta_t L \geq 0$. Then we have:

$$\mathbb{E}_t[\|\boldsymbol{\theta}_{t+1} - \boldsymbol{\theta}_\star\|^2] \leq \left(1 - \frac{\eta_t\mu}{2}\right) \|\boldsymbol{\theta}_t - \boldsymbol{\theta}_\star\|^2 + \frac{\eta_t\alpha^2(1/\mu + \eta_t)\epsilon^2 L^2 d^2}{2}$$

$$+ \frac{\eta_t^2(1-\alpha)^2}{K^1}\sigma^2 + \frac{\eta_t^2\alpha^2 d}{K^0}\sigma^2, \tag{33}$$

Recursively apply the above equation 33 and sum from $t = 0$ to $T - 1$ by setting $\eta_t = \eta \leq \frac{1}{2L}$, we have

$$\mathbb{E}[\|\boldsymbol{\theta}_T - \boldsymbol{\theta}_\star\|^2] \leq \left(1 - \frac{\eta\mu}{2}\right)^T \|\boldsymbol{\theta}_0 - \boldsymbol{\theta}_\star\|^2 + \sum_{j=0}^{T-1} \left(1 - \frac{\eta\mu}{2}\right)^j \frac{\eta\alpha^2(1/\mu + \eta)\epsilon^2 L^2 d^2}{2}$$

$$+ \sum_{j=0}^{T-1} \left(1 - \frac{\eta\mu}{2}\right)^j \left(\frac{\eta^2(1-\alpha)^2}{K^1}\sigma^2 + \frac{\eta^2\alpha^2 d}{K^0}\sigma^2\right)$$

$$\overset{(a)}{\leq} \left(1 - \frac{\eta\mu}{2}\right)^T \|\boldsymbol{\theta}_0 - \boldsymbol{\theta}_\star\|^2 + \frac{\alpha^2(1/\mu + \eta)\epsilon^2 L^2 d^2}{\mu}$$

$$+ \frac{2\eta(1-\alpha)^2}{K^1\mu}\sigma^2 + \frac{2\eta\alpha^2 d}{K^0\mu}\sigma^2 \tag{34}$$

where (a) comes from $0 \leq 1 - \frac{1-\eta\mu}{2} < 1$. This completes the proof

**Corollary G.15.** *By choosing* $\eta = \min\left\{\frac{1}{2L}, \frac{2}{\mu T}\ln\left(T\frac{\mu^2\|\boldsymbol{\theta}^0 - \boldsymbol{\theta}_\star\|^2}{4\left(\frac{(1-\alpha)^2}{K^1} + \frac{\alpha^2 d}{K^0}\right)\sigma^2}\right)\right\}$ *and*

$$\epsilon \leq \frac{\sigma}{Ld}\sqrt{\frac{2\left(\frac{(1-\alpha)^2}{K^1} + \frac{\alpha^2 d}{K^0}\right)}{T\alpha(1/\mu + \eta\alpha)}},$$

*algorithm 1 converges with rate*

$$\mathbb{E}[\|\boldsymbol{\theta}_T - \boldsymbol{\theta}_\star\|^2] \leq \frac{9}{\mu T}\ln\left(T\frac{\mu^2\|\boldsymbol{\theta}^0 - \boldsymbol{\theta}_\star\|^2}{4\left(\frac{(1-\alpha)^2}{K^1} + \frac{\alpha^2 d}{K^0}\right)\sigma^2}\right)\left(\frac{(1-\alpha)^2}{K^1} + \frac{\alpha^2 d}{K^0}\right)\sigma^2$$

$$= \mathcal{O}\left(\frac{\ln(T)}{T}\left(\frac{(1-\alpha)^2}{K^1} + \frac{\alpha^2 d}{K^0}\right)\right).$$

