# OpenReview forum: "Addax: Utilizing Zeroth-Order Gradients to Improve Memory Efficiency and Performance of SGD for Fine-Tuning Language Models"
_ICLR.cc/2025/Conference — ICLR 2025 Poster_

### Official Review · Reviewer_dTUU · 2024-10-20

**Soundness:** 3
**Presentation:** 3
**Contribution:** 2
**Rating:** 6
**Confidence:** 4

**Summary:**

Authors propose Addax, algorithm which adaptively combined zero-order (MeZO) and in place first-order (IP-SGD) optimization methods, to achieve LLM fine-tuning with low memory requirement.

**Strengths:**

An excellent description of problems with MeZO and IP-SGD.
Better results than MeZO while keeping memory consumtion low.

**Weaknesses:**

Authors do memory profiling without gradient checkpointing, a very common technique to lower memory requirements for storing activations for backward computation. I believe this overestimates the memory requirements of IP-SGD.
The main selling point is that Addax has lower memory requirements than IP-SGD. However, this selling point vanishes the moment we turn gradient checkpointing on.

A minor selling point of Addax is that it sometimes produces better results even when zero-order gradients are computed concurrently with first-order ones (Addax-WA). However, it is hard to find which results actually belong to Addax-WA in the paper. E.g. section 4 says "we plot the convergence curves of Addax-WA and MeZO using the same batch size in Figure 11", but figure 11 label says "Addax with 4×
less first-order samples achieves a convergence speed similar to SGD", which suggest we are not running Addax-WA here.

Nitpick: Figure 4 just feels wrong, because SGD should always have higher memory requirements than IP-SGD).

**Questions:**

How would results look like when we run IP-SGD with gradient checkpointing?

---

> ### Author Response · Authors · 2024-11-22
>
> Thank you for your detailed feedback on our work. We respond to your comments below:
>
> $\quad$
> > Authors do memory profiling without gradient checkpointing, a very common technique to lower memory requirements for storing activations for backward computation. I believe this overestimates the memory requirements of IP-SGD. The main selling point is that Addax has lower memory requirements than IP-SGD. However, this selling point vanishes the moment we turn gradient checkpointing on..... How would the results look like when we run IP-SGD with gradient checkpointing?
>
> Thank you for raising this point. We would like to emphasize that gradient checkpointing benefits both IP-SGD and Addax (This is because Addax runs IP-SGD on shorter sequences).  In particular, we observed that when we incorporate gradient checkpointing, Addax continues to demonstrate lower memory requirements than IP-SGD. To illustrate this further, we conducted two sets of experiments.
>
> In the first experiment, we repeated the fine-tuning process with OPT-13B on the CB dataset, enabling gradient checkpointing for both IP-SGD and Addax. We used the same batch size and $(K^1, K^0)$ settings as in our previous experiments. The results of these experiments are presented below.
>
>
> | Metrics               | IP-SGD | IP-SGD (with Gradient Checkpointing) | Addax | Addax (with Gradient Checkpointing) |
> |-----------------------|:------:|:-------------------------------:|:-----:|:------------------------------:|
> | Accuracy (%)          |  85.7  |               85.7              |  89.3 |              89.3              |
> | Batch Size/(K^1, K^0) |    2   |                2                | (4,6) |              (4,6)             |
> | Memory (GB)           |  37.7  |               33.3              |  39.2 |              29.2              |
> | Time (min)            |   2.2  |               3.63              |  13.5 |              15.9              |
>
> Notice that enabling gradient checkpointing reduces memory usage but increases convergence time, without affecting the final model quality. With gradient checkpointing enabled, the memory usage of IP-SGD decreased from 37.3 GB to 33.3 GB, while the memory usage of Addax decreased from 39.2 GB to 29.2 GB. This demonstrates that Addax indeed benefits more  from gradient checkpointing in terms of memory reduction. This advantage arises because Addax stores intermediate activations only for shorter sequences which require less memory.  On the other hand IP-SGD  requires storage of intermediate activations for longer sequences, which require more memory.
>
> In the second experiment, with gradient checkpointing enabled for both Addax and IP-SGD, we selected the largest possible batch sizes to maximize GPU memory utilization on a single A100 (40GB) GPU. The results of these experiments are presented below.
>
> | Metrics               | IP-SGD (Gradient Checkpointing) | Addax (Gradient Checkpointing) |
> |-----------------------|:-------------------------------:|:------------------------------:|
> | Accuracy (%)          |               87.5              |              91.1              |
> | Batch Size/(K^1, K^0) |                10               |             (24, 12)            |
> | Memory (GB)           |               35.5              |              38.8              |
> | Time (min)            |               6.1               |              24.6              |
>
> In this experiment, Addax achieves an accuracy of 91.1, while IP-SGD achieves only 87.5. Due to its data partitioning strategy based on the varying memory requirements of data points, Addax can accommodate a larger first-order batch size than IP-SGD, leading to significantly better final model quality. This result further illustrates that enabling gradient checkpointing benefits Addax as well.
>
>
> In conclusion, the use of gradient checkpointing would benefit Addax in terms of memory requirement. Gradient checkpointing trades convergence time for reduced memory usage without altering the optimization process or the final model quality. This is true both in Addax and IP-SGD.

---

> ### Author Response · Authors · 2024-11-22
>
> > A minor selling point of Addax is that it sometimes produces better results even when zero-order gradients are computed concurrently with first-order ones (Addax-WA). However, it is hard to find which results actually belong to Addax-WA in the paper. E.g. section 4 says "we plot the convergence curves of Addax-WA and MeZO using the same batch size in Figure 11", but figure 11 label says "Addax with 4× less first-order samples achieves a convergence speed similar to SGD", which suggest we are not running Addax-WA here.
>
>
> We are sorry for the confusion in the description of Figure 11. The convergence curves in Figure 11 is the result of Addax-WA, which is a specific case of our main algorithm, Addax.
>
> In Section 4, we said, *"Addax-WA and MeZO using the same batch size in Figure 11"*, referring to the configuration where $ K^1 + K^0 = 16$, which matches the batch size of 16 used for MeZO.
>
> In Figure 11, we wrote, *"Addax with 4× less first-order samples achieves a convergence speed similar to SGD"*. This refers to Addax-WA using a first-order batch size of $K^1 = 4$, compared to SGD's first-order batch size of 16. The phrase *"4× less first-order samples"* reflects this configuration and does not imply that we are not running Addax-WA. Thank you for pointing out this issue. We will ensure this is clarified in the paper for better understanding.
>
>
> In addition, we present the comparison between Addax-WA and IP-SGD for fine-tuning OPT-30B experiments here. This corresponds to the case where $L_T > L_{\text{max}}$. As noted, the table corresponds exactly to the results presented in Table 13 of our original paper. This is because the $L_{\text{max}}$ values for SST-2, WSC, and WIC are 58, 78, and 79, respectively and $L_T$ is 180, which align with the conditions for running Addax-WA. Across these three tasks, Addax-WA consistently outperforms IP-SGD in terms of final model quality.
>
> **Fine-tuning OPT-30B with Addax-WA and IP-SGD**
>
> | Metrics     | Task             | SST-2 |  WSC  |  WIC  |
> |-------------|------------------|:-----:|:-----:|:-----:|
> | Accuracy    | IP-SGD           |  91.2 |  63.5 |  66.5 |
> |             | Addax-WA         |  95.1 |  63.5 |  70.2 |
> | Batch Size  | IP-SGD           |   4   |   4   |   4   |
> | (K^1, K^0)  | Addax-WA         | (4,6) | (4,6) | (4,6) |
> | Memory (GB) | IP-SGD           |  65.2 |  66.3 |  66.5 |
> |             | Addax-WA         |  64.4 |  65.8 |   66  |
> | Time (min)  | IP-SGD           |  1.9  |  1.1  |  9.1  |
> |             | Addax-WA         |  9.7  |  1.5  |  23.5 |
>
> $\quad$
> >Nitpick: Figure 4 just feels wrong, because SGD should always have higher memory requirements than IP-SGD).
>
>
> Thank you for your careful observation. However, the claim that SGD always has higher memory requirements than IP-SGD is not entirely accurate. In certain scenarios, their **peak memory consumption** can be the same. The peak memory consumption of SGD is determined by two key components: 1. Activations stored during the forward pass 2. Gradients stored during backpropagation.
>
> In modern deep learning frameworks like PyTorch, activations are stored incrementally during the forward pass, with memory usage peaking just before backpropagation begins. During backpropagation, activations are freed as their gradients are computed, while memory usage for gradients gradually accumulates. Thus, the peak memory consumption of SGD can be approximately viewed as the maximum of the memory required for storing activations and the memory required for storing gradients.
>
> In IP-SGD, because gradients are updated in place, this approach reduces memory usage for storing gradients but does not affect the memory required for activations. When the memory required for storing activations exceeds that for gradients, the peak memory usage of SGD and IP-SGD becomes the same. This typically occurs in cases with large sequence lengths or large batch sizes, which explains why the memory curves of IP-SGD and SGD overlap in Figure 4.
>
> For a theoretical analysis of peak memory usage for IP-SGD and SGD during training, the following papers provide excellent explanations of memory consumption during fine-tuning. Please let us know if you have further question.
>
> `Zhang, Y., Li, P., Hong, J., Li, J., Zhang, Y., Zheng, W., ... & Chen, T. (2024). Revisiting zeroth-order optimization for memory-efficient llm fine-tuning: A benchmark. arXiv preprint arXiv:2402.11592.`
>
> `Luo, Q., Yu, H., & Li, X. (2024). BAdam: A Memory Efficient Full Parameter Training Method for Large Language Models. arXiv preprint arXiv:2404.02827.`

---

> > ### Comment · Reviewer_dTUU · 2024-11-22
> > **Response**
> >
> > Thanks for clearing the confusion about Addax-WA and memory usage.
> >
> > As for the comparison with gradient checkpointing, it looks very encouraging, and it would be great if it would be present in the paper (at least in the appendix).

---

> ### Author Response · Authors · 2024-11-23
>
> We appreciate your valuable and prompt feedback. In response, we have included experiments comparing gradient checkpointing in Appendix F.3 of the revised paper, with all changes marked in blue for easy reference. Please let us know if you have any further questions.

---

> > ### Comment · Reviewer_dTUU · 2024-11-25
> >
> > Raising my score to 6 (weak accept).

---

> ### Author Response · Authors · 2024-11-26
> **Thank you for raising your score**
>
> Thank you for reviewing our rebuttal and revising your score. We sincerely appreciate your thoughtful feedback, which has enhanced the quality and clarity of our paper.

---

### Official Review · Reviewer_5pxQ · 2024-11-01

**Soundness:** 3
**Presentation:** 4
**Contribution:** 3
**Rating:** 6
**Confidence:** 4

**Summary:**

This paper proposes Addax, which assigns different mini-batch of data to compute zeroth-order and first-order gradients respectively based on the sequence length of the data, and combines the two kinds of gradients to update the model parameter. Addax outperforms existing zeroth-order method MeZO in accuracy/F1 score and achieves comparable memory footprint with MeZO. Furthermore, it surpasses IP-SGD and Adam with significantly less memory requirement.

**Strengths:**

-	The motivation of the proposed method is well supported by the preliminary experiments.
-	This paper provides theoretical guarantee that the propose method enjoys faster convergence and less restrictive hyper-parameter choices than MeZO.
-	This paper conducts extensive experiments to demonstrate its effectiveness on models with various sizes ranging from 350M to 70B, achieving superior performance both in test accuracy, memory and time efficiency.

**Weaknesses:**

- In Algorithm 1 step 7, the mini-batch $\beta^1$ is sampled from $D^1$, while in Line 298, it is claimed that $\beta^1$ is sampled from $\beta^0$, which is inconsistent.
- The performance of the proposed method is sensitive to the mixing constant $\alpha$ for combining the zeroth-order and first-order gradients. Also, it may be sensitive to the length threshold $L_T$ and the random seed.
- It would be better to further demonstrate the effectiveness of the proposed method on more challenged tasks such as commonsense reasoning tasks and mathematics tasks.

**Questions:**

Please refer to the Weaknesses.

---

> ### Author Response · Authors · 2024-11-22
>
> We are glad that you found our motivation clear, our theoretical analysis solid, and our experiments extensive. We also appreciate your detailed feedback on our work. We respond to your comments below:
>
> $\quad$
> > In Algorithm 1 step 7, the mini-batch $\beta^1$ is sampled from $D^1$, while in Line 298, it is claimed that $\beta^1$ is sampled from $\beta^0$, which is inconsistent.
>
> Thank you for pointing that out. We assume you are referring to $\beta^1$ as $\mathcal{B^1}$ and $\beta^0$ as $\mathcal{D}^0$. Please let us know if this interpretation is incorrect. We have corrected the typos in Line 298 in our revised version. The random batch $\mathcal{B^1}$ is drawn from $\mathcal{D}^1$.
>
> $\quad$
> > The performance of the proposed method is sensitive to the mixing constant
>  for combining the zeroth-order and first-order gradients. Also, it may be sensitive to the length threshold $L_T$  and the random seed.
>
> We agree that the performance of our method depends on the choice of alpha, and we have mentioned this in the limitation section. However, in our large autoregressive experiments, we did not perform extensive tuning of the hyper-parameter alpha. The search grid for alpha is small, including only five different values across all autoregressive experiments. Additionally, for most fine-tuning experiments, the runtime is relatively short (e.g., an average of  11.8 minutes for fine-tuning OPT-30B with Addax). Therefore, the hyperparameter search for the optimal alpha is not a significant burden and can be conducted efficiently.
>
> As for the threshold  $L_T$, as explained in Appendix D.6, we set $K^1$ and $K^0$ and chose the largest possible $L_T$ to avoid out-of-memory issues across datasets when fine-tuning OPT-30B, OPT-66B, and Llama-2 70B. Therefore, $L_T$ is simply chosen based on the given size of memory. This removes the need for further tuning of $L_T$ subsequent fine-tuning experiments.
>
> Finally, we would like to mention that the performance of our experiments is not sensitive to the selected random seed. To demonstrate this, we conducted an additional experiment using a different random seed. Specifically, we repeated the OPT-30B fine-tuning experiments with Addax as presented in Table 13, with the only change being the random seeds. The results confirm the robustness of Addax to variations in random initialization/seed.
>
> | Accuracy/F1 (%)                      | SST2 |  RTE | BoolQ |  WSC |  WIC | MultiRC | SQuAD |
> |-----------------------------------|:----:|:----:|:-----:|:----:|:----:|:-------:|:-----:|
> | Random seed 0 (reported in paper) | 95.1 | 85.9 |  82.3 | 63.5 | 70.2 |   67.8  |  88.0 |
> | Random seed 1                     | 94.2 | 86.2 |  82.9 | 63.5 | 69.0 |   69.0  |  89.0 |
> | Random seed 2                     | 95.2 | 85.6 |  82.7 | 63.5 | 69.8 |   67.8  |  88.2 |
> | Random seed 3                     | 95.1 | 85.6 |  82.2 | 63.5 | 69.2 |   68.6  |  88.8 |
> | Random seed 4                     | 95.1 | 84.5 |  83.5 | 63.5 | 69.1 |   66.1  |  88.0 |

---

> ### Author Response · Authors · 2024-11-22
>
> > It would be better to further demonstrate the effectiveness of the proposed method on more challenged tasks such as commonsense reasoning tasks and mathematics tasks.
>
> Thank you for your suggestion. We  conducted additional experiments on fine-tuning tasks involving mathematical reasoning and commonsense reasoning. Specifically, we followed the same setup described in our paper to fine-tune with the Elementary Mathematics subset of MMLU and the commonsense reasoning dataset PIQA. The results of these experiments are presented below.
>
> In these experiments, we used the setting where $L_T>L_{max}$, which means Addax operates as Addax-WA. For Addax, MeZO, and SGD, we selected their largest $(K^1, K^0)$/batch sizes to maximize GPU utilization on a single H100 (80GB) GPU. Addax achieved an accuracy of 34% on the Elementary Math subset of MMLU, outperforming both SGD and Adam, which each achieved 31%. For fine-tuning with the commonsense reasoning dataset PIQA, Addax achieved 80.4%, outperforming Adam and SGD, which achieved 79.2% and 78.9%, respectively.
>
> Additionally, BoolQ, which is already included in our experiments, is also a commonsense reasoning task. If you have further suggestions for additional experiments or analyses, we would be happy to consider and implement them.
>
> |              | Task       |      MMLU       |        PIQA        |
> |--------------|------------|:---------------:|:------------------:|
> |              | Base Model | Llama-2-7B-chat |       OPT-13B      |
> | Metrics      | Task Type  | elementary math | commense reasoning |
> | Accuracy (%) | Zero-shot  |       27.0      |        78.5        |
> |              | MeZO       |       29.0      |         79         |
> |              | SGD        |       31.0      |        78.9        |
> |              | Adam       |       31.0      |        79.2        |
> |              | Addax      |       34.0      |        80.4        |
> | Batch Size   | MeZO       |        32       |         32         |
> |              | SGD        |        10       |         10         |
> |              | Adam       |        8        |          8         |
> | (K^1, K^0)   | Addax      |     (10,12)     |       (10,12)      |
> | Memory (GB)  | MeZO       |       50.7      |        53.8        |
> |              | SGD        |       76.7      |        79.6        |
> |              | Adam       |      185.8      |        275.1       |
> |              | Addax      |       76.7      |         80         |
> | Time (min)   | MeZO       |       64.7      |        153.8       |
> |              | SGD        |       5.3       |         5.6        |
> |              | Adam       |       2.9       |         1.4        |
> |              | Addax      |       14.5      |         7.7        |

---

### Official Review · Reviewer_nhLb · 2024-11-03

**Soundness:** 3
**Presentation:** 3
**Contribution:** 2
**Rating:** 6
**Confidence:** 4

**Summary:**

This paper proposes a novel LLM fine-tuning methods, which try to combine the first-order (sgd) and zeroth-order (mezo) to efficiently fine-tune LLM. The experiments on OPT and LLaMA Illustrate that the proposed method can achieve a better performance than SGD and faster convergence than MeZO.

**Strengths:**

1. The intuition and motivation are very easy to understand and the paper is also easy to follow.

2. The paper provides the results on different models with different sizes to verify the performance of the proposed method, which also makes the paper more convincing.

**Weaknesses:**

1. The opinion that "SGD can match the performance of Adam in fine-tuning tasks." is not very reasonable to me. I think the main reason why the authors reached this conclusion is because the tasks in this paper are too easy (mainly focusing on classification tasks). If the authors conduct some experiments on math and code generation tasks, Adam could be better than vanilla SGD.

2. Ablation Study. We can notice that the paper provides the results of ip-sgd and addax in Table 1, 2, 3. And the main difference of these two methods is some long sequences use different optimizers. Specially, for addax, the long sequences use zero-order optimization. What is the result if we set the learning rate of these zeroth order optimization to 0 (That means we can skip these long sequences and no update with these data). I just want to see whether the zeroth-order optimization really works. I guess update too much times with sgd may cause overfitting and just skip some training steps (long sequences in addax) can improve the performance (even if don't use zeroth-order on this data). So I would like to see the results if we just skip these long sequence data in addax.

3. The paper mainly focuses on the tasks on SuperGulu. These tasks are too easy to LLM fine-tuning with first-order methods. It would be better if the authors could provide the results of complex tasks. This is just a suggestion and should not be a weakness.

**Questions:**

1. From my experience about MeZO, it usually begin to work after a large number of epochs. So I would like to ask whether Addax and SGD use a same epoch value?

---

> ### Author Response · Authors · 2024-11-22
>
> We appreciate your kind words about the clarity of our work and the extensiveness of our experiments. Thank you for your detailed feedback. Below, we address your comments:
>
> $\quad$
> > The opinion that "SGD can match the performance of Adam in fine-tuning tasks." is not very reasonable to me. I think the main reason why the authors reached this conclusion is because the tasks in this paper are too easy (mainly focusing on classification tasks). If the authors conduct some experiments on math and code generation tasks, Adam could be better than vanilla SGD.
>
> The observation that SGD can match Adam's performance in fine-tuning tasks has also been mentioned in prior research. For example, Zhang et al. (2024) provide a benchmark showing FO-SGD outperforming FO-Adam on COPA across four models and performing comparably on WinoGrande, with a gap of under 3% (Table 3, [1]). Similarly, Lv et al. (2023) state that SGD, despite its simplicity, can be a viable alternative to Adam for fine-tuning LLMs [2]. In Zhang et al.’s fine-tuning experiments with GPT-2 (Figure 6b, [3]), SGD achieved similar loss values as Adam, attributed to the strong initialization from pretraining. These findings align with our main message.
>
>
>
> To investigate further, we also conducted additional experiments on fine-tuning tasks involving *mathematical reasoning* and *commonsense reasoning*. Specifically, we followed the same setup described in our paper to fine-tune with the Elementary Mathematics subset of MMLU and the commonsense reasoning dataset PIQA. The results of these experiments are presented below.
>
> |              | Task       |      MMLU       |        PIQA        |
> |--------------|------------|:---------------:|:------------------:|
> |              | Base Model | Llama-2-7B-chat |       OPT-13B      |
> | Metrics      | Task Type  | elementary math | commense reasoning |
> | Accuracy (%) | Zero-shot  |       27.0      |        78.5        |
> |              | MeZO       |       29.0      |         79         |
> |              | SGD        |       31.0      |        78.9        |
> |              | Adam       |       31.0      |        79.2        |
> |              | Addax      |       34.0      |        80.4        |
> | Batch Size   | MeZO       |        32       |         32         |
> |              | SGD        |        10       |         10         |
> |              | Adam       |        8        |          8         |
> | (K^1, K^0)   | Addax      |     (10,12)     |       (10,12)      |
> | Memory (GB)  | MeZO       |       50.7      |        53.8        |
> |              | SGD        |       76.7      |        79.6        |
> |              | Adam       |      185.8      |        275.1       |
> |              | Addax      |       76.7      |         80         |
> | Time (min)   | MeZO       |       64.7      |        153.8       |
> |              | SGD        |       5.3       |         5.6        |
> |              | Adam       |       2.9       |         1.4        |
> |              | Addax      |       14.5      |         7.7        |
>
> In these experiments, we used the setting where $L_T>L_{max}$, which means Addax operates as Addax-WA. For Addax, MeZO, and SGD, we selected their largest $(K^1, K^0)$/batch sizes to maximize GPU utilization on a single H100 (80GB) GPU. Addax achieved an accuracy of 34% on the Elementary Math subset of MMLU, outperforming both SGD and Adam, which each achieved 31%. For fine-tuning with the commonsense reasoning dataset PIQA, Addax achieved 80.4%, outperforming Adam and SGD, which achieved 79.2% and 78.9%, respectively. These results also supports our claim that SGD can have a comparable performance than Adam in finetuning tasks.
>
> Another potential reason why SGD performs well could be the reduction of "block heterogeneity" in pre-trained models, as explored in [3]. In [3], the authors explain that SGD can perform on par with Adam in problems without block heterogeneity but tends to underperform when heterogeneity exists.
>
> It is important to note that we did not claim that *SGD matches* Adam's performance across all possible tasks. Instead, we stated that it *''can'' match* Adam, as observed consistently in all our experiments. We used the term *"can match"* to acknowledge that while our empirical results support this claim, we do not  have theoretical rigorous proof.
>
>
> `[1] Zhang, Yihua, et al. "Revisiting zeroth-order optimization for memory-efficient llm fine-tuning: A benchmark." arXiv preprint arXiv:2402.11592 (2024).`
>
> `[2] Lv, Kai, et al. "Full parameter fine-tuning for large language models with limited resources." arXiv preprint arXiv:2306.09782 (2023).`
>
> `[3] Zhang, Yushun, et al. "Why transformers need adam: A hessian perspective." arXiv preprint arXiv:2402.16788 (2024).`

---

> ### Author Response · Authors · 2024-11-22
>
> > Ablation Study. We can notice that the paper provides the results of ip-sgd and Addax in Table 1, 2, 3. And the main difference of these two methods is some long sequences use different optimizers. Specially, for Addax, the long sequences use zero-order optimization. What is the result if we set the learning rate of these zeroth order optimization to 0 (That means we can skip these long sequences and no update with these data). I just want to see whether the zeroth-order optimization really works. I guess update too much times with sgd may cause overfitting and just skip some training steps (long sequences in Addax) can improve the performance (even if don't use zeroth-order on this data). So I would like to see the results if we just skip these long sequence data in Addax.
>
>
>
> Thank you for raising this great question; it highlights an area we can clarify better in our paper. To address it, we conducted an additional ablation study and analyzed the results in two different regimes: $L_T > L_{max}$ and $L_T \leq L_{max}$.
>
>
> - In the first case ($L_T > L_{max}$), Addax operates as Addax-WA, running first- and zeroth-order optimization on all data points. In this case, IP-SGD on short data points (data points with length smaller than $L_T$) calcuates the first-order gradients estimates on all data points.
> - In the second case ($L_T < L_{max}$), Addax utilizes first-order gradient estimates for shorter sequences and zeroth-order estimates for longer sequences. Your suggested IP-SGD calculate the first-order gradient estimates on data with shorter sequence length (sequence length $\leq L_{T}$) and skip the data with longer sequence length (sequence length $> L_{T}$)
>
>
> The results of fine-tuning OPT-30B are presented below.
>
> **Case 1. $L_T > L_{max}$**
>
> | Metrics     | Task             | SST-2 |  WSC  |  WIC  |
> |-------------|------------------|:-----:|:-----:|:-----:|
> | Accuracy    | IP-SGD (L_T=180) |  91.2 |  63.5 |  66.5 |
> |             | Addax (L_T=180)  |  95.1 |  63.5 |  70.2 |
> | Batch Size  | IP-SGD (L_T=180) |   4   |   4   |   4   |
> | (K^1, K^0)  | Addax (L_T=180)  | (4,6) | (4,6) | (4,6) |
> | Memory (GB) | IP-SGD (L_T=180) |  65.2 |  66.3 |  66.5 |
> |             | Addax (L_T=180)  |  64.4 |  65.8 |   66  |
> | Time (min)  | IP-SGD (L_T=180) |  1.9  |  1.1  |  9.1  |
> |             | Addax (L_T=180)  |  9.7  |  1.5  |  23.5 |
>
>
> **Case 2. $L_T < L_{max}$**
> | Metrics     | Task             |  RTE  | BoolQ | MultiRC | SQuAD |
> |-------------|------------------|:-----:|:-----:|:-------:|:-----:|
> | Accuracy    | IP-SGD (L_T=180) |  85.2 |  63.1 |   53.8  |   88  |
> |             | Addax (L_T=180)  |  85.9 |  82.3 |   67.8  |   88  |
> | Batch Size  | IP-SGD (L_T=180) |   4   |   4   |    4    |   4   |
> | (K^1, K^0)  | Addax (L_T=180)  | (4,6) | (4,6) |  (4,6)  | (4,6) |
> | Memory (GB) | IP-SGD (L_T=180) |  79.5 |  79.3 |   80.8  |  71.3 |
> |             | Addax (L_T=180)  |  79.5 |  79.5 |   80.8  |  71.3 |
> | Time (min)  | IP-SGD (L_T=180) |  22.1 |  1.8  |   17.4  |  4.2  |
> |             | Addax (L_T=180)  |  23.1 |  25.5 |   48.6  |  11.3 |
>
> Please note that the table for Case 1 corresponds exactly to the results presented in Table 13 of our original paper. This is because the $L_{max}$ values for SST-2, WSC, and WIC are 58, 78, and 79, respectively, aligning with the conditions in Case 1.
>
> In both Case 1 and Case 2, Addax outperforms IP-SGD across all seven tasks. The ablation study underscores the effectiveness of the regularization effect introduced by zeroth-order optimization, showing that the improved model quality achieved by Addax is not solely due to skipping longer sequences but rather the combined effect of both first- and zeroth-order optimization.

---

> ### Author Response · Authors · 2024-11-22
>
> > The paper mainly focuses on the tasks on SuperGulu. These tasks are too easy to LLM fine-tuning with first-order methods. It would be better if the authors could provide the results of complex tasks. This is just a suggestion and should not be a weakness.
>
> We shared the results of additional fine-tuning experiments on the mathematical reasoning dataset MMLU and the commonsense reasoning dataset PIQA in our earlier response. If you have any further suggestions for experiments or analyses, we would be glad to consider and implement them.
>
>
> $\quad$
> >From my experience about MeZO, it usually begin to work after a large number of epochs. So I would like to ask whether Addax and SGD use a same epoch value?
>
> You are correct that MeZO requires a large number of steps to converge. However, Addax needs much smaller number of steps to converge. As mentioned in Appendix D.5 of our paper, for all large autoregressive experiments, we fixed the max number of training steps at 1,000 for Addax, SGD, and IP-SGD. For MeZO, we used 20,000 training steps, consistent with the original setup described in Malladi et al. (2023).

---

> > ### Comment · Reviewer_nhLb · 2024-12-03
> > **Thank you very much for your response**
> >
> > Thank you very much for your response and my concerns have been addressed.

---

### Official Review · Reviewer_covw · 2024-11-04

**Soundness:** 3
**Presentation:** 3
**Contribution:** 3
**Rating:** 6
**Confidence:** 4

**Summary:**

This paper presents Addax, a novel approach to fine-tuning large language models (LMs) with reduced memory usage and improved convergence speed. Addax aims to bridge the gap between two existing methods—IP-SGD, which is memory-intensive, and MeZO, which suffers from slow convergence due to its zeroth-order gradient estimation. By combining zeroth-order and first-order gradient estimates based on the memory consumption of different data points, Addax attempts to balance the benefits of each approach, reducing memory requirements without sacrificing performance.

**Strengths:**

The concept behind Addax is promising. By selectively applying zeroth-order and first-order updates depending on memory requirements, Addax cleverly attempts to sidestep some of the limitations of both IP-SGD and MeZO. The added benefit of using zeroth-order gradients as a regularizer for first-order gradients is an interesting approach that theoretically could lead to better model generalization.

- it assigns data points with higher memory demands to a zeroth-order gradient estimation, while data points with lower memory demands use a more standard first-order gradient approach.

- the wallclock time details provided are very helpful.

**Weaknesses:**

- Addax introduces an additional layer of complexity by dynamically selecting gradient types based on memory consumption, which may require extra engineering effort to implement effectively in various environments.
-  The algorithm involves splitting data into two subsets, resetting random seeds, and clearing gradients for each update step. This is quite complicated.
- While the hybrid gradient approach is novel, it’s unclear from the presented results to what extent this technique actually enhances performance, or to be specific, in results shown in figure 1 e.g. some of the tasks's improvement seem very marginal compared to MEZO.

**Questions:**

- the experiments done here are all done with high-end h100 gpus, I am curious who are actually the targeted users for Addax? does this work on lower-end GPUs? say 3090s?
- with the empirical improvement presented, and the complexity of the methods. I am curious if the authors could further list your No.1 selling point vs. well-established first-order finetuning methods, what would be your pitch to convince the community to adopt Addax in practise?

---

> ### Author Response · Authors · 2024-11-22
>
> Thank you for your constructive feedback. We are glad that you found Addax method novel and appreciate its potential to address key limitations in existing methods. Below, we respond to your main comments:
>
> $\quad$
> > Addax introduces an additional layer of complexity by dynamically selecting gradient types based on memory consumption, which may require extra engineering effort to implement effectively in various environments. The algorithm involves splitting data into two subsets, resetting random seeds, and clearing gradients for each update step. This is quite complicated.
>
> The implementation of Addax is straightforward and not overly complex. We have open-sourced our code with clear documentation and highlighted areas requiring modification. Moreover, MeZO, which forms the basis of our approach, is already widely used in the ML community. Based on these point and our experience, re-implementing Addax from MeZO would require only one to two hours of effort.
>
> In addition, in scenarios such as fine-tuning OPT-30B with a single H100 GPU on datasets like MultiRC or SQuAD, MeZO and Addax may be the only viable solutions. Thefore, one has no other option to choose from. Addax stands out by being faster than MeZO and achieving significantly better final model quality. Thus, the minor upfront engineering effort required to implement Addax is well worth it. As you mentioned, this effort is a one-time O(1) engineering overhead, making it both manageable and justified.
>
> $\quad$
>
> > While the hybrid gradient approach is novel, it's unclear from the presented results to what extent this technique actually enhances performance, or to be specific, in results shown in figure 1 e.g. some of the tasks's improvement seem very marginal compared to MEZO.
>
> To highlight the  clear benefits of Addax over MeZO, we here present Table 12 again. As can be seen, Addax outperforms MeZO in both accuracy/F1-score while it converges significantly faster.
>
>
>
> |                 | Task  | SST-2 | RTE   | CB    | BoolQ | WSC  | WIC   | MultiRC | ReCoRD | SQuAD |
> |-----------------|-------|-------|-------|-------|-------|------|-------|---------|--------|-------|
> | Accuracy/F1 (%) | MeZO  | 91.9  | 65.3  | 69.6  | 66.5  | 61.5 | 59.7  | 59.4    | 86.0   | 82.6  |
> |                 | Addax | 94.5  | 84.8  | 89.3  | 81.0  | 63.5 | 68.3  | 71.2    | 90.0   | 88.4  |
> | Memory (GB)     | MeZO  | 29.7  | 39.0  | 38.7  | 39.6  | 31.6 | 31.4  | 36.9    | 27.6   | 36.8  |
> |                 | Addax | 28.7  | 35.6  | 39.2  | 38.0  | 29.4 | 29.3  | 39.2    | 27.7   | 33.3  |
> | Time (Min)      | MeZO  | 222.5 | 289.2 | 182.8 | 255.3 | 40.3 | 103.9 | 363.8   | 31.7   | 245.5 |
> |                 | Addax | 10.2  | 23.2  | 13.5  | 35.5  | 2.1  | 17.4  | 5.3     | 0.9    | 10.8  |
>
>
> In the OPT-13B experiments, Addax outperforms MeZO across all nine tasks, achieving an average score of **81.22% compared to MeZO's 64.55%**. Besides, Addax **runs 15x faster**, while having comparable memory footprint to MeZO. In our OPT-30B experiments, Addax achieves an average score of 80.11, surpassing MeZO's 68.98. Similiarly, for OPT-66B and Llama-2-70B, Addax consistently outperforms MeZO in accuracy across all tasks (see Tables 14 and 15). In terms of time, Addax runs for an average of 15.1 minutes, whereas MeZO takes 700.9 minutes to fine-tune OPT-30B models. For fine-tuning OPT-66B models, Addax requires 29.8 minutes, while MeZO takes 492.6 minutes. In both experiments, Addax runs significantly faster than MeZO.

---

> ### Author Response · Authors · 2024-11-22
>
> > the experiments done here are all done with high-end h100 gpus, I am curious who are actually the targeted users for Addax? does this work on lower-end GPUs? say 3090s?
>
> We conducted our experiments mainly on A100 and H100 GPUs for two main reasons: (1) Their memory capacities (40 GB and 80 GB) are widely used and respected standards in both the research community and production environments, making them ideal for our study; and (2) Using consistent hardware across experiments allows for straightforward comparisons by readers.
>
> Also, our RoBERTa-large experiments were conducted on Tesla V100 GPUs, which are mid-range compared to A100 or H100 GPUs. We will clarify this in the revision. Having said that, we totally understand relevance of your question. To further demonstrate the compatibility of our approach with lower-end GPUs, we conducted additional experiments fine-tuning Llama-2-7B on RTX 3090 GPUs. The results are presented below.
>
> | Metrics         | Task            | SST-2 |  RTE  | BoolQ |  WIC  |
> |-----------------|-----------------|:-----:|:-----:|:-----:|:-----:|
> | Accuracy/F1 (%) | Zero-shot       |  58.0 |  62.5 |  66.0 |  50.2 |
> |                 | MeZO            |  94.4 |  63.2 |  70.2      |    63.0 |
> |                 | IP-SGD          |  95.1 |   87  |   *   |  70.5 |
> |                 | Addax (L_T=170) |  95.6 |  90.3 |  84.5 |  73.2 |
> | Batch Size      | MeZO            |   32  |   14  |   8   |   32  |
> |                 | IP-SGD          |   10  |   2   |   *   |   6   |
> | (K^1, K^0)      | Addax           | (4,6) | (4,6) | (4,6) | (4,6) |
> | Memory (GB)     | MeZO            | 19.8  |  23.7 |   24.1    |    23.2 |
> |                 | IP-SGD          |  24.2 |  22.2 |   *   |  23.0 |
> |                 | Addax           |  17.4 |  24.1 |  23.7 |  19.2 |
> | TIme (min)      | MeZO            |  489.2   |  592.4 |   43.5    | 657.9 |
> |                 | IP-SGD          |  12.9 |  6.9  |   *   |  10.4 |
> |                 | Addax           |  8.1  |  46.5 |  35.2 |  19.1 |
>
> In the experiments conducted on RTX 3090 GPUs, Addax outperformed IP-SGD, MeZO, and zero-shot baselines in accuracy across all tasks. Notably, IP-SGD failed to fine-tune BoolQ even with the smallest possible batch size due to out-of-memory issues. The total cost for running all experiments was 12 dollars for IP-SGD, 12 for Addax, and 46 for MeZO. While MeZO is feasible for running all experiments, running MeZO is significantly more expensive than Addax (because of MeZO's slow convergence), further demonstrating that Addax is both practical and resource-efficient.
>
>
> $\quad$
> > with the empirical improvement presented, and the complexity of the methods. I am curious if the authors could further list your No.1 selling point vs. well-established first-order fine-tuning methods, what would be your pitch to convince the community to adopt Addax in practise?
>
> Our short pitch is: *Addax is resource-efficient and can adapt to the available hardware memory, making it practical and accessible for a wide range of hardware setups. These benefits come with almost no loss in the performance of the fine-tuned model.*
>
> To elaborate, Addax optimizes **two critical resources—time and memory**. While MeZO offers memory efficiency, it requires significantly more training epochs, leading to higher compute costs. Conversely, methods like IP-SGD or standard SGD are less memory-efficient compared to Addax. Importantly, Addax not only achieves memory efficiency but also maintains or improves accuracy across most scenarios. This makes Addax a win-win solution for resource-constrained labs, balancing both cost-effectiveness and performance.
>
> As you also highlighted in the strengths, Addax smartly assigns data points with higher memory demands to zeroth-order gradient estimation, while utilizing first-order gradient estimation for less demanding data points. This adaptive strategy optimizes resource usage without compromising performance. Thus, one can fully utilize the available memory (by adjusting the threshold $L_T$).

---

> > ### Comment · Reviewer_covw · 2024-11-22
> > **Thank you!**
> >
> > Dear authors,
> > Thank you for the detailed replies. Unfortunately, your code link expired. I would be more convinced if I have taken a look at your implmentation and consider adjusting the score.
> > Thanks!

---

> ### Author Response · Authors · 2024-11-23
>
> Thank you for your prompt response. We apologize for the inconvenience caused by the expired code link. The updated link is now available: [https://anonymous.4open.science/r/Addax-62ED](https://anonymous.4open.science/r/Addax-62ED).
>
> Our implementation is primarily adapted from MeZO (Malladi et al., 2024). The Addax implementation is designed to be straightforward and not overly complex. Compared with the original MeZO implementation, we made the following major modifications to achieve Addax:
>
> - **Dataset Partitioning**: In lines 589–735 of [`run_2_loader.py`](https://anonymous.4open.science/r/Addax-62ED/large_models/run_2_loader.py), we partition the dataset based on sequence length, corresponding to lines 2–5 in Algorithm 1 of our paper.
> - **In-Place Backward Propagation**: In lines 762–774 of [`run_2_loader.py`](https://anonymous.4open.science/r/Addax-62ED/large_models/run_2_loader.py), we enable in-place backward propagation, corresponding to lines 9–12 in Algorithm 1.
> - **Update Rule Implementation**: In lines 1365–1420 of [`trainer_2_loader.py`](https://anonymous.4open.science/r/Addax-62ED/large_models/trainer_2_loader.py), we utilise the existing MeZO implementation to implement the update rule described in Equation (3) of our paper.
>
> We have open-sourced our code using the popular framework Transformer, allowing researchers and machine learning practitioners to easily adapt it for their own use cases. Additionally, although not shown here, we have implemented a version of Addax in JAX, further demonstrating its ease of implementation and maximum flexibility. Please do not hesitate to reach out if you have any further questions or require additional clarification.

---

> > ### Comment · Reviewer_covw · 2024-11-25
> > **Thank you!**
> >
> > Thank you for the pointers! i have raise my score to 6 acknowledging the value and effort authors put in this work.

---

> ### Author Response · Authors · 2024-11-26
> **Thank you for raising your score**
>
> Thank you for reviewing our rebuttal and raising your score. Your constructive feedback was invaluable in enhancing the clarity and quality of our paper.

---

> > ### Comment · Reviewer_covw · 2024-11-26
> > **Encouragement.**
> >
> > I raised my score, but I think this should not be an end to your effort, rather should be an encouragement.
> > Since I did take the time skimming your code/script, it looks at best to me like a hacked version of the MeZo opensource codebase. I did not run your recipe yet.
> > I strongly encourage the authors to clean up your code base, and make this a nice opensource project. This would truly benefit the community.

---

> ### Author Response · Authors · 2024-11-27
>
> Thank you for your valuable feedback and for raising your score! We're thrilled that you recognize the potential of our package to democratize access to AI. We acknowledge that our code is currently more suited for research purposes rather than production use.  Your input serves as an encouragement for us to refine our algorithm and make it more user-friendly and widely accessible. We would greatly appreciate any specific suggestions you have to help us achieve this goal. Please feel free to share any thoughts on how we can further improve the accessibility and ease of use of our code.

---

### Meta-Review · Area_Chair_d7vF · 2024-12-13

**Metareview:**

This paper introduces Addax, a novel fine-tuning algorithm for large language models (LLMs) that efficiently combines zeroth-order and first-order gradient optimization techniques to address memory and convergence challenges. The approach dynamically allocates zeroth-order gradients to memory-intensive sequences and first-order gradients to less demanding ones, optimizing memory usage without compromising performance. The hybrid gradient strategy is innovative, leveraging zeroth-order gradients as implicit regularizers, which improves generalization. The authors provide both theoretical guarantees and extensive empirical evidence, demonstrating faster convergence and superior accuracy compared to existing methods like MeZO and IP-SGD across a variety of models and tasks. The implementation's adaptability to hardware constraints and its open-source availability further enhance its practical appeal.

**Additional Comments On Reviewer Discussion:**

There were concerns about the complexity of Addax's implementation, the impact of zeroth-order optimization, sensitivity to hyperparameters, and the exclusion of gradient checkpointing in memory profiling. The authors responded thoroughly, providing additional ablation studies, experiments on challenging tasks, and evidence showing Addax's robustness and efficiency, even with gradient checkpointing. They clarified implementation details and demonstrated the method's practicality through improved documentation and extended comparisons. These responses adequately addressed the reviewers’ concerns

---

### Decision · Program_Chairs · 2025-01-22

Accept (Poster)